# INSIGHT-O3: EMPOWERING MULTIMODAL FOUNDATION MODELS WITH GENERALIZED VISUAL SEARCH

**Kaican Li**[1*], **Lewei Yao**[2*], **Jiannan Wu**[2*], **Tiezheng Yu**[2], **Jierun Chen**[2], **Haoli Bai**[2],
**Lu Hou**[2], **Lanqing Hong**[2], **Wei Zhang**[2], **Nevin L. Zhang**[1†]
[1]Hong Kong University of Science and Technology, [2]Huawei

## ABSTRACT

The ability for AI agents to "think with images" requires a sophisticated blend of reasoning and perception. However, current open multimodal agents still largely fall short on the reasoning aspect crucial for real-world tasks like analyzing documents with dense charts/diagrams and navigating maps. To address this gap, we introduce O3-BENCH, a new benchmark designed to evaluate multimodal reasoning with interleaved attention to visual details. O3-BENCH features challenging problems that require agents to piece together subtle visual information from distinct image areas through multi-step reasoning. The problems are highly challenging even for frontier systems like OpenAI o3, which only obtains 40.8% accuracy on O3-BENCH. To make progress, we propose INSIGHT-O3, a multi-agent framework consisting of a visual reasoning agent (vReasoner) and a visual search agent (vSearcher) for which we introduce the task of generalized visual search—locating relational, fuzzy, or conceptual regions described in free-form language, beyond just simple objects or figures in natural images. We then present a multimodal LLM purpose-trained for this task via reinforcement learning. As a plug-and-play agent, our vSearcher empowers frontier multimodal models (as vReasoners), significantly improving their performance on a wide range of benchmarks. This marks a concrete step towards powerful o3-like open systems. Our code and dataset can be found at https://github.com/m-Just/InSight-o3.

## 1 INTRODUCTION

Thinking with images is an important and very useful skill for multimodal agents (OpenAI, 2025c). The skill rests on two crucial and fundamental cognitive abilities: reasoning and perception. Recent efforts at developing such a skill based on open models mainly focus on the perception component, *e.g.*, searching for a particular object or figure in natural images and then answering a simple visual query about them (Wu & Xie, 2024; Shen et al., 2024; Li et al., 2025; Zhang et al., 2025b; Su et al., 2025a; Zheng et al., 2025; Wang et al., 2025c; Zhu et al., 2025b; Wang et al., 2025a; Lai et al., 2025). While this feature is useful, it is still far from being able to handle many real-world tasks that require deeper and more abstract reasoning. Typical examples include extracting information from complex reports and navigating through intricate maps. Solving these tasks often require both organized reasoning and focused attention to visual details scattered across an image (or images).

Currently, the reasoning capability of open multimodal models is still relatively weak in comparison with frontier proprietary models (Yue et al., 2024a; Yuan et al., 2025; Hao et al., 2025). This makes it very difficult to replicate the kind of reasoning-driven image-thinking behavior demonstrated by OpenAI o3 (OpenAI, 2025c). In this work, we take a concrete step towards building such an intelligent system with open models. First, we propose a new multimodal benchmark, O3-BENCH, to help better evaluate the general capability of multimodal models to think with images. Complementary to most of the existing benchmarks which only deal with object attributes and spatial relations in natural images (Wu & Xie, 2024; Wang et al., 2025g; Lai et al., 2025; Wang et al., 2025a), O3-BENCH consists of a set of high-quality, *reasoning-oriented* questions on images of *high information density*. The questions involve real-world tasks such as map navigation and cross-chart/diagram analysis that are highly challenging even for frontier systems like OpenAI o3. Compared with benchmarks like

---

*Equal contribution.     †Corresponding author (lzhang@cse.ust.hk).

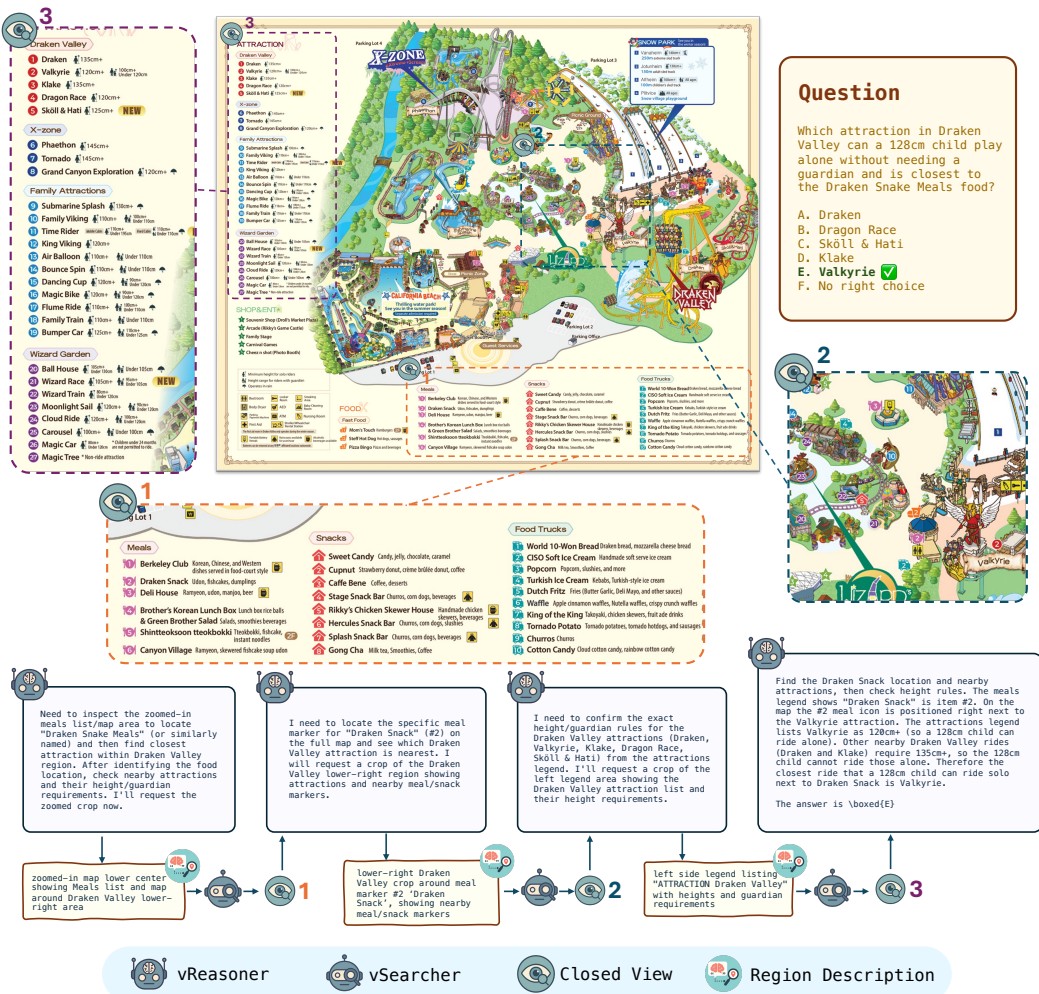

Figure 1: A multi-step visual reasoning example of INSIGHT-O3 on O3-BENCH. For clarity, the internal reasoning processes are omitted. More examples can be found in Appendix D.2.

MME-RealWorld (Zhang et al., 2024b), O3-BENCH is significantly harder, requiring the evaluated system to collect detailed visual information from *multiple* distinct image areas while performing *complex*, *interleaved* reasoning using the information collected in the process.

To make substantive progress on O3-BENCH, we introduce a multi-agent framework, INSIGHT-O3, that comprises a visual reasoning agent (**vReasoner**) and a visual search agent (**vSearcher**). The former is responsible for high-level reasoning and general image understanding, while the latter is to help vReasoner locate specific regions of interest and collect the visual information therein. As such, INSIGHT-O3 reduces the burden of a single agent, allowing us to build an o3-like system via divide-and-conquer. This kind of specialization has been shown to work in prior art (Dayan & Hinton, 1992; Zeng et al., 2023; Shen et al., 2023; Li et al., 2023a; Hong et al., 2023; Castrejon et al., 2024). In this work, we focus on vSearcher and how it should interact with a given vReasoner. Different from current practices (Wu & Xie, 2024; Lai et al., 2025) for natural images and discrete object references, we aim to solve *generalized visual search*, where the input image can be arbitrary, *e.g.*, a map, a poster, or a screenshot; and the referring description may specify a relational, fuzzy, or conceptual region, *e.g.*, "the area to the left of the wooden chair," and "the chart showing the company's revenue in the last decade," rather than a specific object or figure. Such fuzzy descriptions are more in line with how humans reason and direct their attention to a general region of interest.

To address this broader challenge, we present InSight-o3-vS, a vSearcher model specialized in generalized visual search through reinforcement learning. InSight-o3-vS combines multimodal understanding with spatial reasoning to localize regions described in completely free-form language. The

name of our model, InSight-o3-vS, reflects its dual role: providing deeper *insight* into multimodal semantics while bringing the target region *in sight* through precise localization. Our model empowers existing multimodal foundation models (as vReasoners) in a *plug-and-play* fashion, significantly improving the performance of frontier models across a wide range of benchmarks, *e.g.*, from 39.0% to 61.5% on O3-BENCH for GPT-5-mini (OpenAI, 2025a), and from 80.1% to 87.6% on V*-Bench for Gemini-2.5-Flash (Comanici et al., 2025).

To summarize, we make the following key contributions in this work:

- We propose a new benchmark, O3-BENCH, to evaluate complex, reasoning-oriented visual tasks. This benchmark features challenges like map navigation and cross-chart analysis, which require collecting information from multiple image areas and performing interleaved reasoning, making it significantly harder than existing benchmarks.

- We introduce INSIGHT-O3, a multi-agent framework that divides the task of "thinking with images" between a high-level reasoning agent (**vReasoner**) and a visual search agent (**vSearcher**). This divide-and-conquer design greatly simplifies the complex interleaved reasoning, allowing us to build o3-like systems that surpass OpenAI o3 across a variety of benchmarks.

- We present InSight-o3-vS, a specialized vSearcher model that excels at *generalized visual search*. It is designed to be a "plug-and-play" component that empowers existing multimodal foundation models, demonstrably and significantly improving the performance of frontier systems on a wide range of benchmarks including O3-BENCH.

## 2 RELATED WORK

We provide a brief overview of the most relevant related work in this section. For a more comprehensive discussion, please refer to Appendix A.

**Multimodal benchmarks.** Classical multimodal benchmarks (Goyal et al., 2017; Saikh et al., 2022; Liu et al., 2023; Ge et al., 2024) mainly test coarse image-level or salient-attribute recognition, where modern MLLMs are near-saturated (Bai et al., 2025b; Wang et al., 2025e). Recent multimodal reasoning benchmark split into (i) cognition-centric STEM benchmarks (Lu et al., 2023; Yue et al., 2024a;b) that emphasize multi-step/world-knowledge reasoning but use visually simple images, and (ii) perception-centric datasets (Wu & Xie, 2024; Zhang et al., 2024b; Lai et al., 2025) that require fine-grained recognition in high-resolution, text-rich scenes yet often limited to single-region lookups. Motivated by the "think with images" paradigm (OpenAI, 2025c), O3-BENCH jointly evaluates search/localization and higher-level reasoning on high-information-density charts and maps, requiring cross-region evidence aggregation via interleaved, multi-hop reasoning.

**Multimodal reasoning models.** Reinforcement learning (RL) has long been used to align model behavior with human preferences (Schulman et al., 2017). DeepSeek-R1 applies group relative policy optimization (GRPO) (Guo et al., 2025a; Shao et al., 2024b), reliably eliciting planning, reflection, and long chain-of-thought reasoning under simple rewards. Building on this idea, recent multimodal models (Yang et al., 2025c) adopt GRPO-style training and report strong gains, while cascaded RL stages (*e.g.*, InternVL3.5 (Wang et al., 2025e), Keye-VL1.5 (Yang et al., 2025a)) further push reasoning, approaching proprietary models. There are also pioneering work utilizing Python code execution to call various vision tools to solve tasks via divide-and-conquer or to help with reasoning (Gupta & Kembhavi, 2023; Surís et al., 2023; Ke et al., 2024). Nevertheless, most multimodal reasoners remain text-centric, overlooking the distinctive demands of visual reasoning.

**Visual search models.** Visual search is a core multimodal capability, requiring active region perception for fine-grained understanding. Early methods relied on external detectors or scripted workflows to localize regions and triggered tools via instruction tuning, leading to rigid outputs and typically single-round search (Wu & Xie, 2024). The "think with images" paradigm (OpenAI, 2025c) internalizes zoom/crop operations and has inspired end-to-end search (DeepEyes (Zheng et al., 2025)), synthetic warm-starts (Pixel-Reasoner (Su et al., 2025a)), and multi-turn RL (Mini-o3 (Lai et al., 2025)). Nonetheless, most systems still emphasize finding a single region in natural images, with limited support for multi-hop reasoning. We broaden this scope by decoupling visual search from visual reasoning and enabling multi-region search on arbitrary images.

## 3 O3-BENCH

We conceptualize "thinking with images" as an iterative perception-reasoning process. Perception focuses on searching and localizing task-relevant visual details, while reasoning needs to organize these cues into structured facts and performs higher-order inference (*e.g.*, planning, arithmetic, use of world knowledge) to complete the task. These two critical skills should be executed effectively and cooperate tightly to achieve strong performance. Existing benchmarks (Wu & Xie, 2024; Zhang et al., 2024b; Lai et al., 2025) primarily emphasize perception, where their questions hardly require multi-step reasoning and thus induce short reasoning chains. To bridge this gap, we introduce O3-BENCH, a benchmark that jointly assesses high-resolution perception and multi-hop visual reasoning. O3-BENCH is designed with two principles:

- **High resolution & high information density.** Images are large, high-resolution, cluttered, and information-dense, making evidence gathering genuinely non-trivial.
- **Multi-hop solution paths.** Questions require decomposing the goal, retrieving evidence from multiple regions, and composing it via intermediate steps before answering.

To instantiate these principles, O3-BENCH comprises two complementary domains: (1) *Composite charts*. Each image contains multiple heterogeneous charts (*e.g.*, bar/line/pie/tables). Our crafted questions demand cross-chart retrieval (series, axes, units), lightweight calculations (differences, ratios, aggregates), and consistency checks (scale, legend, time ranges) to derive the final answer. (2) *High-resolution digital maps*. The images typically include a map along with auxiliary components such as legends and building indices. We meticulously design questions that require visual search for targets (*e.g.*, matching symbols, categories, or toponyms) and spatial reasoning about relations and routes (*e.g.*, proximity constraints or shortest paths), conditioned on the provided context.

Overall, O3-BENCH comprises 204 images (117 charts, 87 maps) and 345 QA samples (163 for charts, 182 for maps) in total. The majority of samples fall into the more challenging *map* category, underscoring our prioritization of complex visual perception and multi-hop reasoning. The questions of O3-BENCH are multi-choice questions with six choices and one correct answer. Among the six choices, there are four distractors that appear in the image or look similar to the correct one. We also include an option F as *No Right Choice* if there are no correct options provided. Below, we present the construction process of O3-BENCH. For other details about O3-BENCH, see Appendix B.1.

### 3.1 SOURCE DATA COLLECTION

**Chart.** The chart images in O3-BENCH are curated from the "Diagram and Table" subset from MME-RealWorld (Zhang et al., 2024b) and the Internet. To ensure high information density, we run a layout detection model, PP-DocLayout_plus-L (Cui et al., 2025), on the candidate images and only keep those with at least 8 detected layouts. As a result, 256 of 2,539 images (from MME-RealWorld) that contain sufficient number of sub-figures and rich recognizable texts are left.

**Map.** We manually collect high-resolution digital maps from the Internet via keyword search. We center on the venue-level maps that require reading the provided legend/index and visually locating entities within the image to answer the question. We exclude all the country-, state-, or city-scale cartography that could be potentially answered with world knowledge. Through this process, we end up with 87 high-density map images spanning the categories over bus routes, campus, park, *etc*.

### 3.2 ANNOTATION PIPELINE

After the collection and initial filtering process, all images then undergo further manual screening to ensure clarity and completeness of key visual cues (*e.g.*, axes, units and legends). Next, we combine automated machine pre-annotation with human verification and authoring to generate the question-answer (QA) instances. The detailed process of data annotation is presented as follows.

**Machine pre-annotation.** To relieve the burden of human annotators and increase the data diversity, we first apply a three-step automated data pipeline to generate five questions for each image. (1) *Layout detection*. We divide the high-resolution images into several structured layouts (*e.g.*, tables, charts, legends) using PP-DocLayout_plus-L (Cui et al., 2025). For map images, we review

the predictions, correct erroneous regions, and supplement missing areas via manual annotation. (2) *Information extraction*. For each layout, we prompt Qwen2.5-VL-32B (Bai et al., 2025b) to produce a detailed caption for the layout and extract OCR text from it. In addition, we obtain global context by generating a caption and extracting the OCR text for the full image. (3) *Automated question synthesis*. For each image, we provide the layout set (with captions and OCR texts) and the global context to GPT-5 (OpenAI, 2025a) to generate five questions (with answers and explanations) that compose evidence from the provided layout regions. Note that we do not provide the full image to GPT-5, compelling it to focus on region-level details and encourages multi-hop composition. More details about the whole pre-annotation process can be found in Appendix B.2.

**Human annotation.** (1) *Filtering and validation*. Annotators start by discarding ill-posed or low-quality machine-generated QAs (*e.g.*, those with factual inconsistencies, ambiguous prompts, or spurious multi-hop reasoning). For the retained QAs, annotators verify that the six-option set contains exactly one unambiguous correct answer and confirm that the explanation faithfully, step by step, justifies the choice. The annotators also ensures that the target layouts are relevant to answering the question; these layouts are either derived from the explanation or added via manual annotation. (2) *Human-authored questions*. For information-dense images, machine-generated QAs often contain logical errors, wrong answers, or missed visual details. These QAs are reworked or completely rewritten by the annotators, adhering to our design principles: requiring fine-grained detail retrieval and multi-hop reasoning. Each QA includes a detailed explanation to aid verification and have exactly one unambiguous correct choice among the six.

**Difficulty filtering and secondary review.** We evaluate all candidate items with three strong proprietary MLLMs, *i.e.*, GPT-5-mini (OpenAI, 2025a), Gemini-2.5-Flash (Comanici et al., 2025) and Doubao-Seed-1.6 (Bytedance, 2025), using the same evaluation prompt. We discard any items solved by all three models to ensure difficulty. Subsequently, independent reviewers (distinct from the original annotators) conducts cross-verification: a final pass over the QAs and the explanations to confirm factual correctness, clarity, and formatting consistency. Finally, we confirm with experiments that attention to visual details is vital to good performance on O3-BENCH (see Appendix B.3).

## 4 INSIGHT-O3

In the previous section, we introduced O3-BENCH, a meticulously-crafted benchmark that require problem-solving systems to have both good reasoning and perception capabilities, as well as the ability to integrate them in a natural, synergetic manner. Recent approaches towards such systems mostly build upon a single MLLM agent which handles both reasoning and perception workloads within a single context window (Su et al., 2025a; Zheng et al., 2025; Lai et al., 2025). While this is reasonable for tasks primarily focusing on either reasoning or perception, the agent may struggle when the workloads are heavy and intertwined.

To address the issue, we propose INSIGHT-O3, a two-agent system that largely decouples the aforementioned burden by a visual reasoning agent (**vReasoner**) and a visual search agent (**vSearcher**). The former specializes in high-level, abstract reasoning (with some general image understanding), while the latter is mainly responsible for locating detailed visual information and presenting them to vReasoner. For instance, given a question, vReasoner first decomposes the question via reasoning, and, if needed, issues relevant image region descriptions to vSearcher; vSearcher then localizes the requested evidence (with help from tools like image cropping) and returns it for subsequent rounds until a final answer is produced. This process is illustrated in Figure 2(a). However, jointly training both agents in a system like INSIGHT-O3 is notoriously difficult—their objectives differ yet are highly interdependent, causing difficulties such as credit assignment across calls and non-stationary updates when both policies learn. Additionally, in our case, even the frontier open MLLMs, *e.g.*, Qwen2.5-VL (Bai et al., 2025b), tend to produce overly concise replies (Lai et al., 2025).

To avoid overcomplication, we consider a simpler, more manageable setting in this paper. Specifically, we delegate higher-order reasoning at training time to a strong external model (*e.g.*, GPT-5-mini as vReasoner) and focus on training vSearcher to cooperate with the given vReasoner effectively. This separation helps simplify optimization and improve data efficiency. Our experiment shows that a well-trained vSearcher can significantly improve the performance of a wide range of vReasoners as a plug-and-play callable agent. Furthermore, the resulting system may help syn-

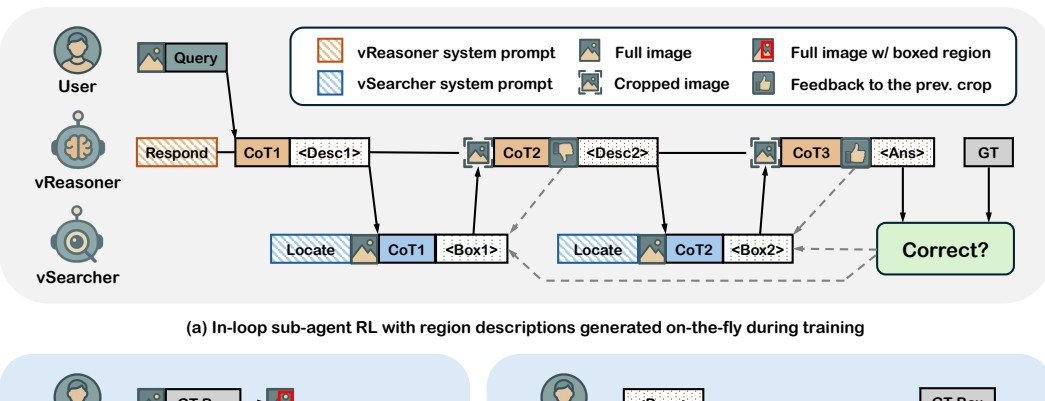

(a) In-loop sub-agent RL with region descriptions generated on-the-fly during training

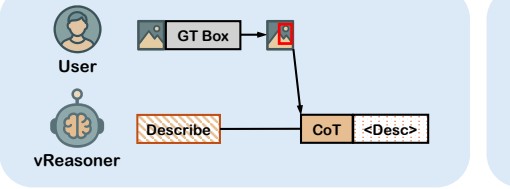

(b.1) Pre-generating region descriptions

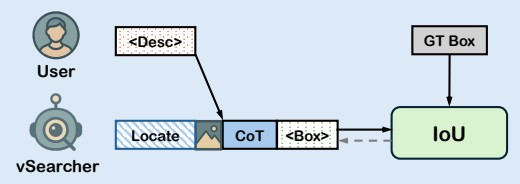

(b.2) Out-of-loop sub-agent RL with pre-generated region descriptions

Figure 2: **Training pipeline.** We use a hybrid RL algorithm to train vSearcher. **(a)** In the in-loop component, vReasoner generates visual search tasks on-the-fly during training as it tries to answer a user query. We use vReasoner's feedback and final answer correctness as supervision (denoted by dashed arrows) for vSearcher. **(b)** In the out-of-loop component, we use pre-generated descriptions with ground-truth bounding boxes, allowing us to train vSearcher efficiently via IoU supervision.

thesize multi-turn supervised-finetuning (SFT) traces with interleaved reasoning and visual search, paving the way for a larger, potentially unified model. While prior works (Su et al., 2025a; Jiang et al., 2025a) have explored role-playing to construct similar SFT traces, the data quality is often not very high as the role-playing agents are only loosely coordinated. In this respect, our approach helps strength this coordination with reinforcement learning (RL), as detailed next.

## 4.1 TRAINING ALGORITHM

We propose a hybrid sub-agent RL algorithm that consists of an *in-loop* component and an *out-of-loop* component to train vSearcher (see Figure 2). In the out-of-loop component, we pre-generate region descriptions with predefined bounding boxes, allowing us to train vSearcher very efficiently via direct IoU supervision. In the in-loop component, we use real descriptions generated on-the-fly during training by vReasoner. Compared with pre-generated tasks, these dynamically generated tasks are more aligned in nature with the tasks that vSearcher will see during inference time.

**Reward design.** For the out-of-loop RL, we use the following reward function for vSearcher:

$$r = \mathbb{I}[n_{\text{tool}} > 0] \cdot (\lambda_{\text{format}} \cdot r_{\text{format}} + \lambda_{\text{IoU}} \cdot r_{\text{IoU}}), \tag{1}$$

where $n_{\text{tool}}$ is the number of tool calls made by vSearcher, $r_{\text{format}} \in \{0, 1\}$ is the format reward, and $r_{\text{IoU}} \in (0, 1)$ is the IoU reward ($\lambda_{\text{format}}$ and $\lambda_{\text{IoU}}$ are weighting coefficients for the rewards). In Eq.(1), the IoU reward $r_{\text{IoU}}$ encourages vSearcher to propose an accurate region that matches the description. In addition, we encourage vSearcher to use image cropping[1] at least once to verify that the returned region matches the given region description. For every predicted box $b$ and the ground-truth box $b^*$, we define the IoU reward as $r_{\text{IoU}} = \max\{0, \text{IoU}(b, b^*) - \alpha\}/(1 - \alpha)$ where $\alpha \in (0, 1)$ controls the reward threshold. Boxes with an IoU less than $\alpha$ are not rewarded.

For the in-loop RL component, we replace the IoU reward $r_{\text{IoU}}$ in Eq.(1) with a pseudo IoU reward $\hat{r}_{\text{IoU}} \in \{0, 1\}$. We obtain $\hat{r}_{\text{IoU}}$ by asking vReasoner to rate each vSearcher's prediction, with the rating criterion being whether the prediction is relevant to the assigned task and can help vReasoner answer the user query. The rating $s \in \{0, 1\}$ is a binary score indicating if the prediction is helpful. However, as this rating is not always reliable (vReasoner may sometimes make mistakes), we further

---

[1]For simplicity, we only allow vSearcher to use the most essential image cropping tool. Our framework, however, does not impose such a constraint. It is easy to incorporate other kinds of tools for vSearcher to use.

incorporate outcome supervision as a safeguard. Let $c \in \{0, 1\}$ stand for whether the final answer of vReasoner is correct. The pseudo IoU reward is defined as $\hat{r}_{\text{IoU}} = \mathbb{I}[s = c = 1]$.

**Advantage estimation.** We follow GRPO (Shao et al., 2024b) to estimate the advantages for the out-of-loop RL component. As for the in-loop component, we normalize the rewards with respect to the *global* mean and standard deviation instead of the *group* mean and standard deviation since the concept of "group" no longer exists for the dynamically generated tasks. Formally, the advantage of an output token $o_t$ at time step $t$ is computed as $\hat{A}_t = [r - \text{mean}(\boldsymbol{r})]/\text{std}(\boldsymbol{r})$, where for the out-of-loop component, $\boldsymbol{r} = \{r_i\}_{i=1}^{G}$ with $G$ being the group size; and for the in-loop component, $\boldsymbol{r} = \{r_i\}_{i=1}^{N}$ with $N$ being the total number of visual search tasks generated on-the-fly by vReasoner.

**Objective function.** The objective function we use is based on GRPO (Shao et al., 2024b), with some modifications (*e.g.*, global advantage estimations) to incorporate the in-loop RL component. Given a policy model $\pi_\theta$, the old policy $\pi_{\theta_{\text{old}}}$, and a reference policy $\pi_{\text{ref}}$, the objective function for a batch of $M$ vSearcher outputs (including both in-loop and out-of-loop ones) is defined as

$$J(\theta) = \frac{1}{M} \sum_{i=1}^{M} \frac{1}{|o_i|} \sum_{t=1}^{|o_i|} \left\{ \min \left[ \gamma_t(\theta)\hat{A}_t, \text{clip}\left(\gamma_t(\theta), 1 - \epsilon, 1 + \epsilon\right) \hat{A}_t \right] - \beta \mathbb{D}_{\text{KL}}[\pi_\theta \| \pi_{\text{ref}}] \right\}, \quad (2)$$

where $\gamma_t(\theta) = \frac{\pi_\theta(o_{i,t}|q,o_{i,<t})}{\pi_{\theta_{\text{old}}}(o_{i,t}|q,o_{i,<t})}$ is the importance ratio of the output token $o_{i,t}$ given a query $q$ and all previous output tokens $o_{i,<t}$ including tool-response tokens. During training, we mask the loss for tool-response tokens as they are not generated by the policy model.

## 4.2 TRAINING DATA CONSTRUCTION

As high-resolution, information-dense images with challenging questions are difficult to collect, we construct training data by synthesizing collages (for in-loop RL) and generating pseudo visual search targets (for out-of-loop RL). The source data are mostly from existing VQA training datasets, which follow a largely different distribution from evaluation benchmarks we consider in our experiments.

**In-loop RL data.** A key criterion for in-loop RL data is that they must be difficult enough to incentivize visual search; otherwise vReasoner could simply answer on its own and vSearcher would receive no reward. To raise search difficulty and ensure meaningful credit assignment, we build image collages by stitching multiple low-to-medium-resolution images into a canvas. We construct collages from a filtered combination of Visual CoT (Shao et al., 2024a) and the V* training data (Wu & Xie, 2024), where each item provides a QA pair and a target bounding box. For each collage, we choose one target image (carrying the QA) and add several filler images as distractors. After difficulty filtering, we obtain 15,303 hard problems that vReasoner must rely on vSearcher to solve reliably. For more construction details and visualizations of the data, see Appendix C.1.

**Out-of-loop RL data.** We use InfographicVQA (Mathew et al., 2022) as the image source of the out-of-loop RL data. Most InfographicVQA images have high information density and feature more organic and diverse layouts than collages. We detect layout components in the source images with PP-DocLayout_plus-L (Cui et al., 2025). The candidate layout boxes are filtered and further processed, resulting in 10,186 high-quality layout boxes. We then use GPT-5-nano to generate concise, high-level region descriptions for each box, as illustrated in Figure 2(b.1). Through prompting, we make GPT-5-nano mimic the style it would use when invoking vSearcher in the in-loop setting. This process yields a set of *(image, region description, bbox)* which enables the out-of-loop RL. More construction details and visualizations are provided in the Appendix C.2.

## 5 EXPERIMENT

In our main experiments, we train Qwen2.5-VL-7B-Instruct (Bai et al., 2025b) as vSearcher under GPT-5-mini-2025-08-07 (OpenAI, 2025a) as vReasoner for balanced efficiency and reasoning capability. The resulting vSearcher, named InSight-o3-vS, is evaluated under various vReasoners including Gemini-2.5-Flash (Comanici et al., 2025). For comparison, we evaluate these vReasoners normally as standalone models and as vReasoner with the untrained Qwen2.5-VL-7B-Instruct as

Table 1: **Performance comparison with frontier models/systems.** All models/systems are evaluated under their default configurations unless specified otherwise. Performance of open models are mostly cited from the literature (Wu & Xie, 2024; Zheng et al., 2025; Wang et al., 2025a; Lai et al., 2025). Other results are averaged over 3 trials, except MME-RW$_{Lite}$ (single-trial). Small-size numbers indicate performance gaps between vReasoners w/ and w/o access to vSearchers. For more comprehensive benchmark results on O3-BENCH, see Appendix B.6.

| Model/System | V$^\star$-Bench | HR-Bench$_{4K}$ | Tree-Bench | VProbe$_{Hard}$ | MME-RW$_{Lite}$ | O3-Bench | Average |
|---|---|---|---|---|---|---|---|
| LLaVA-OV-7B | 70.9 | 62.0 | 37.3 | 13.4 | 48.5 | 20.2 | 42.1 |
| InternVL3.5-8B | 64.0 | 64.5 | 40.5 | 11.0 | 48.0 | 24.3 | 42.1 |
| Qwen2.5-VL-7B | 75.5 | 68.2 | 37.0 | 23.9 | 46.7 | 27.4 | 46.5 |
| Qwen3-VL-8B | 86.4 | 78.9 | 48.3 | 31.6 | 53.0 | 43.6 | 57.0 |
| Qwen3-VL-32B | 86.0 | 81.1 | 48.2 | 28.6 | 54.2 | 60.4 | 59.8 |
| Pixel Reasoner | 86.3 | 74.0 | 28.8 | 28.8 | - | - | - |
| DeepEyes | 83.3 | 73.2 | 37.5 | 35.1 | - | 27.0 | - |
| Mini-o3 | 88.2 | 77.5 | - | 48.0 | - | 29.1 | - |
| OpenAI o3 | 76.4 | 74.3 | 52.3 | 23.6 | 55.2 | 40.8 | 54.0 |
| GPT-4o | 68.6 | 65.1 | 47.4 | 26.4 | 51.2 | 28.0 | 47.8 |
| + Qwen2.5-VL-7B | 75.2 $_{+6.6}$ | 69.7 $_{+4.6}$ | 45.9 $_{-1.6}$ | 15.4 $_{-11.0}$ | 44.6 $_{-6.6}$ | 29.5 $_{+1.5}$ | 46.7 $_{-1.1}$ |
| + InSight-o3-vS | 80.4 $_{+11.8}$ | 76.2 $_{+11.1}$ | 49.5 $_{+2.1}$ | 25.5 $_{-1.1}$ | 50.1 $_{-1.1}$ | 36.4 $_{+8.4}$ | 53.0 $_{+5.2}$ |
| GPT-5-nano | 64.0 | 60.6 | 45.4 | 21.7 | 47.7 | 26.5 | 44.3 |
| + Qwen2.5-VL-7B | 70.1 $_{+6.1}$ | 67.3 $_{+6.7}$ | 45.7 $_{+0.3}$ | 18.2 $_{-3.5}$ | 44.9 $_{-2.8}$ | 25.3 $_{-1.2}$ | 45.3 $_{+1.0}$ |
| + InSight-o3-vS | 75.1 $_{+11.1}$ | 72.3 $_{+11.7}$ | 47.7 $_{+2.3}$ | 31.4 $_{+9.7}$ | 48.4 $_{+0.7}$ | 34.6 $_{+8.1}$ | 51.6 $_{+7.3}$ |
| GPT-5-mini | 73.8 | 72.0 | 54.6 | 26.4 | 56.1 | 39.0 | 53.7 |
| + Qwen2.5-VL-7B | 80.6 $_{+6.8}$ | 83.2 $_{+11.2}$ | 53.1 $_{-1.5}$ | 37.7 $_{+11.3}$ | 58.1 $_{+2.0}$ | 47.5 $_{+8.5}$ | 55.4 $_{+1.7}$ |
| + InSight-o3-vS | 86.9 $_{+13.1}$ | 86.7 $_{+14.7}$ | 54.1 $_{-0.5}$ | 41.2 $_{+14.6}$ | 59.0 $_{+2.9}$ | 61.5 $_{+22.5}$ | 64.9 $_{+11.2}$ |
| + InSight-o3-vS$^\dagger$ | 86.2 $_{+12.4}$ | 85.7 $_{+13.7}$ | 55.0 $_{+0.4}$ | 39.6 $_{+13.2}$ | 58.4 $_{+2.3}$ | 59.9 $_{+20.9}$ | 64.1 $_{+10.4}$ |
| Gemini-2.5-Flash$^\#$ | 72.8 | 75.0 | 48.9 | 17.9 | 55.6 | 49.8 | 53.4 |
| + Qwen2.5-VL-7B | 76.3 $_{+3.5}$ | 76.7 $_{+1.7}$ | 51.3 $_{+2.4}$ | 16.7 $_{-1.2}$ | 50.9 $_{-4.7}$ | 47.9 $_{-1.9}$ | 53.3 $_{-0.1}$ |
| + InSight-o3-vS | 80.8 $_{+8.0}$ | 80.2 $_{+5.2}$ | 52.1 $_{+3.2}$ | 19.8 $_{+1.9}$ | 55.1 $_{-0.5}$ | 58.0 $_{+8.2}$ | 57.7 $_{+4.3}$ |
| + InSight-o3-vS$^\dagger$ | 85.5 $_{+12.7}$ | 82.7 $_{+7.7}$ | 52.6 $_{+3.7}$ | 26.4 $_{+8.5}$ | 56.1 $_{+0.5}$ | 61.1 $_{+11.3}$ | 60.7 $_{+7.3}$ |
| Gemini-2.5-Flash | 80.1 | 83.5 | 49.9 | 39.6 | 56.5 | 60.4 | 61.7 |
| + Qwen2.5-VL-7B | 80.9 $_{+0.8}$ | 79.0 $_{-4.5}$ | 49.1 $_{+0.8}$ | 31.4 $_{-8.2}$ | 52.0 $_{-4.5}$ | 55.6 $_{-4.8}$ | 58.0 $_{-3.7}$ |
| + InSight-o3-vS | 87.6 $_{+7.5}$ | 82.3 $_{-1.2}$ | 50.1 $_{+0.2}$ | 36.2 $_{-3.4}$ | 56.3 $_{-0.2}$ | 69.7 $_{+9.3}$ | 63.7 $_{+2.0}$ |
| + InSight-o3-vS$^\dagger$ | 88.3 $_{+8.2}$ | 83.0 $_{-0.5}$ | 53.6 $_{+3.7}$ | 38.3 $_{-1.3}$ | 56.4 $_{-0.1}$ | 68.3 $_{+7.9}$ | 64.7 $_{+3.0}$ |

$^\dagger$ Trained with Gemini-2.5-Flash as vReasoner.
$^\#$ Image-size constraint set to 1280×1280px, roughly the maximum supported size for OpenAI models/systems via API.

vSearcher. We use the default configuration for proprietary models/systems (per official API) except from setting image detail to `high`[2]. More implementation details can be found in Appendix D.1.

**Evaluation datasets.** We evaluate a range of open and proprietary models/systems on the following benchmarks: (1) Natural-image benchmarks: V$^\star$-Bench (Wu & Xie, 2024), Tree-Bench (Wang et al., 2025a), and VisualProbe-Hard (Lai et al., 2025). (2) Mixed benchmarks: HR-Bench (Wang et al., 2025f) and MME-RealWorld (Zhang et al., 2024b). For efficient evaluation, we use the lite version of MME-RealWorld, which has 1,919 questions, still much heavier than the other benchmarks. (3) Our O3-BENCH. More information about the benchmarks can be found in Appendix G.

## 5.1 MAIN RESULTS

**Cross-domain performance improvement for frontier models.** As shown in Table 1, INSIGHT-O3 significantly improves frontier models such as GPT-5-mini and Gemini-2.5-Flash on most benchmarks. On average, the performance of GPT-5-mini has improved by 20.9% (relatively) with the help of our vSearcher (InSight-o3-vS). In particular, the accuracy of GPT-5-mini on O3-BENCH has improved from 39.0% to 61.5%. Meanwhile, INSIGHT-O3 also significantly outperforms their pre-RL counterparts, *i.e.*, vReasoner + Qwen2.5-VL-7B, across all the benchmarks. The results suggest that InSight-o3-vS is able to generalize *out-of-distribution* across various domains since its training data distribution is distinct from the evaluation data distributions. To gain more insight on how INSIGHT-O3 improves the baselines, see Appendix D.3 for a comparative analysis.

---

[2] When image detail is `high`, OpenAI scales down oversize images to ∼1280×1280px, as per OpenAI API (https://platform.openai.com/docs/guides/images-vision#calculating-costs). This is the maximum supported image resolution for OpenAI models/systems via API. Gemini-2.5-Flash API does not impose such a constraint, so we use a much larger, 3500×3500px budget, which is ample for the task.

Table 2: **Performance of Gemini-2.5-Flash (+ InSight-o3-vS$^\dagger$) under different maximum training/test image resolutions.** All results are averaged over 3 trials, except for MME-RW$_{Lite}$ (which is based on a single trial). Small-size numbers indicate performance gaps between settings with and without vSearcher. $^\dagger$ Trained with Gemini-2.5-Flash as vReasoner.

| Train res. | Test res. | V*-Bench | HR-Bench$_{4K}$ | Tree-Bench | VProbe$_{Hard}$ | MME-RW$_{Lite}$ | O3-Bench | Average |
|---|---|---|---|---|---|---|---|---|
| - | $1280^2$ | 72.8 | 75.0 | 48.9 | 17.9 | 55.6 | 49.8 | 53.3 |
| $1280^2$ | $1280^2$ | 85.5 $_{+12.7}$ | 82.7 $_{+7.7}$ | 52.6 $_{+3.7}$ | 26.4 $_{+8.5}$ | 56.1 $_{+0.5}$ | 61.1 $_{+11.3}$ | 60.7 $_{+7.4}$ |
| $3500^2$ | $1280^2$ | 85.3 $_{+12.5}$ | 81.3 $_{+6.3}$ | 53.1 $_{+4.2}$ | 22.6 $_{+4.7}$ | 55.1 $_{-0.5}$ | 58.8 $_{+9.0}$ | 59.4 $_{+6.1}$ |
| - | $3500^2$ | 80.1 | 83.5 | 49.9 | 39.6 | 56.5 | 60.4 | 61.7 |
| $1280^2$ | $3500^2$ | 87.8 $_{+7.7}$ | 84.3 $_{+0.8}$ | 52.1 $_{+2.2}$ | 39.6 $_{+0.0}$ | 56.4 $_{-0.1}$ | 67.8 $_{+7.4}$ | 64.7 $_{+3.0}$ |
| $3500^2$ | $3500^2$ | 88.3 $_{+8.2}$ | 83.0 $_{-0.5}$ | 53.6 $_{+3.7}$ | 38.3 $_{-1.3}$ | 56.4 $_{-0.1}$ | 68.3 $_{+7.9}$ | 64.7 $_{+3.0}$ |

Table 3: **Ablation study on reward design and advantage estimation.** All results are averaged over 3 trials. Small-size numbers indicate performance changes w.r.t. the proposed setting.

| Setting | V*-B. | VP$_{Hard}$ | O3-B. | Avg. |
|---|---|---|---|---|
| Proposed | 86.9 | **41.2** | **61.5** | **63.2** |
| w/o tool cond. | 86.4 $_{-0.5}$ | 39.3 $_{-1.9}$ | 60.6 $_{-0.9}$ | 62.1 $_{-1.1}$ |
| w/o feedback | 86.5 $_{-0.4}$ | 37.1 $_{-4.1}$ | 58.1 $_{-3.4}$ | 60.6 $_{-2.6}$ |
| w/o outcome | 86.9 $_{+0.0}$ | 38.7 $_{-2.5}$ | 60.9 $_{-0.6}$ | 62.2 $_{-1.0}$ |
| w/o GN | **87.3** $_{+0.4}$ | 36.8 $_{-4.4}$ | 61.3 $_{-0.2}$ | 61.8 $_{-1.4}$ |

Table 4: **Sensitivity analysis w.r.t. max. input resolution of vSearcher.** "#vSearch" is the number of vSearcher calls made by vReasoner per QA.

| Max. pixels | V*-B. | O3-B. | #vSearch |
|---|---|---|---|
| 0.8M | 85.3 | 56.0 | 2.82 |
| 1.6M | 86.7 | 60.5 | 2.75 |
| 3.2M | 89.4 | 62.3 | 2.69 |
| 6.4M | 86.4 | 61.9 | 2.66 |
| 12.8M | 86.9 | 61.5 | 2.58 |

**Generalization under different vReasoners.** We observe that InSight-o3-vS, which was trained as a sub-agent under GPT-5-mini, generalizes under other vReasoner models as well. As shown in Table 1, InSight-o3-vS improves the performance of a much smaller model, GPT-5-nano, from 21.7% to 31.4% on VisualProbe-Hard, from 26.5% to 34.6% on O3-BENCH, and from 44.3% to 51.6% overall. Under Gemini-2.5-Flash (a different model family), the advantage remains significant, showing about 7–10% lead over the baselines on V*-Bench and O3-BENCH. We have also explored training InSight-o3-vS under Gemini-2.5-Flash instead of GPT-5-mini, and observed similar generalization (see "+ InSight-o3-vS$^\dagger$" rows in Table 1). In few cases where InSight-o3-vS fails to improve the performance of the vReasoner, *e.g.*, GPT-4o and Gemini-2.5-Flash on VisualProbe-Hard, we see a sharp decrease in performance as we allow these models to call Qwen2.5-VL-7B. This suggests that these models are relatively weak at tool calling and multi-turn reasoning. In Appendix D.4, we present typical failure cases of INSIGHT-O3; we find that even GPT-5-mini (despite the good performance) still makes a lot of mistakes.

**Performance gaps on O3-BENCH.** Interestingly, on O3-BENCH, we observe that Gemini-2.5-Flash has a huge edge over GPT-5-mini (when they have no access to vSearcher), but on the other benchmarks, the edge is not so prominent—Gemini-2.5-Flash is even slightly worse than GPT-5-mini on Tree-Bench. This suggests that O3-BENCH is indeed quite different from the other benchmarks, and Gemini-2.5-Flash is particularly good at solving the kind of tasks in O3-BENCH on its own. Notably, with InSight-o3-vS, GPT-5-mini is able to drastically reduce the gap (from 21.4% to 8.2%) with Gemini-2.5-Flash, demonstrating the importance of thinking with images for addressing O3-BENCH, and also highlighting the effectiveness of our approach.

**Effect of input image resolution.** Comparing the results of Gemini-2.5-Flash in Table 1 under different maximum input image resolutions, we see that a much higher resolution offers clear advantages. However, the improvement brought by vSearcher is less when vReasoner can see clearer. In addition, we find that training under one resolution and evaluating under another seems to have little impact on the performance (see Table 2). Meanwhile, the input image resolution for vSearcher has less impact. Table 4 shows the performance of GPT-5-mini + InSight-o3-vS under varying maximum image resolution of InSight-o3-vS during evaluation. We can see that InSight-o3-vS is not sensitive to the resolution, maintaining decent performance on V*-Bench and O3-BENCH even when the maximum image resolution is only 0.8M (25% of that during training). When the resolution is low, the average number of vSearcher calls is relatively high. This is expected as low-resolution images often obscure fine details, making it harder for vSearcher to locate the targets.

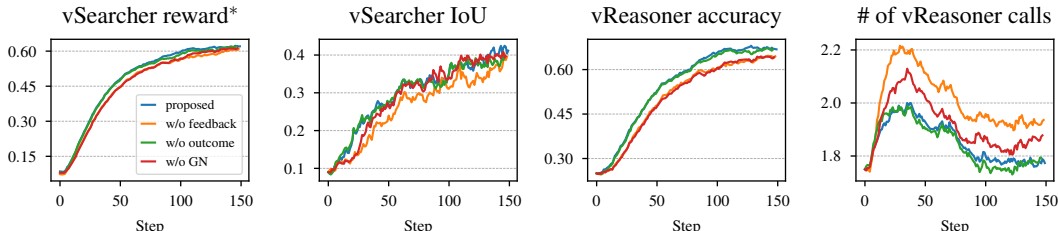

Figure 3: **Training dynamics of INSIGHT-O3.** The rightmost chart, "# of vReasoner calls", shows the average number of times vReasoner calls vSearcher per QA. * For fair comparison, the reward curves are plotted under the same setting ("w/o feedback") for all the settings.

## 5.2 ABLATION STUDY

**Hybrid RL training.** Table 5 shows the results of ablating the in-loop and the out-of-loop sub-agent RL components. Without the in-loop RL component, training is much faster (80%+ reduction in time per training step) but the final performance is worse on average. Dropping the out-of-loop RL component also hurts the performance; moreover, the training time increases due to more in-loop training. As mentioned in Section 4.2, the two RL components use different training data. The better performance of hybrid RL training can be partly explained by the combined use of the training data. Another contributing factor is the combination of two different sources of supervision (high-level vReasoner feedback and low-level IoU with ground-truth boxes). Overall, combining the two components leads to the best result.

Table 5: **Ablation study on hybrid RL training.** "I." and "O." stand for the in-loop and out-of-loop RL components, respectively. "T/step" is the average time per training step. All results are averaged over 3 trials. Small-size numbers indicate performance changes w.r.t. the untrained baselines.

| | I. | O. | V*-B. | VP$_{Hard}$ | O3-B. | T/step |
|---|---|---|---|---|---|---|
| **GPT-5-nano** | | | 70.1 | 18.2 | 25.3 | - |
| | ✓ | | **74.9** +4.8 | 23.9 +5.7 | 27.9 +2.6 | 846s |
| | | ✓ | 73.7 +3.6 | 25.1 +6.9 | 31.5 +6.2 | 130s |
| | ✓ | ✓ | 74.5 +4.4 | **27.4** +9.2 | **32.4** +7.1 | 693s |
| **GPT-5-mini** | | | 80.6 | 37.7 | 47.5 | - |
| | ✓ | | 86.4 +5.8 | 39.0 +1.3 | 59.6 +12.1 | 1223s |
| | | ✓ | 84.8 +4.2 | 41.2 +3.5 | 58.8 +11.3 | 105s |
| | ✓ | ✓ | **86.9** +6.3 | **41.2** +3.5 | **61.5** +14.0 | 941s |

**Reward design and advantage estimation.** In Table 3, we compare the setting we *proposed* in Section 4.1 on GPT-5-mini + InSight-o3-vS with the following ablated variants: "*w/o tool cond.*" drops the tool condition in the reward function; "*w/o feedback*" removes vReasoner feedback, only using outcome supervision for pseudo IoU reward; "*w/o outcome*" is the opposite of "w/o feedback"; and "*w/o GN*" drops the global normalization for advantage estimation. The originally proposed setting outperforms all the variants with a small average lead on the three benchmarks. The training dynamics under these settings are shown in Figure 3. As vSearcher learns to better locate the regions described by vReasoner, we observe that both the out-of-loop localization IoU and the in-loop vReasoner accuracy improve. The non-monotonic "# of vReasoner calls" shows two RL phases of INSIGHT-O3: vSearcher first learns to obey the formatting instructions, and then learns to localize more accurately (so vReasoner could solve the same problem with less vSearcher calls).

Although we encourage vSearcher to use the image-cropping tool, we find the average tool call count often ends up close to 1. There are two underlying reasons for this behavior. First, as mentioned by Zheng et al. (2025); Lai et al. (2025), Qwen2.5-VL-7B-Instruct is often reluctant to call the tool, and does not seem to know how to use the tool properly. Second, vReasoner usually describe a rough region that is not very hard for vSearcher to locate, so the tool has little utility for vSearcher.

## 6 CONCLUSION

In this work, we introduced O3-BENCH, a high-information-density benchmark that jointly evaluates visual localization and multi-hop reasoning. To advance the research on this challenging benchmark, we proposed INSIGHT-O3, a multi-agent framework that decomposes the "think with images" workflow into high-level reasoning (vReasoner) and visual search (vSearcher). We focus on the training of vSearcher via reinforcement learning to seamlessly cooperate with vReasoner. The specialized InSight-o3-vS can be used as a "plug-and-play" component for existing multimodal foundation models and helps significantly improve the performance of frontier models.

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

## A    ADDITIONAL DISCUSSION ON RELATED WORK

### A.1    MULTIMODAL BENCHMARKS

Classical multimodal benchmarks (Goyal et al., 2017; Hudson & Manning, 2019; Gurari et al., 2018; Saikh et al., 2022; Li et al., 2023b; Fu et al., 2023; Liu et al., 2023; Ge et al., 2024) primarily focus on coarse image-level understanding or target at salient-object attributes, on which current multimodal models (Bai et al., 2025b; Wang et al., 2025e; Yang et al., 2025a) show near-saturated performance. With growing attention to multimodal reasoning, more challenging benchmarks have emerged, which could be categorized as two groups. (1) Cognition-centric benchmarks. STEM (science, technology, engineering, and mathematics) benchmarks (Lu et al., 2023; Wang et al., 2024; Zhang et al., 2024a; Yue et al., 2024a;b) evaluate the model's multi-step reasoning, integration of world knowledge, and complex calculations to solve scientific problems, whereas the accompanying images are generally straightforward to interpret. (2) Perception-centric benchmarks. These benchmarks (Wu & Xie, 2024; Wang et al., 2025g; Zhang et al., 2024b; Lai et al., 2025) require fine-grained perception in high-resolution images and strong OCR recognition on text-rich scenes. Nevertheless, many questions become routine once the model precisely localizes the target region, allowing for single-glance solutions. Though the recent TreeBench (Wang et al., 2025a) evaluates second-order reasoning over object spatial transformations, depth ordering, and *etc*, it still centers on a single region in natural images, leaving cross-region evidence aggregation largely underexplored. With the emergence of the "think with images" paradigm (OpenAI, 2025c), we argue that a well-designed benchmarks should evaluate the joint perceptual and cognitive skills. Our proposed O3-BENCH fills the research gap by meticulously constructing hard questions on high-information density images (*e.g.*, composite graphs, maps), therefore requiring models to gather information from multiple, spatially distinct regions and to perform complex, interleaved reasoning.

### A.2    MULTIMODAL REASONING MODELS

Reinforcement learning (RL) (Schulman et al., 2017; Rafailov et al., 2023; Hu et al., 2025) has long been used to align the response of large language models (LLMs) and multimodal LLMs (MLLMs) with human preferences. Recently, DeepSeek-R1 (Guo et al., 2025a) creatively applied group relative policy optimization (GRPO) (Shao et al., 2024b) to LLMs, estimating the mean and variance of advantages across response groups under a simple reward signal. This strategy reliably elicits behaviors such as planning, thinking, and self-reflection, enabling long chain-of-thought (CoT) reasoning and moving toward more general-purpose reasoning capabilities. Building on this success, several works (Huang et al., 2025; Yang et al., 2025c; Meng et al., 2025; Chen et al., 2025a; Wang et al., 2025b; Shen et al., 2025a; Chen et al., 2025b; Deng et al., 2025; Wei et al., 2025; Peng et al., 2025; Wang et al., 2025d; Shen et al., 2025b) explore cold-start initialization and GRPO-based RL training for multimodal models (*e.g.*, Qwen2.5-VL (Bai et al., 2025b)) and report substantial gains on science- and math-oriented benchmarks. Concurrently, InternVL3.5 (Zhu et al., 2025a; Wang et al., 2025e) and Keye-VL1.5 (Yang et al., 2025b;a) further leverage cascaded, iterative RL stages to push the frontier of reasoning ability, achieving performance competitive with proprietary models (OpenAI, 2025a; Comanici et al., 2025). Nevertheless, current multimodal reasoning models (Xiaomi, 2025; Hong et al., 2025; Guo et al., 2025b; Du et al., 2025) still focus on text-centric reasoning, neglecting the distinctive demands of visual reasoning in multimodal scenarios.

### A.3    HIERARCHICAL AGENT FRAMEWORKS AND TOOL-USING MULTIMODAL AGENTS

Recent works have shown that hierarchical collaboration among specialized agents can significantly improve performance on complex tasks. Socratic Models (Zeng et al., 2023) and HuggingGPT (Shen et al., 2023) demonstrate early examples of LLM-based orchestration over expert models via language. More structured frameworks like CAMEL (Li et al., 2023a), MetaGPT (Hong et al., 2023), and HAMMR (Castrejon et al., 2024) explore role-based or modular specialization with coordinated task decomposition. Others, including AutoGen (Wu et al., 2024), HALO (Hou et al., 2025), Puppeteer (Dang et al., 2025), and AgentOrchestra (Zhang et al., 2025a), further extend this paradigm with explicit planning hierarchies, adaptive execution, and learned orchestration, consistently outperforming flat-agent baselines across diverse domains.

There are also recent works exploring equipping multimodal models with tool-use and programmatic reasoning to tackle complex visual tasks. VisProg (Visual Programming) (Gupta & Kembhavi, 2023) and ViperGPT (Surís et al., 2023) are two pioneering approaches that use large language models to generate and execute code for orchestrating vision modules. Beyond static program generation, other agents use LLMs as high-level controllers. For example, HuggingGPT (Shen et al., 2023) demonstrates an LLM (ChatGPT) orchestrating numerous specialized models (for vision, language, etc.) More recently, HYDRA (Ke et al., 2024) introduces a dynamic multi-stage framework for visual reasoning: it integrates an LLM-based planner and reasoner with a reinforcement learning–based controller that adapts the sequence of operations via feedback loops, yielding more reliable step-by-step reasoning. These tool-augmented systems highlight the power of combining learned vision-language models with external modules or code execution to improve flexibility and compositional reasoning. In contrast, our work (INSIGHT-O3) targets a complementary gap by introducing a dedicated visual search agent that can be invoked by reasoning agents to locate fine-grained, conceptually described regions within images. This specialized capability, absent in prior tool-using frameworks, allows an INSIGHT-O3-enabled system to pinpoint relevant visual details based on free-form descriptions, thereby enhancing multimodal reasoning with more precise visual understanding.

## A.4 VISUAL SEARCH MODELS

Visual search is an important functionality in the multimodal domain, requiring the models to perform active perception over regions of interest (RoIs) for fine-grained visual understanding. Early approaches (Wu & Xie, 2024; Shao et al., 2024a; Qi et al., 2024; Hu et al., 2024; Li et al., 2025) rely on external tools or predefined workflows for region localization and use instruction tuning to trigger tool use. These models exhibit rigid output patterns and typically support only a single round of visual search, which limits their effectiveness in complex scenes. Recently, the milestone OpenAI o3 (OpenAI, 2025c) established the "think with images" paradigm, in which image manipulations (*e.g.*, zooming, cropping) are internalized as intrinsic capabilities, enabling image–text interleaved reasoning. The community has rapidly turn the attention to the promising field. DeepEyes (Zheng et al., 2025) exploits the inherent grounding ability of MLLMs and incentivizes visual search via end-to-end reinforcement learning. Pixel-Reasoner (Su et al., 2025a) improves search accuracy by warm-start instruction tuning on synthesized data with error-induced self-correction trajectories. Mini-o3 (Lai et al., 2025) introduces an over-turn masking technique during RL to encourage multi-turn interaction, markedly enhancing reasoning adaptability and diversity. Other lines of work (Zhao et al., 2025; Zhang et al., 2025c; Liu et al., 2025) resort to write codes for executing multiple image manipulations (*e.g.*, cropping, rotation, enhancement), pointing to an open-ended toolkit for visual reasoning. Although effective, recent advanced methods (Zhu et al., 2025b; Fan et al., 2025; Zhang et al., 2025b; Su et al., 2025b; Ni et al., 2025; Cheng et al., 2025; Jiang et al., 2025a;b; Wang et al., 2025c;h) still largely prioritize locating a single region on natural images, which sidelines the model's capacity for reasoning. This paper broadens the capability scope of visual search models by decoupling visual reasoning and visual search agents, allowing the receiving of any images and searching of multiple distinct regions.

## B ADDITIONAL INFORMATION ON O3-BENCH

### B.1 BENCHMARK STATISTICS

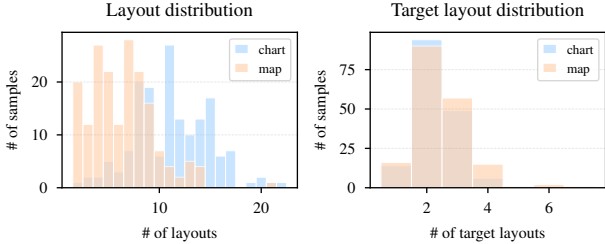
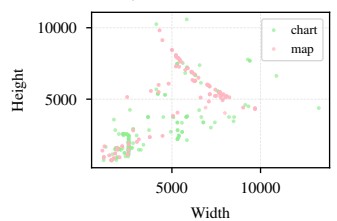

Figure 4: Distribution of layout numbers in O3-BENCH.     Figure 5: Resolution distribution.

We summarize the benchmark statistics from following three aspects. **(1) Distribution of layouts.** The benchmark features 8.7 layouts and 2.4 target layouts for each sample on average, indicating high information density and the need for multi-step reasoning. And the layout distribution by category is displayed in Figure 4. It can be seen that chart images typically exhibit a larger set of total layouts, whereas map images require more target layouts for reasoning. **(2) Distribution of resolution.** We collect high-resolution imagery in O3-BENCH. As shown in Figure 5, most images have side lengths in the 2K–5K range, while some images reach up to ∼10K pixels on the longer side, yielding high information density. On average, image height and width are 3,967 and 4,602 pixels, respectively. **(3) Distribution of options.** We randomly shuffle options A–E, ensuring an approximately uniform distribution of correct-answer positions. And a small portion (7.2%) of samples use option F (*No Right Choice*) as correct answer, which compels models to aggregate evidence across the entire image and determine that none of the other options is valid.

## B.2 DETAILS OF MACHINE PRE-ANNOTATION

(1) *Layout detection*. We first divide the high-resolution images into several structured layouts (*e.g.*, tables, charts, legends). We use PP-DocLayout_plus-L (Cui et al., 2025) to detect layout bounding boxes in the image and construct the set $\mathcal{L} = \{l_i\}_{i=1}^m$, where $l_i \in \mathbb{R}^4$ denotes layout coordinates. For chart images, we directly use the detector outputs. For map images, we review the predictions, correct erroneous regions, and supplement missing areas via manual annotation.

(2) *Information extraction*. For each detected layout $l_i$ in image $I$, we obtain the cropped image $I_{l_i}$ according to its coordinates. We then prompt Qwen2.5-VL-32B (Bai et al., 2025b) to produce a detailed caption $c_i$ and extract OCR text $o_i$ for $I_{l_i}$, thereby forming the layout triplet $\tau_i = (I_{l_i}, c_i, o_i)$. And then we aggregate all region triplets into $\mathcal{T} = \{\tau_i\}_{i=1}^m$. In addition, we obtain global context by generating a caption and OCR text for the full image, denoted as $\mathcal{G} = (c_g, o_g)$. The prompts we use are provided in Appendix E.1.

(3) *Automated question synthesis*. We provide the layout set $\mathcal{T}$ and the global context $\mathcal{G}$ to GPT-5 (OpenAI, 2025a) and explicitly prompt it to generate five questions that compose evidence from multiple regions. For each question, GPT-5 must produce six options (A–F) with exactly one correct answer, and option F is reserved for *No Right Choice*. It also need to supply a step-by-step explanation that interprets the reasoning chain. It is noted that we do not provide the full image to GPT-5, which compels the model to focus on region-level details and encourages multi-hop composition. The prompts we use are provided in Appendix E.2.

## B.3 MORE EXPERIMENTAL RESULTS FOR O3-BENCH

We present additional results for O3-BENCH in this section. Because our annotation pipeline supplies the target layouts most relevant to each question, we can provide these region crops alongside the original full image at test time. We evaluate GPT-5-Mini (OpenAI, 2025a) and Qwen2.5-VL-7B (Bai et al., 2025b), with results shown in Table 6. Both models exhibit significant performance gains when given the additional target layouts, underscoring the need for models to actively locate task-critical regions and perform interleaved visual reasoning.

Table 6: Ablation on target layouts in O3-BENCH.

| Model | O3-BENCH | | |
|---|---|---|---|
| | chart | map | overall |
| GPT-5-mini | 34.4 | 43.2 | 39.0 |
| + target layouts | 74.2 | 61.5 | 67.5 |
| Qwen2.5-VL-7B | 30.9 | 24.4 | 27.4 |
| + target layouts | 39.3 | 31.9 | 35.4 |

## B.4 VISUALIZATION OF O3-BENCH

In Figures 6–11, we present six representative visualizations (four map items and two chart items), showing how O3-BENCH couples high-resolution perception with multi-step reasoning. Each annotation includes: (i) the multiple-choice question and answer; (ii) highlighted target layouts that mark the regions consulted along the solution path; and (iii) a concise, ordered explanation that composes the evidence, which allows readers to verify the answer quickly.

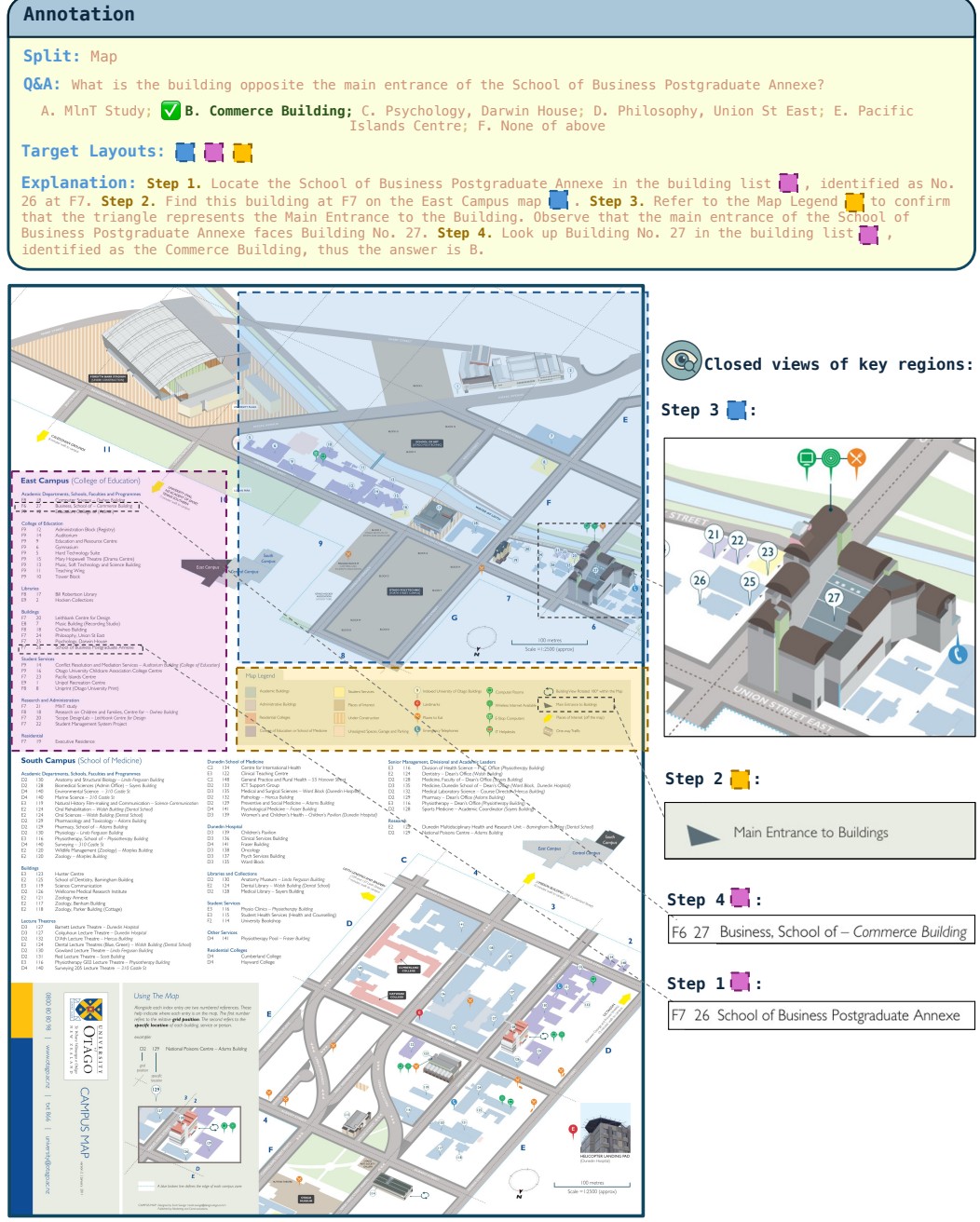

Figure 6: Example from O3-BENCH (Map-1). Each annotation comprises a six-choice QA and a brief explanation with highlighted target layouts for quick verification; additionally, we also provide step-wise close-ups (outside the annotation) to reveal the evidence chain in large images where fine details may be hard to see.

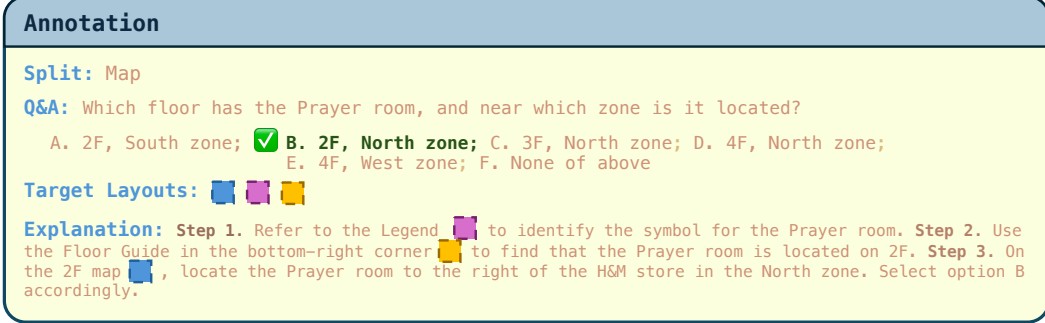

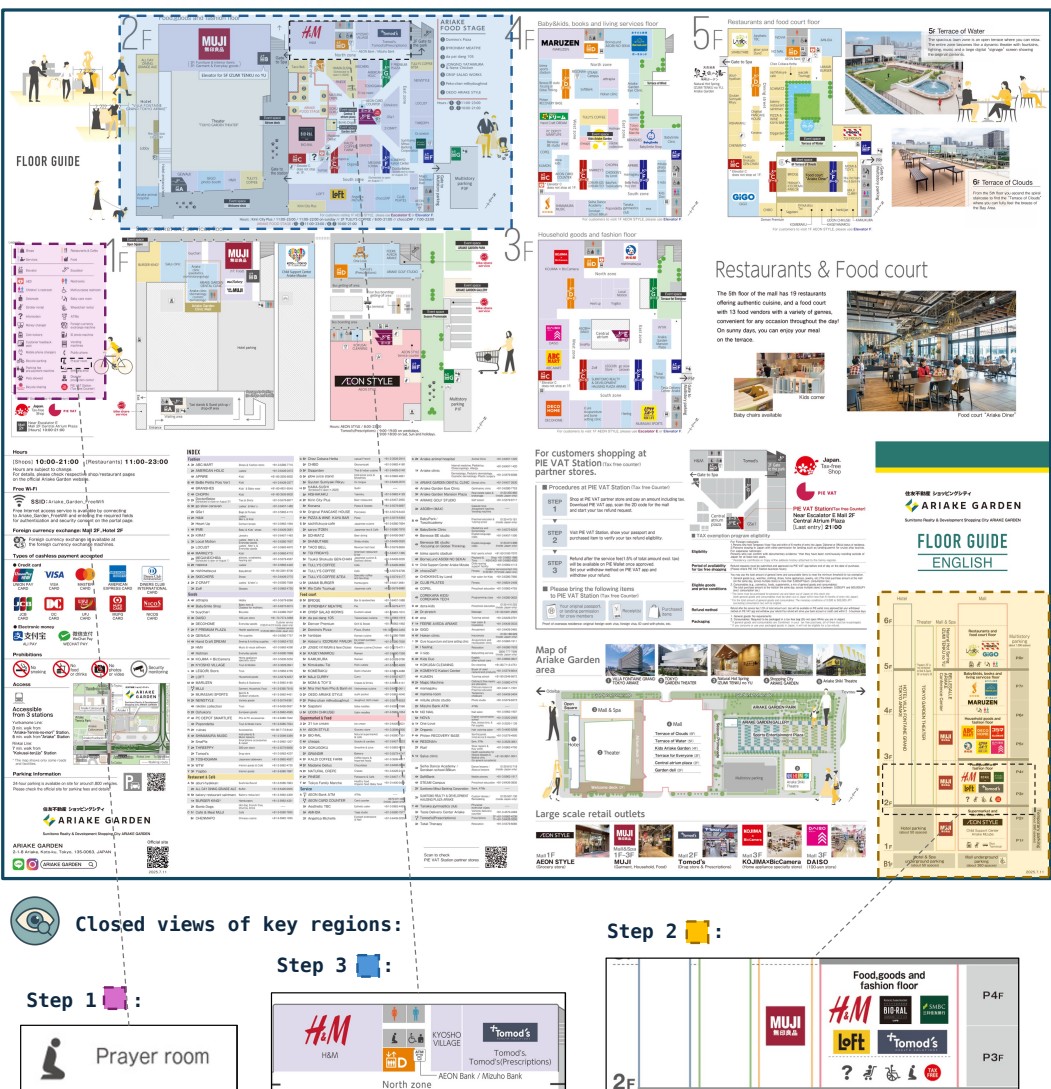

Figure 7: Example from O3-BENCH (Map-2). Each annotation comprises a six-choice QA and a brief explanation with highlighted target layouts for quick verification; additionally, we also provide step-wise close-ups (outside the annotation) to reveal the evidence chain in large images where fine details may be hard to see.

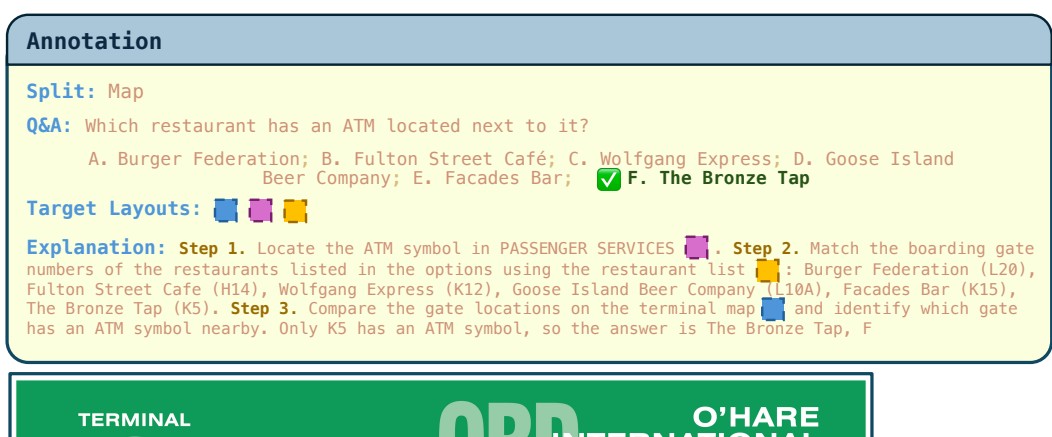

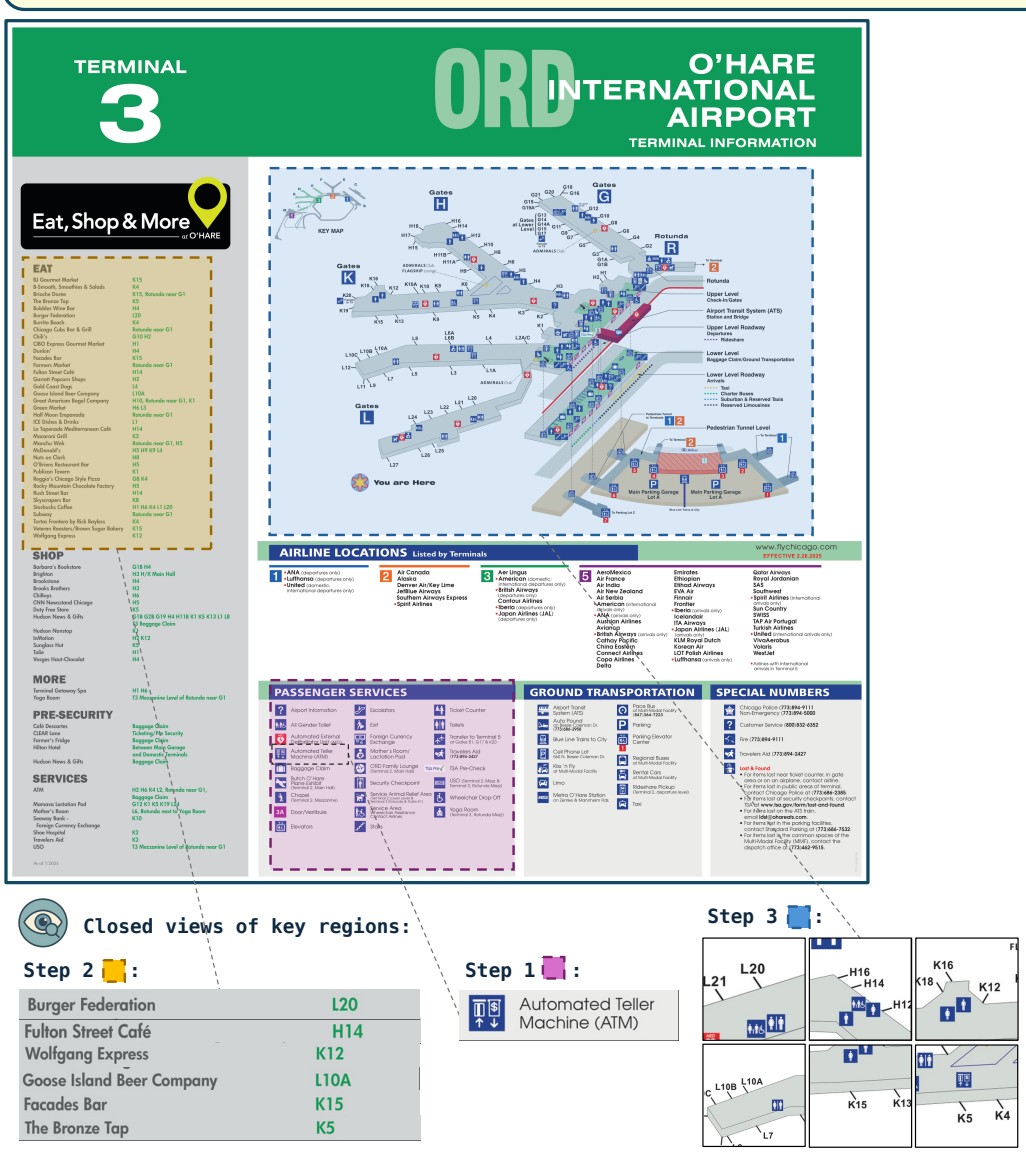

Figure 8: Example from O3-BENCH (Map-3). Each annotation comprises a six-choice QA and a brief explanation with highlighted target layouts for quick verification; additionally, we also provide step-wise close-ups (outside the annotation) to reveal the evidence chain in large images where fine details may be hard to see.

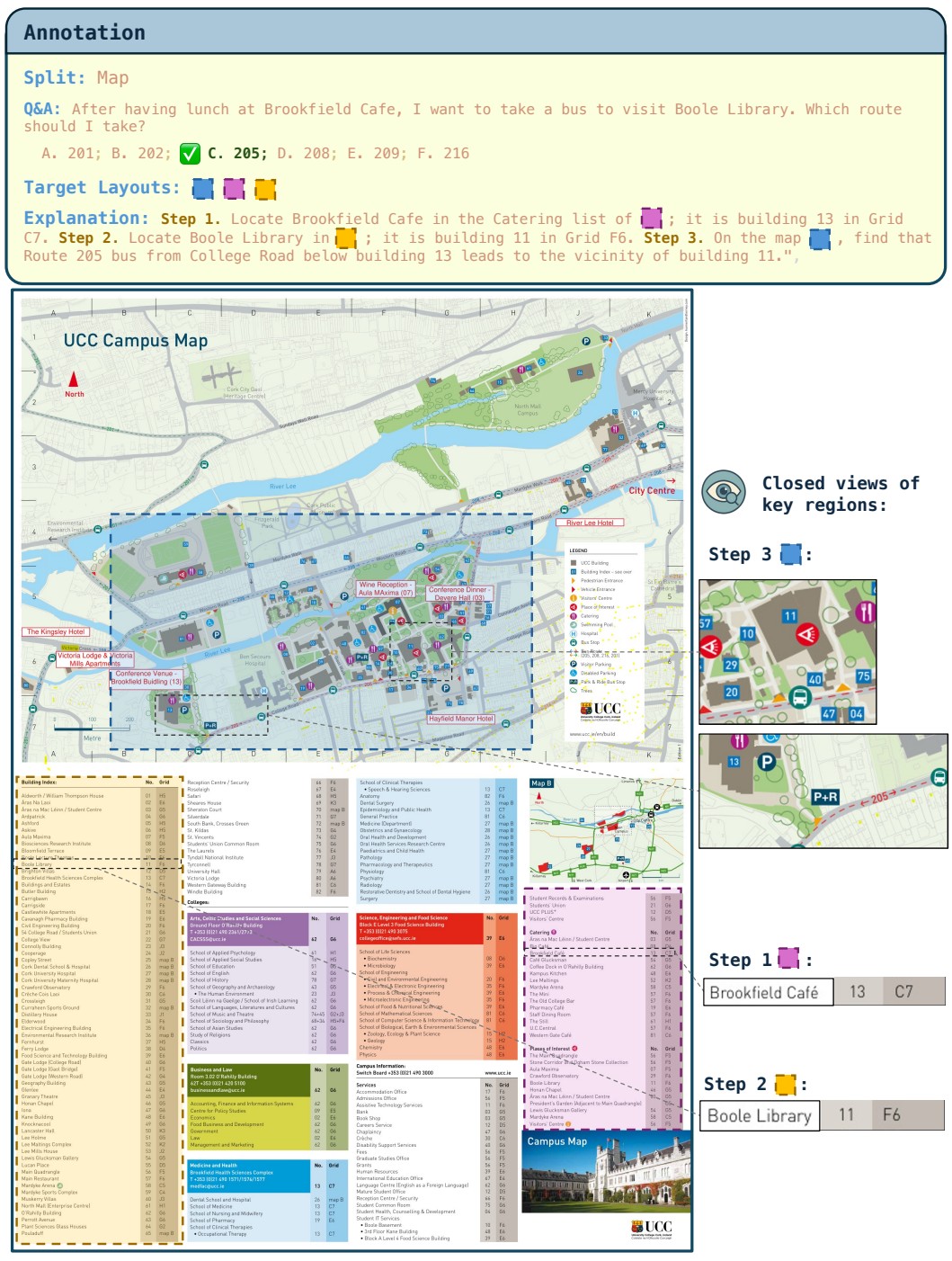

Figure 9: Example from O3-BENCH (Map-4). Each annotation comprises a six-choice QA and a brief explanation with highlighted target layouts for quick verification; additionally, we also provide step-wise close-ups (outside the annotation) to reveal the evidence chain in large images where fine details may be hard to see.

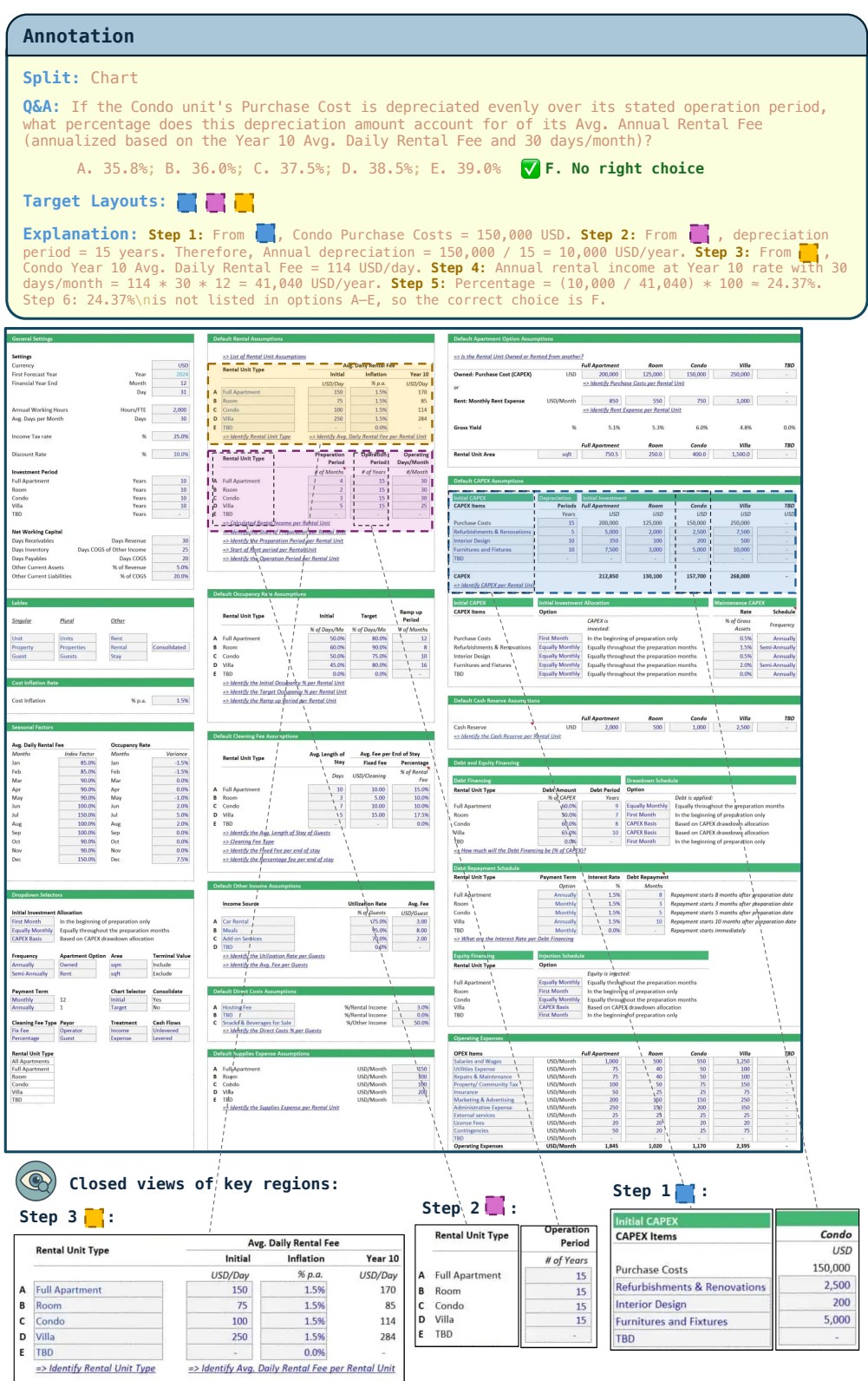

Figure 10: Example from O3-BENCH (Chart-1). Each annotation comprises a six-choice QA and a brief explanation with highlighted target layouts for quick verification; additionally, we also provide step-wise close-ups (outside the annotation) to reveal the evidence chain in large images where fine details may be hard to see.

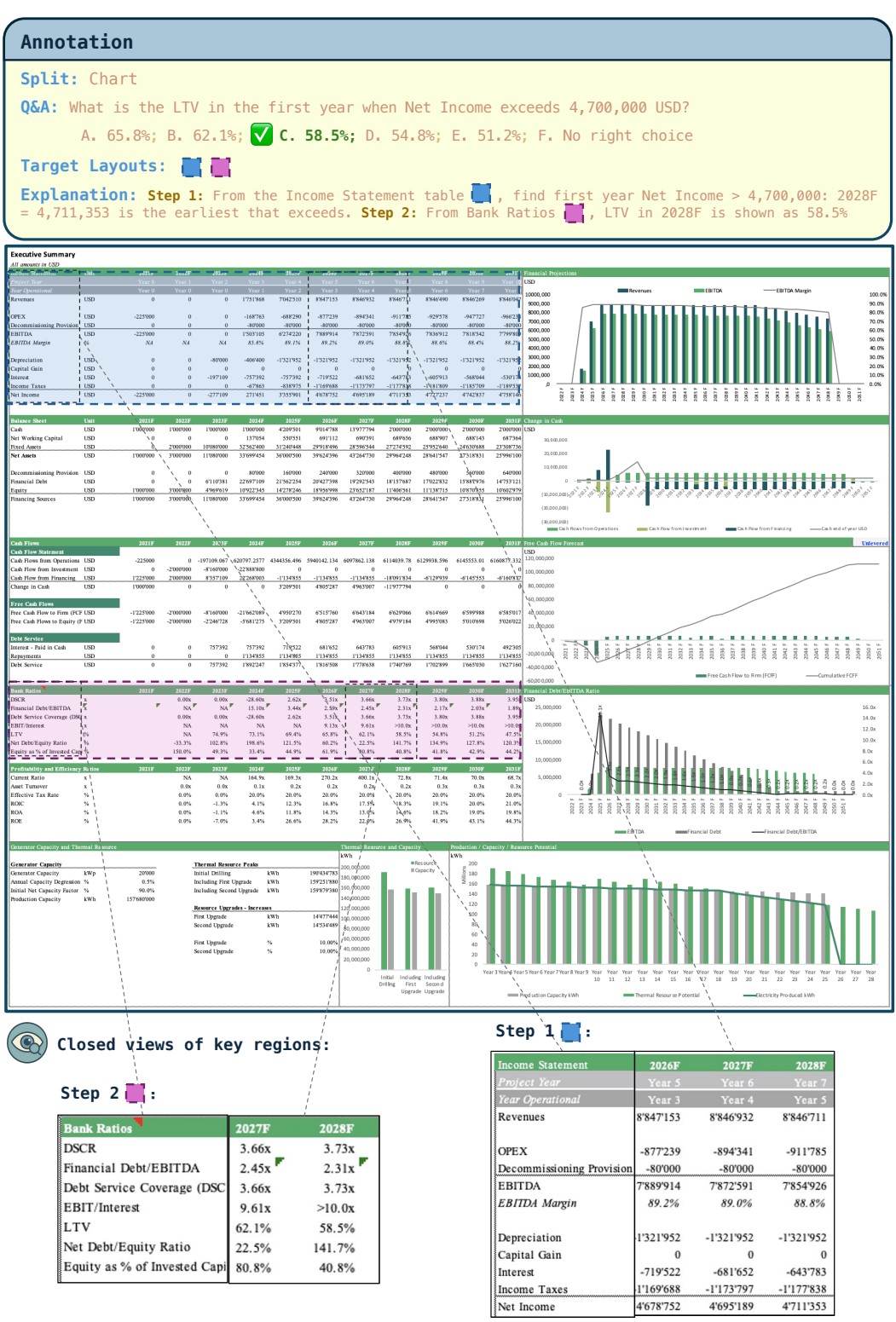

Figure 11: Example from O3-BENCH (Chart-2). Each annotation comprises a six-choice QA and a brief explanation with highlighted target layouts for quick verification; additionally, we also provide step-wise close-ups (outside the annotation) to reveal the evidence chain in large images where fine details may be hard to see.

### B.5    COMPARISON WITH MME-REALWORLD (CHART)

The chart images of O3-BENCH are mostly selected from the "Diagram & Table" subset of MME-RealWorld. The original questions from MME-RealWorld are relatively simple, usually focusing on a single value in a chart, which do not require any kind of multi-hop reasoning. For example, the original question of MME-RealWorld for the chart we show in Figure 10 is:

```
What is the cost inflation rate in the General Settings section of the
General Assumptions table?
```

In comparison, the new question in O3-BENCH is:

```
If the Condo unit's Purchase Cost is depreciated evenly over its stated
operation period, what percentage does this depreciation amount account
for of its Avg. Annual Rental Fee (annualized based on the Year 10 Avg.
Daily Rental Fee and 30 days/month)?
```

As illustrated in Figure 10, answering this question requires piecing together detailed information from *three* different tables through multi-step reasoning and arithmetic.

Overall, the questions of O3-BENCH (chart) are much more challenging than the MME-RealWorld counterparts. This can also be seen from the following statistics:

- The average accuracy of GPT-5-mini on O3-BENCH (chart) is about 38.2%, whereas on MME-RealWorld (chart), the accuracy is about 82.4%.

- The average number of vSearch steps of InSight-o3 on O3-BENCH (chart) is about 3.1, whereas on MME-RealWorld (chart), the number is about 1.1.

- The average multi-turn response length (including reasoning tokens) of GPT-5-mini vReasoner on O3-BENCH (chart) is about 1942.3 characters, whereas on MME-RealWorld (chart), the average response length is about 730.0 characters.

### B.6    FULL BENCHMARK RESULTS

Table 7 shows our full benchmark results of frontier multimodal models/systems on O3-BENCH. For OpenAI models/systems, oversize input images are resized to $1280{\times}1280$px (this is roughly the maximum supported size per OpenAI API, as mentioned in the main paper) and image detail is set to `high`. For other models/systems, oversize input images are resized to $3500{\times}3500$px (this translates to about 16K tokens/image for Qwen2.5-VL). All models/systems are given a 16K tokens/response budget (including reasoning tokens), which should be more than enough to solve the problems in O3-BENCH. Incomplete responses beyond this budget are considered wrong without checking. Open models are evaluated via self-hosted vLLM instances. Proprietary models are evaluated via official APIs. Our evaluation code can be found at https://github.com/m-Just/InSight-o3.

## C    TRAINING DATA CONSTRUCTION DETAILS

### C.1    IN-LOOP RL DATA

Our collage sources come from the training split of Visual CoT (Shao et al., 2024a) and V* (Wu & Xie, 2024). We first filter both datasets by target bounding box size, retaining items with area(bbox)/area(image) $< 0.04$. From Visual CoT, we keep all Chart/OCR-centric subsets (`dude`, `cub`, `textvqa`, `docvqa`, `infographicsvqa`, `sroie`, `vsr`, `textcap`), and treat the natural-image subsets (`flickr30k`, `gqa`, `openimages`, `v7w`) together with V* as a separate stream due to lower QA reliability (*e.g.*, weaker question–image alignment and non-unique answers). To ensure stable RL rewards, we filter this stream with an MLLM check using Qwen2.5-VL-7B and GPT-5-nano under a deterministic prompt. An item is retained only if both models return correct answer; otherwise it is discarded, including ambiguous or poorly aligned cases. After this pipeline, we retain $\sim$100K items as the source pool $\mathcal{D}$ for collage synthesis.

Table 7: Benchmark of frontier multimodal models/systems on O3-BENCH. Default model/system settings are used unless stated otherwise. All results are averaged over 3 random trials.

| | Chart | Map | Overall |
|---|---|---|---|
| LLaVA-OneVision-7B (Li et al., 2024) | $21.1_{\pm3.2}$ | $19.4_{\pm4.3}$ | $20.2_{\pm3.7}$ |
| InternVL3.5-8B (Wang et al., 2025e) | $26.2_{\pm2.5}$ | $22.7_{\pm0.7}$ | $24.3_{\pm1.1}$ |
| InternVL3.5-30B-A3B (Wang et al., 2025e) | $24.5_{\pm3.5}$ | $21.2_{\pm1.7}$ | $22.8_{\pm2.5}$ |
| GLM-4.6V (Hong et al., 2025) | $51.5_{\pm2.2}$ | $38.5_{\pm2.9}$ | $44.6_{\pm2.4}$ |
| Qwen2.5-VL-7B-Instruct (Bai et al., 2025b) | $30.9_{\pm1.8}$ | $24.4_{\pm1.1}$ | $27.4_{\pm0.3}$ |
| Qwen2.5-VL-32B-Instruct (Bai et al., 2025b) | $35.4_{\pm1.0}$ | $33.5_{\pm1.2}$ | $34.4_{\pm1.0}$ |
| Qwen3-VL-8B-Instruct (Bai et al., 2025a) | $54.4_{\pm0.3}$ | $33.9_{\pm4.3}$ | $43.6_{\pm0.4}$ |
| Qwen3-VL-8B-Thinking (Bai et al., 2025a) | $49.1_{\pm2.2}$ | $33.0_{\pm0.9}$ | $40.6_{\pm0.7}$ |
| Qwen3-VL-30B-A3B-Instruct (Bai et al., 2025a) | $49.3_{\pm1.4}$ | $32.1_{\pm1.9}$ | $40.2_{\pm0.4}$ |
| Qwen3-VL-30B-A3B-Thinking (Bai et al., 2025a) | $51.1_{\pm1.5}$ | $36.8_{\pm1.2}$ | $43.6_{\pm1.3}$ |
| Qwen3-VL-32B-Instruct (Bai et al., 2025a) | $73.7_{\pm1.3}$ | $48.5_{\pm2.1}$ | $60.4_{\pm1.7}$ |
| Qwen3-VL-32B-Thinking (Bai et al., 2025a) | $52.4_{\pm3.1}$ | $40.5_{\pm1.4}$ | $46.1_{\pm1.3}$ |
| Qwen3-VL-235B-A22B-Instruct (Bai et al., 2025a) | $73.4_{\pm1.9}$ | $53.8_{\pm2.0}$ | $63.1_{\pm0.8}$ |
| Qwen3-VL-235B-A22B-Thinking (Bai et al., 2025a) | $57.3_{\pm1.2}$ | $47.8_{\pm2.0}$ | $52.3_{\pm0.8}$ |
| GPT-4o (OpenAI, 2024) | $22.1_{\pm0.9}$ | $33.3_{\pm1.0}$ | $28.0_{\pm0.8}$ |
| GPT-5-nano (OpenAI, 2025a) | $19.2_{\pm2.3}$ | $33.3_{\pm3.9}$ | $26.5_{\pm3.1}$ |
| GPT-5-mini (OpenAI, 2025a) | $34.4_{\pm3.5}$ | $43.2_{\pm2.0}$ | $39.0_{\pm0.6}$ |
| GPT-5 (OpenAI, 2025a) | $30.9_{\pm0.8}$ | $52.6_{\pm0.7}$ | $42.3_{\pm0.0}$ |
| GPT-5.2 (OpenAI, 2025b) | $31.9_{\pm2.3}$ | $39.0_{\pm2.7}$ | $35.7_{\pm2.3}$ |
| OpenAI o3 (OpenAI, 2025c) | $27.8_{\pm1.3}$ | $52.4_{\pm2.0}$ | $40.8_{\pm0.9}$ |
| Gemini-2.5-Flash[#](Comanici et al., 2025) | $46.6_{\pm1.3}$ | $52.6_{\pm3.0}$ | $49.8_{\pm1.4}$ |
| Gemini-2.5-Flash (Comanici et al., 2025) | $61.8_{\pm1.2}$ | $59.2_{\pm1.8}$ | $60.4_{\pm0.5}$ |
| Gemini-2.5-Pro (Comanici et al., 2025) | $67.3_{\pm2.5}$ | $63.7_{\pm2.5}$ | $65.4_{\pm2.5}$ |
| Gemini-3-Flash (Google, 2025) | $68.1_{\pm2.6}$ | $69.0_{\pm3.4}$ | $68.6_{\pm1.6}$ |
| Gemini-3-Pro-Preview (Google, 2025) | $67.7_{\pm2.0}$ | $69.6_{\pm3.6}$ | $68.7_{\pm2.7}$ |
| doubao-seed-1-6-250615 (Bytedance, 2025) | $55.4_{\pm1.5}$ | $48.5_{\pm4.4}$ | $51.8_{\pm2.7}$ |
| **INSIGHT-O3 (w/ GPT-4o)** | $34.4_{\pm0.7}$ | $38.3_{\pm0.8}$ | $36.4_{\pm0.2}$ |
| **INSIGHT-O3 (w/ GPT-5-nano)** | $35.3_{\pm2.2}$ | $34.1_{\pm1.6}$ | $34.6_{\pm1.9}$ |
| **INSIGHT-O3 (w/ GPT-5-mini)** | $67.3_{\pm1.4}$ | $56.4_{\pm2.1}$ | $61.5_{\pm0.4}$ |
| **INSIGHT-O3 (w/ Gemini-2.5-Flash)** | $75.6_{\pm2.0}$ | $64.4_{\pm3.0}$ | $69.7_{\pm0.7}$ |

[#] Image-size constraint set to $1280\times1280$px, roughly the maximum supported size for OpenAI models/systems via API.

Given the filtered source pool $\mathcal{D}$, we synthesize collage-style training images around one primary target (the image the model should attend to) and auxiliary fills (other images used to occupy remaining space and control background complexity). The full procedure is presented in Algorithm 1. Specifically, we sample and grid-quantize a canvas, then determine a feasible target scale by intersecting global bounds with a bbox-to-canvas cap and a minimum short-edge constraint after light aspect jitter (Steps 1–2). We plan & place the target using a fit-then-shrink heuristic with a single enlarge-canvas fallback (Steps 3–4). Remaining area is panelized into grid-aligned regions under simple aspect/size guards (Step 5). Panels are then filled (largest-first) by sampling images $\tilde{t} \in \mathcal{D}$ using usage-aware weights (favoring less-frequently used candidates) that also roughly match panel aspect; when needed, we apply a light center crop and bounded scaling (Step 6). If packing remains incomplete after a brief extra fill pass, we resample from the canvas; otherwise we finalize the collage (Steps 7–8). To avoid ambiguity when querying the target image, we annotate each collage tile with an ID and include this ID in the question as a reference. Figure 12 shows representative visualizations of synthesized collages.

The canvas is sized so that the target box occupies only a tiny fraction of the canvas area, enforcing $\mathrm{area(bbox)}/\mathrm{area(canvas)} < 0.0002$. We filter out items that vReasoner can already solve without calling vSearcher using a pass@3 check (three attempts; any success leads to removal).

---

**Algorithm 1** Target-and-Fill Collage Synthesis (High-level)

---

**Require:** Metadata table $\mathcal{D}$ (image path, $W_{\text{src}}$, $H_{\text{src}}$, object bbox), **target image** $t^{\star} \in \mathcal{D}$, grid $G$, min short edge $M$, canvas area/aspect ranges $[A_{\min}, A_{\max}]$ and $[a_{\min}, a_{\max}]$, target scale bounds $[\lambda_{\min}, \lambda_{\max}]$, target aspect jitter $\tau_{\text{tgt}}$, fill jitter $\tau_{\text{fill}}$, fill scales $[\lambda_{\min}^{\text{fill}}, \lambda_{\max}^{\text{fill}}]$, panel aspect range $[\text{AR}_{\min}^{\text{panel}}, \text{AR}_{\max}^{\text{panel}}]$, max effective source area $S_{\max}^{\text{eff}}$, **bbox coverage cap** $\rho_{\text{cap}} = 2 \times 10^{-4}$, placement retries $R$, max attempts $T$

**Ensure:** Canvas $\mathcal{C}$ and placements $\mathcal{P} = \{\text{target}, \text{fills}\}$

1: **Precompute target meta.** From $t^{\star}$, read $W_{\text{src}}$, $H_{\text{src}}$, bbox ratio $\rho_{\text{src}}$; set $S_{\star} \leftarrow (W_{\text{src}} H_{\text{src}}) \cdot \min\!\big(1, \sqrt{S_{\max}^{\text{eff}}/(W_{\text{src}} H_{\text{src}})}\big)^2$, and $r_{\text{src}} \leftarrow W_{\text{src}}/H_{\text{src}}$.

2: **for** attempt $= 1$ to $T$ **do**            $\triangleright$ rejection loop

3:      **Step 1 — Sample canvas.** Draw $A_{\text{canvas}} \sim [A_{\min}, A_{\max}]$, $a \sim [a_{\min}, a_{\max}]$; snap to grid to obtain $(W, H)$.

4:      **Step 2 — Compute feasible target scale interval.**

       Bounds: $\lambda \in [\lambda_{\min}, \lambda_{\max}]$; occupancy: $\lambda S_{\star} \leq A_{\text{canvas}}$; bbox: $\dfrac{\rho_{\text{src}} \lambda S_{\star}}{A_{\text{canvas}}} \leq \rho_{\text{cap}}$.

       Choose $r_{\star}$ by log-jittering $r_{\text{src}}$ within $\pm\tau_{\text{tgt}}$ and raise the lower bound on $\lambda$ so that $\min(\sqrt{\lambda S_{\star} r_{\star}}, \sqrt{\lambda S_{\star}/r_{\star}}) \geq M$.

       Let $I_{\lambda}$ be the intersection of the above constraints; if $I_{\lambda} = \varnothing$, optionally enlarge $A_{\text{canvas}}$ once and recompute; if still empty, **continue**       $\triangleright$ reject $\to$ restart at Step 1

5:      **Step 3 — Plan target box.** Pick $\lambda \in I_{\lambda}$ (*e.g.*, midpoint); set $w_{\star} = \sqrt{(\lambda S_{\star}) r_{\star}}$, $h_{\star} = \sqrt{(\lambda S_{\star})/r_{\star}}$; snap $(w_{\star}, h_{\star})$ to multiples of $G$.

6:      **Step 4 — Place target.** Best-fit on the free list (grid-aligned). If no fit, iteratively shrink $(w_{\star}, h_{\star})$ and update $\lambda = w_{\star} h_{\star}/S_{\star}$, keeping feasibility in $I_{\lambda}$, up to $R$ retries; if still not placed, optionally enlarge canvas once and re-plan; if it fails, **continue**     $\triangleright$ reject $\to$ restart at Step 1

7:      **Step 5 — Normalize free space.** Recursively split free regions into grid-aligned panels subject to aspect $\text{AR}_P \in [\text{AR}_{\min}^{\text{panel}}, \text{AR}_{\max}^{\text{panel}}]$ and minimal size.

8:      **Step 6 — Fill panels.** For each panel (largest-first), sample a fill image $\tilde{t} \in \mathcal{D}$ within a $\pm\tau_{\text{fill}}$ aspect band around $\text{AR}_P$; if needed, center-crop $\tilde{t}$ to $\text{AR}_P$; scale with $\lambda^{\text{fill}} \in [\lambda_{\min}^{\text{fill}}, \lambda_{\max}^{\text{fill}}]$ and place.

9:      **Step 7 — Resample if needed.** If residual free space remains after one extra fill pass, **continue**                    $\triangleright$ reject $\to$ restart at Step 1

10:     **Step 8 — Finalize. return** $\mathcal{C}, \mathcal{P}$

11: **end for**

12: **return None**             $\triangleright$ no feasible collage after $T$ attempts

---

## C.2 OUT-OF-LOOP RL DATA

For PP-DocLayout_plus-L, we use the following configuration: `{"threshold": 0.01, "layout_nms": True, "layout_merge_bboxes_mode": "union"}`.

To construct meaningful visual search targets from the boxes produced by PP-DocLayout_plus-L, we start by dropping boxes of trivial layout classes such as `header` and `footer`, keeping only `text`, `image`, `table`, `chart`, `figure_title`, and `paragraph_title` boxes. We then drop boxes that are too large as they usually merge disparate things together. Boxes with area larger than a quarter of the whole image area are dropped, except for charts which are usually clean and we use a much higher threshold (0.8) for them. Next, we merge boxes that are very close to each other, measured by the effort required to enclose them in one box. The effort is computed as `1 - summed_area / min_enclosing_area`. We start with the boxes that require the least effort to merge, and stop until the required effort reaches a threshold. For `figure_title` and `paragraph_title` boxes, the threshold is 0.15, while for other boxes, it is 0.1. This helps merging small, auxiliary boxes such as figure titles and chart legends with their closest neighbors in the vicinity, avoiding truncating important context and preventing trivial search targets dominating

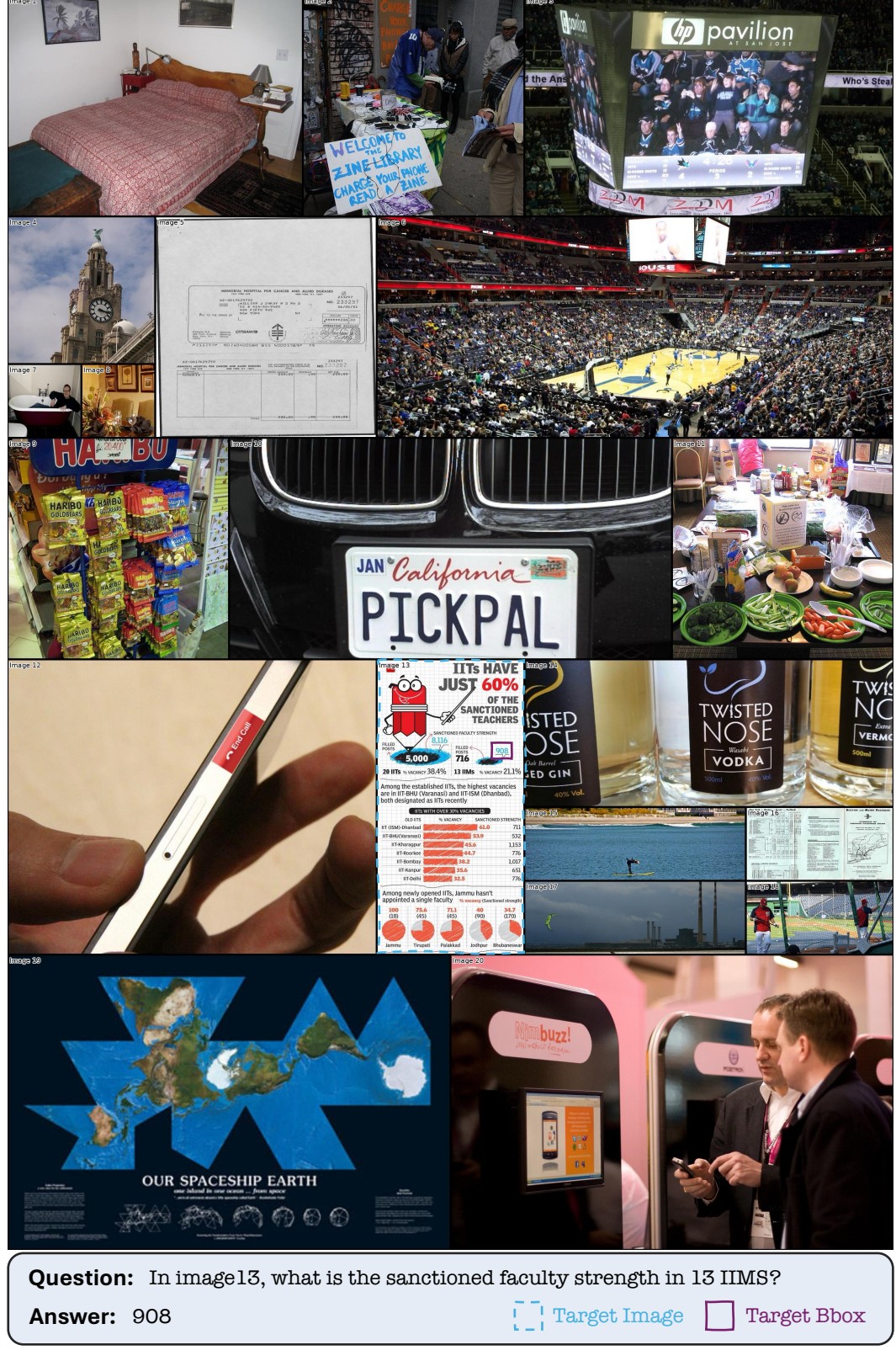

Figure 12: Example of synthesized collage for the in-loop RL. Multiple low-resolution images are stitched to raise visual density. The blue dashed box highlights the target tile; the magenta box marks the target bbox. Remaining tiles are distractors.

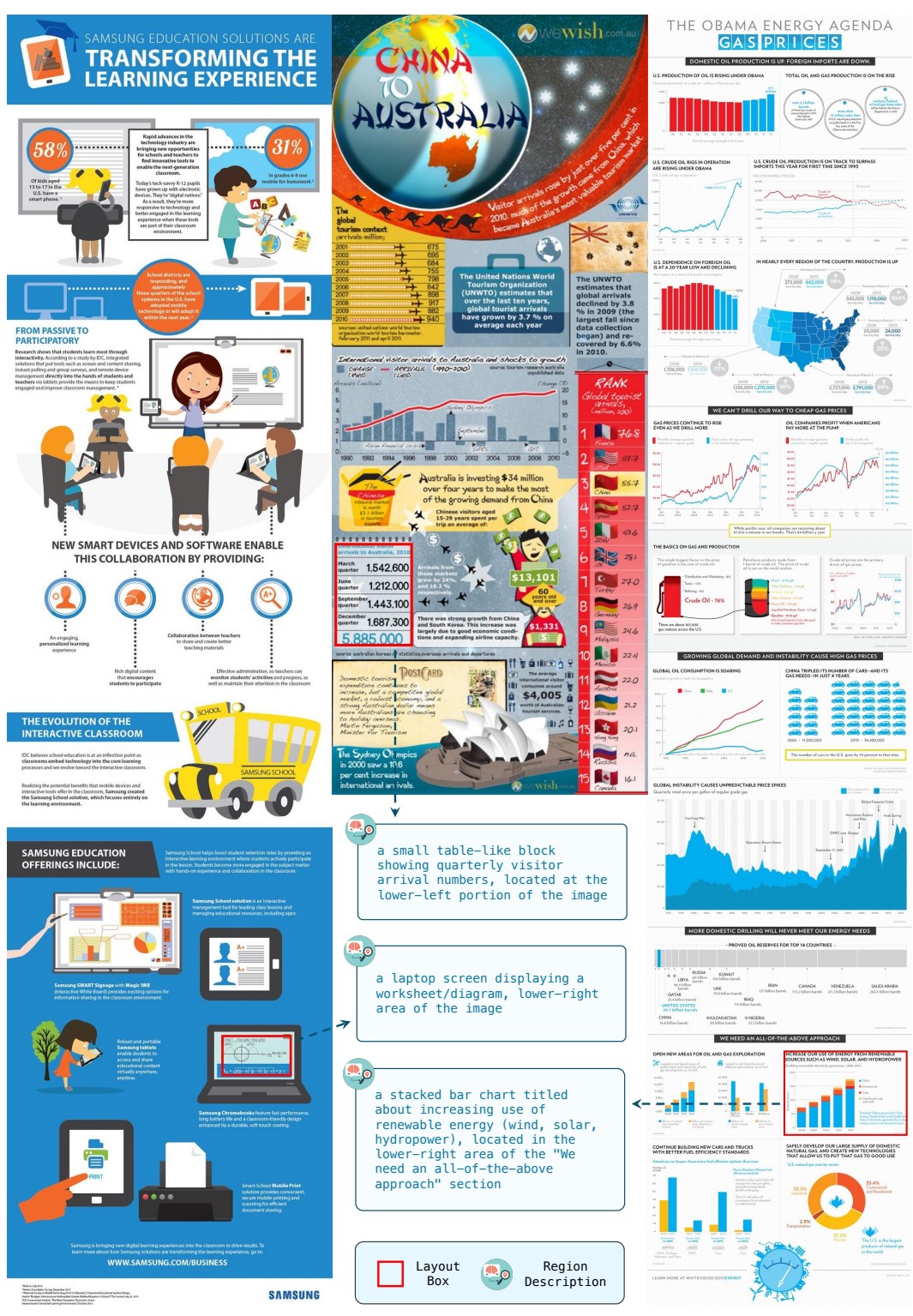

Figure 13: Examples of InfographicVQA images with pre-generated layout boxes and region descriptions for the out-of-loop RL.

the dataset. We skip a merge if the new box would be too large (box-to-image area ratio more than 0.25) or have an extreme aspect ratio (not within 1:5 and 5:1). We do not merge charts and tables.

After merging the boxes, we drop (1) charts/tables enclosing other charts/tables, (2) unmerged images that do not contain any text or titles, (3) unmerged titles, (4) unmerged texts, (5) boxes that are too small (box-to-image area ratio less than 0.001), and (6) boxes with extreme aspect ratios (not within 1:5 and 5:1). These boxes often contain little information (*e.g.*, icons, short texts). In the end, an image may still have multiple visual search targets; they are treated as separate data entries.

To generate region descriptions for the visual search targets obtained earlier, we first draw a red box around the target on the image, and then prompt GPT-5-nano as follows:

```
[SYSTEM]
You are a visual assistant. Your goal is to help the user to locate the region indicated by
 the red bounding box in an image.

When the user asks you to describe the region, you must follow the following rules:
- Keep it super simple and short as if you can't see clearly what is in the region.
- Don't mention any details, specific content, or small text in the region.
- Use concise, visually grounded targets (e.g., a chart, an object, a text block, a
distinct area).
- Optionally include approximate location (e.g., top-left of the image, bottom-right of the
 big chart, center column).
- Optionally include the title of the region (e.g., the table about XXX, the section titled
 XXX).
- Avoid non-visual or ordinal references (e.g., "the third largest bar", "the second row's
number").
- Don't mention the red bounding box.

Output format: region_description={...}

[USER]
Describe the region in the red bounding box.
```

In Figure 13, three examples of the final data are shown. Note that unlike collages, these examples are not stitched together; they are simply displayed side-by-side to save space.

## D    ADDITIONAL INFORMATION ON INSIGHT-O3

### D.1    INSIGHT-O3 IMPLEMENTATION DETAILS

The maximum image resolution of vSearcher is set to $\sim$3.2M pixels (4K tokens/image) during training, and $\sim$12.8M pixels (16K tokens/image) during evaluation. Oversize images are downsampled to meet the constraint. We allow both vReasoner and vSearcher to make at most 6 sub-agent/tool calls during both training and evaluation. Image crops returned by sub-agent/tool calls are obtained from original images, and then resized if they exceed the size limit. For vSearcher, we use a maximum response length (including results returned by sub-agent/tool calls) of 9K and 32K tokens (with sampling temperature 1 and 0) for training and evaluation, respectively. Other hyperparameters include: training batch size 24, rollout number 8, learning rate $10^{-6}$, KL loss coefficient 0.01, reward weights $\lambda_{\text{format}} = 0.2$, $\lambda_{\text{IoU}} = 0.8$, and IoU reward threshold $\alpha = 0.25$. The composition of in-loop/out-of-loop training data is 1:1. We train vSearcher fully on-policy for 150 steps. We freeze the vision tower and the adapter of Qwen2.5-VL-7B-Instruct during the whole training process. We use GPT-5-nano (OpenAI, 2025a) for evaluating answer correctness. Our code is based on verl (Sheng et al., 2024). The prompts we use can be found in Appendix F.

### D.2    MORE VISUALIZATIONS OF INSIGHT-O3 REASONING PROCESS

Figure 14 and 15 show examples of inference process of INSIGHT-O3 (GPT-5-mini as vReasoner). The vReasoner issues natural-language target descriptions; the vSearcher localizes evidence and returns them. Across a few rounds, the pair composes multi-step evidence and produces the final answer, demonstrating that INSIGHT-O3-VS plugs in cleanly and supports effective reasoning.

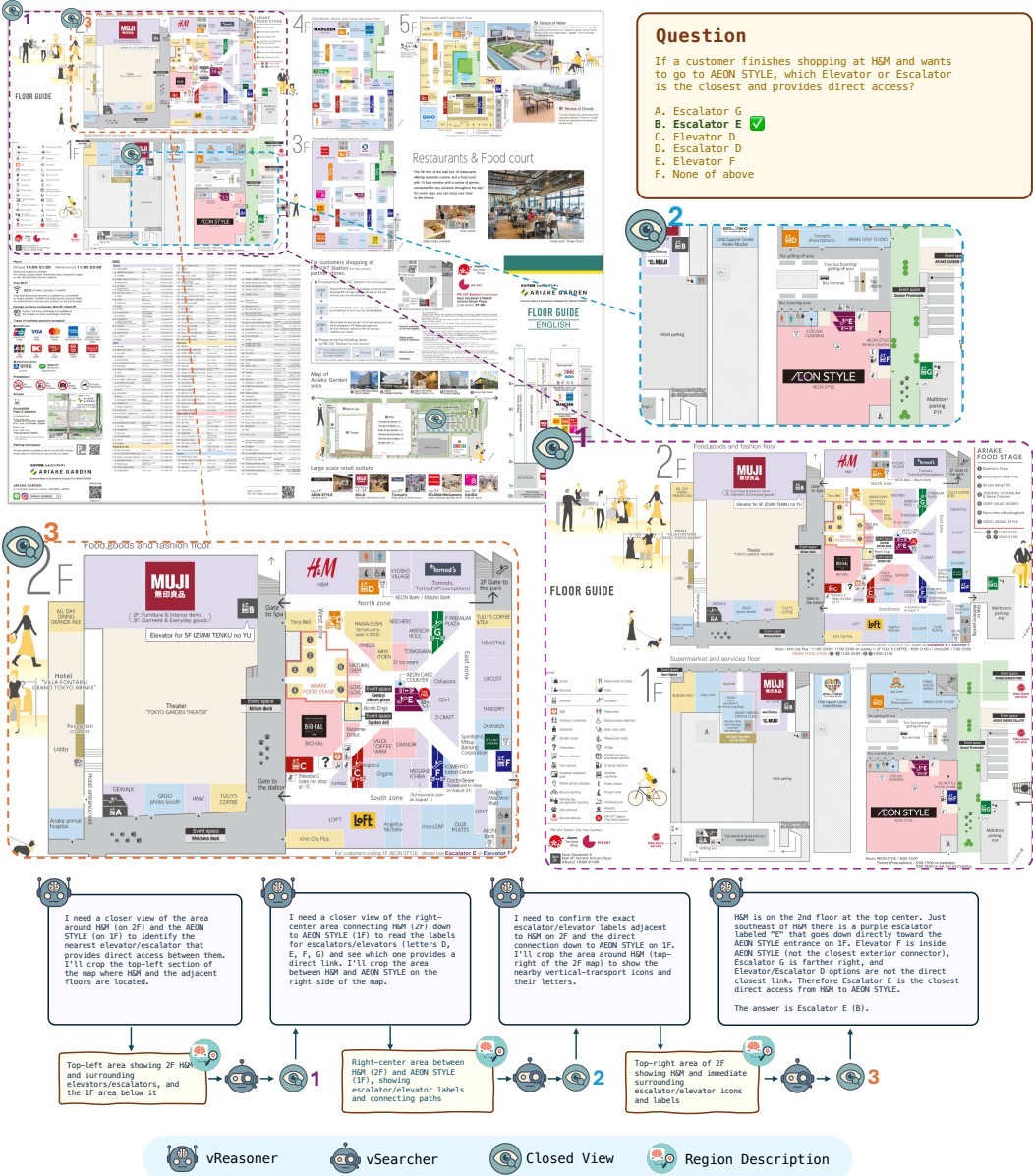

Figure 14: Qualitative example 1 with GPT-5-mini as vReasoner and INSIGHT-O3-VS as vSearcher. The reasoner requests venue-level cues (*e.g.*, legend/index lookups); the searcher returns localized regions and snippets, iterating to a correct answer.

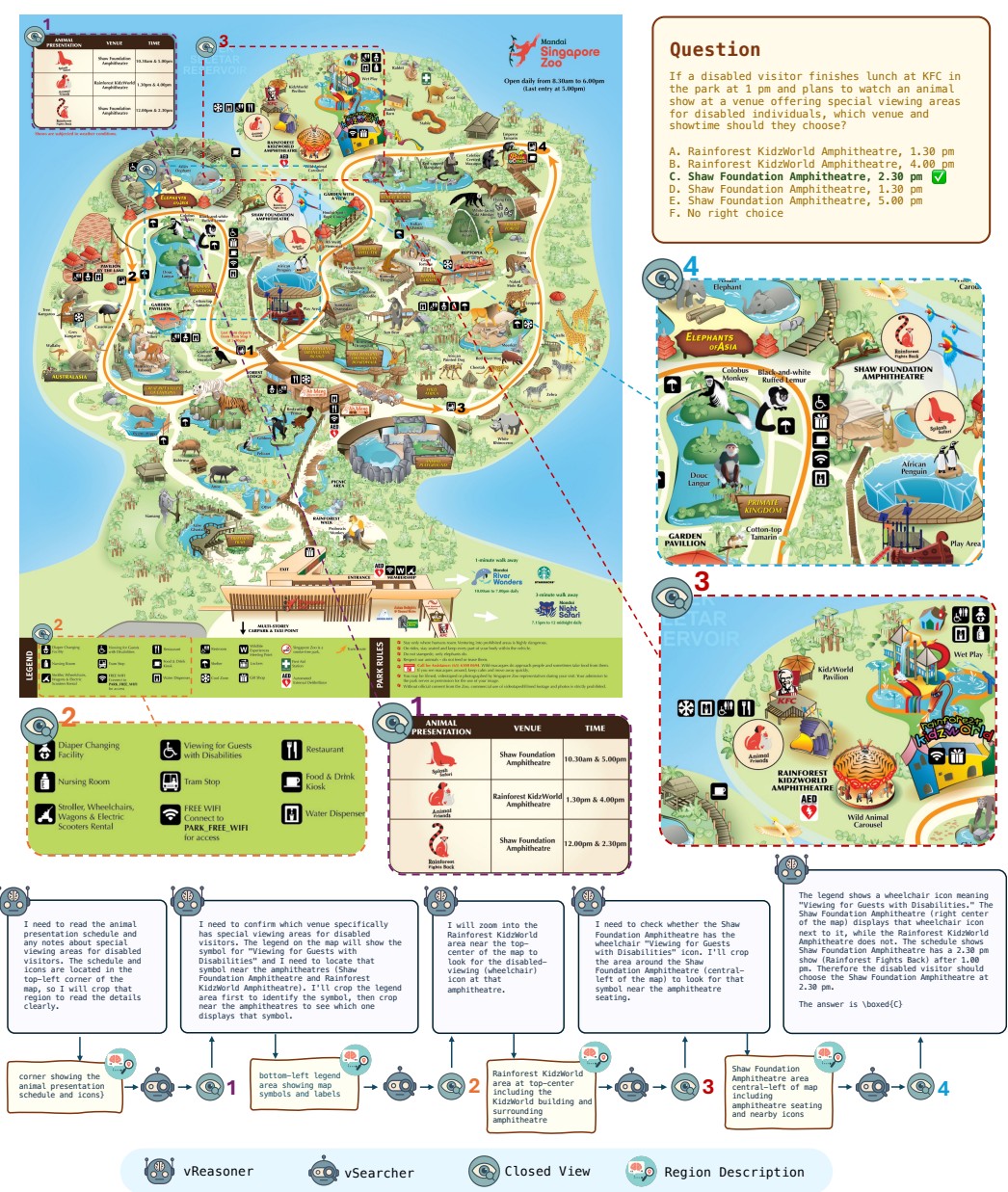

Figure 15: Qualitative example 2 with GPT-5-mini as vReasoner and INSIGHT-O3-VS as vSearcher. The reasoner requests venue-level cues (*e.g.*, legend/index lookups); the searcher returns localized regions and snippets, iterating to a correct answer.

### D.3 COMPARATIVE ANALYSIS BETWEEN INSIGHT-O3 AND BASELINES

In Figure 16–19, we compare the behavior of INSIGHT-O3 (GPT-5-mini + InSight-o3-vS) with two baselines: (1) GPT-5-mini and (2) GPT-5-mini + Qwen2.5-VL-7B. This comparative analysis is based on three examples of O3-BENCH. We rate each crop returned by vSearcher on three levels:

- High-quality crops tightly enclose the visual search targets and the relevant context.
- Medium-quality crops contain the visual search targets but include too much context.
- Low-quality crops miss or truncate the visual search targets or relevant context.

In the most basic setting where vReasoner does not have access to vSearcher, it often uses similar reasoning patterns as follows to reach its conclusion: "I first locate ... I see ... I then look for ... There is ... Therefore ..." (see the top parts of Figure 17-19). During this process, vReasoner often hallucinates and makes factual errors about what it sees, suggesting that it does not really see the relevant visual details clearly but still pretends so anyway.

With the vanilla Qwen2.5-VL-7B vSearcher, vReasoner is able to take closer looks at regions of interest and makes less factual errors (see the bottom part of Figure 17 and the middle part of Figure 18-19). However, the vanilla vSearcher is often unreliable, returning inaccurate/wrong crops to vReasoner or simply concluding that the target is not in the image, usually after a minimal amount of (sometimes none) reasoning. In such cases, vReasoner would eventually give up and resort to its own perception after multiple failed visual search attempts, leading to wrong final answers.

In comparison, our visual search agent, InSight-o3-vS, would usually first reason about the visual search request and then crop the candidate region to verify before returning it to vReasoner. For example, in Figure 16, InSight-o3-vS first reasons about what the bounding box should cover:

```
Based on the description, the right section of the map includes the
legend/index and the "Catering venues" list with numbered cafes. The
legend/index is located at right of the map, and the "Catering venues"
list is further down, under the "Catering venues" heading. The bounding
box should cover these areas.
```

Then, after viewing the cropped region, it concludes:

```
Based on the tool response, the right section of the map showing the
legend/index and the "Catering venues" list with numbered cafes is
already covered by the bounding box provided. Therefore, no further
zooming is necessary.
```

In some cases, InSight-o3-vS is able to correct an initial bad crop after reviewing the crop:

```
Based on the response, the zoomed-in area does not match the intended
area around grid B4. The bounding box needs to be adjusted to better
capture the car parks and nearby cafe icons and labels.
```

The returned crops are usually medium-to-high-quality crops as shown in Figure 16. From these cases, we can see that vSearcher mostly helps vReasoner by (i) precisely locating the regions requested by vReasoner and (ii) presenting them to vReasoner and directing its attention to those regions, thereby reducing hallucination and facilitating evidence-based reasoning.

### D.4 FAILURE CASES OF INSIGHT-O3

Figure 20-21 show typical failure cases of INSIGHT-O3. In the first three failure cases, vSearcher (InSight-o3-vS) provided the correct crops for the search targets but vReasoner (GPT-5-mini) answers incorrectly due to its own errors, e.g., ignoring visual evidence due to internal knowledge bias, and jumping to conclusion without examining key visual information. This suggests that existing frontier models, even the proprietary ones, are still not very good at thinking with images in complex scenarios. The last failure cases is due to vSearcher failing to understand the structure of the map and/or the relatively complicated region descriptions involving multiple visual cues. Finally, we note that, at least on O3-BENCH, most wrong answers are due to vReasoner, not vSearcher.

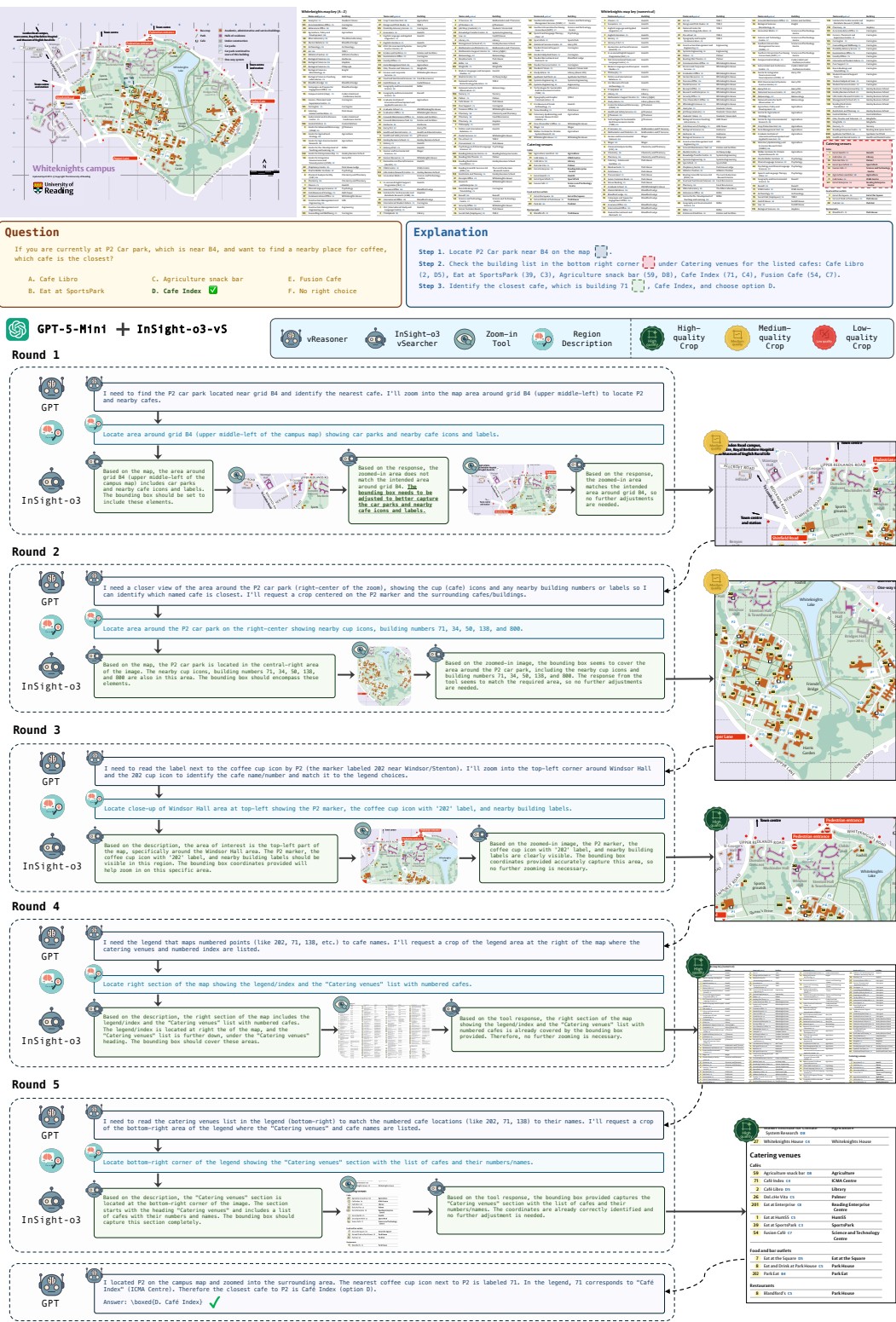

Figure 16: Example 1 (MAP case 1) of complete reasoning trace of our model (GPT-5-mini as vReasoner and INSIGHT-O3-VS as vSearcher). The vReasoner iteratively guides vSearcher through natural-language region descriptions. INSIGHT-O3-VS precisely retrieves high-quality image crops that match the described regions, effectively supporting vReasoner in locating the target area and producing the correct answer.

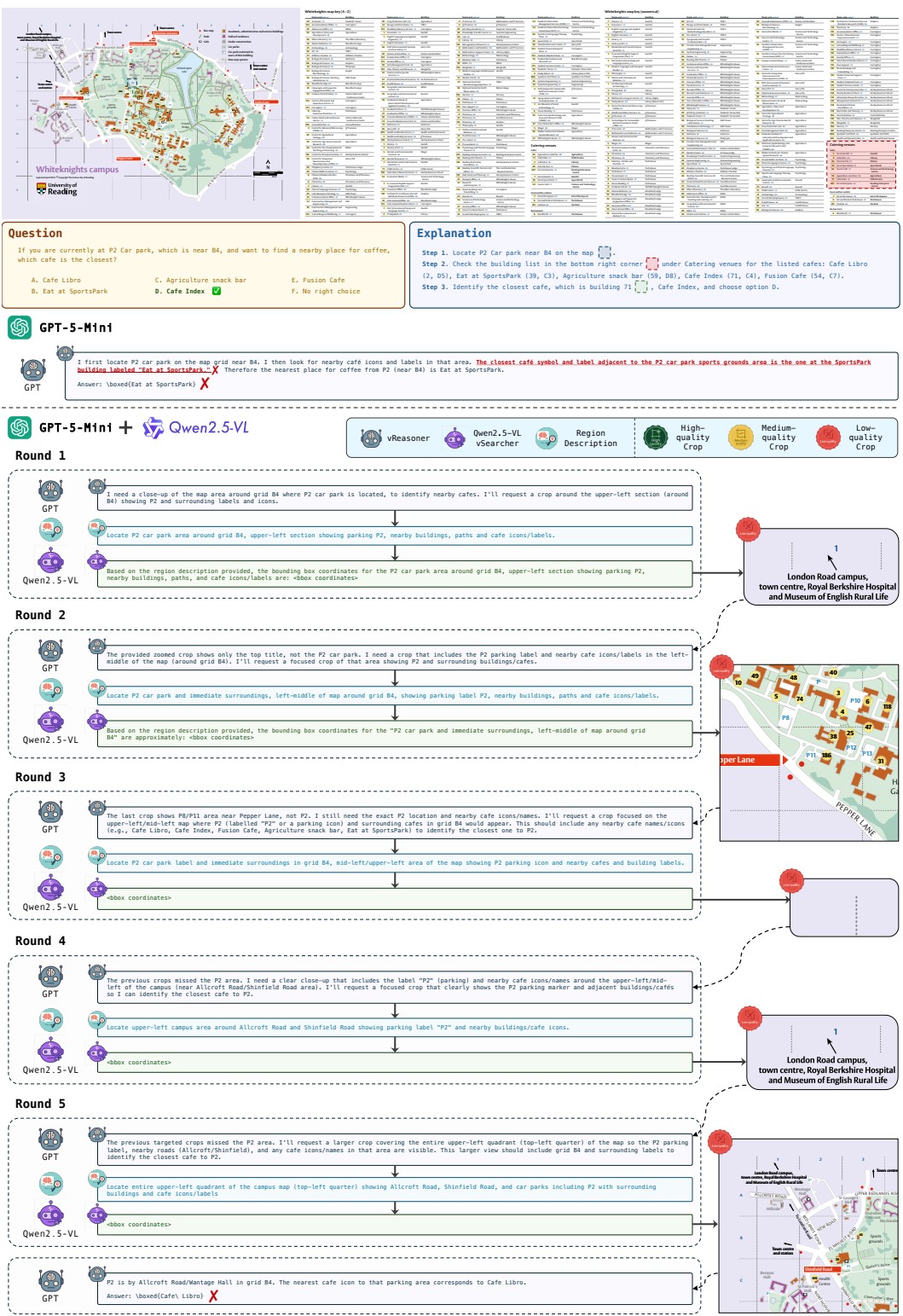

Figure 17: Example 1 (MAP case 1, continuation of Fig. 16): reasoning traces of GPT-5-mini and GPT-5-mini + Qwen2.5-VL-7B-Instruct. GPT-5-mini exhibits misperception and reasoning drift, while Qwen2.5-VL-7B-Instruct frequently fails to follow vReasoner's instructions, producing low-quality crops misaligned with the described regions. Consequently, both baselines yield incorrect answers.

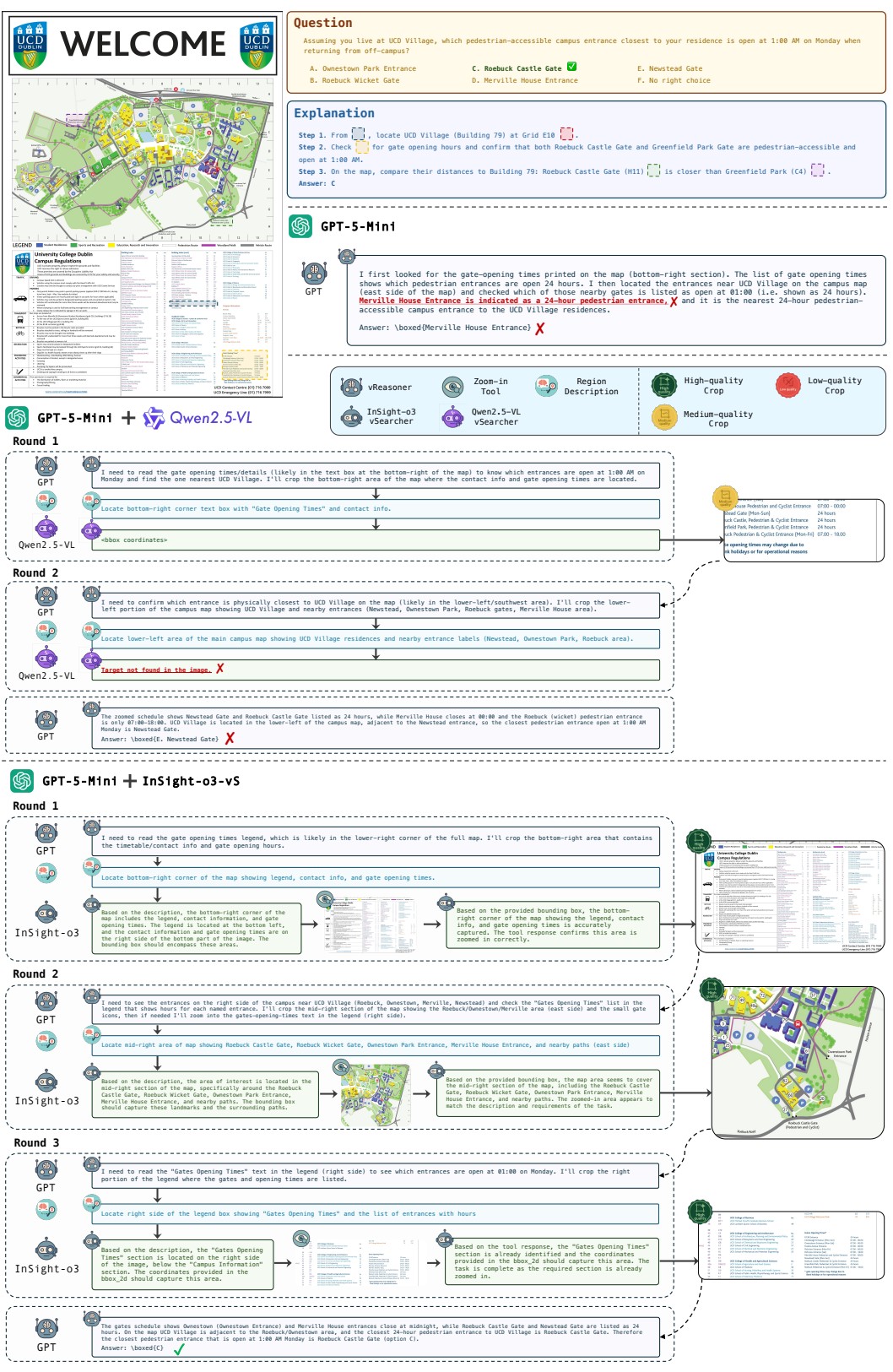

Figure 18: Example 2 (MAP case 2): reasoning traces of our model, GPT-5-mini, and GPT-5-mini + Qwen2.5-VL-7B-Instruct. Our INSIGHT-O3-VS accurately follows vReasoner's instructions and returns high-quality crops aligned with the described regions, leading to a correct answer. In contrast, Qwen2.5-VL-7B fails to return a valid crop in the final reasoning round, resulting in an incorrect answer.

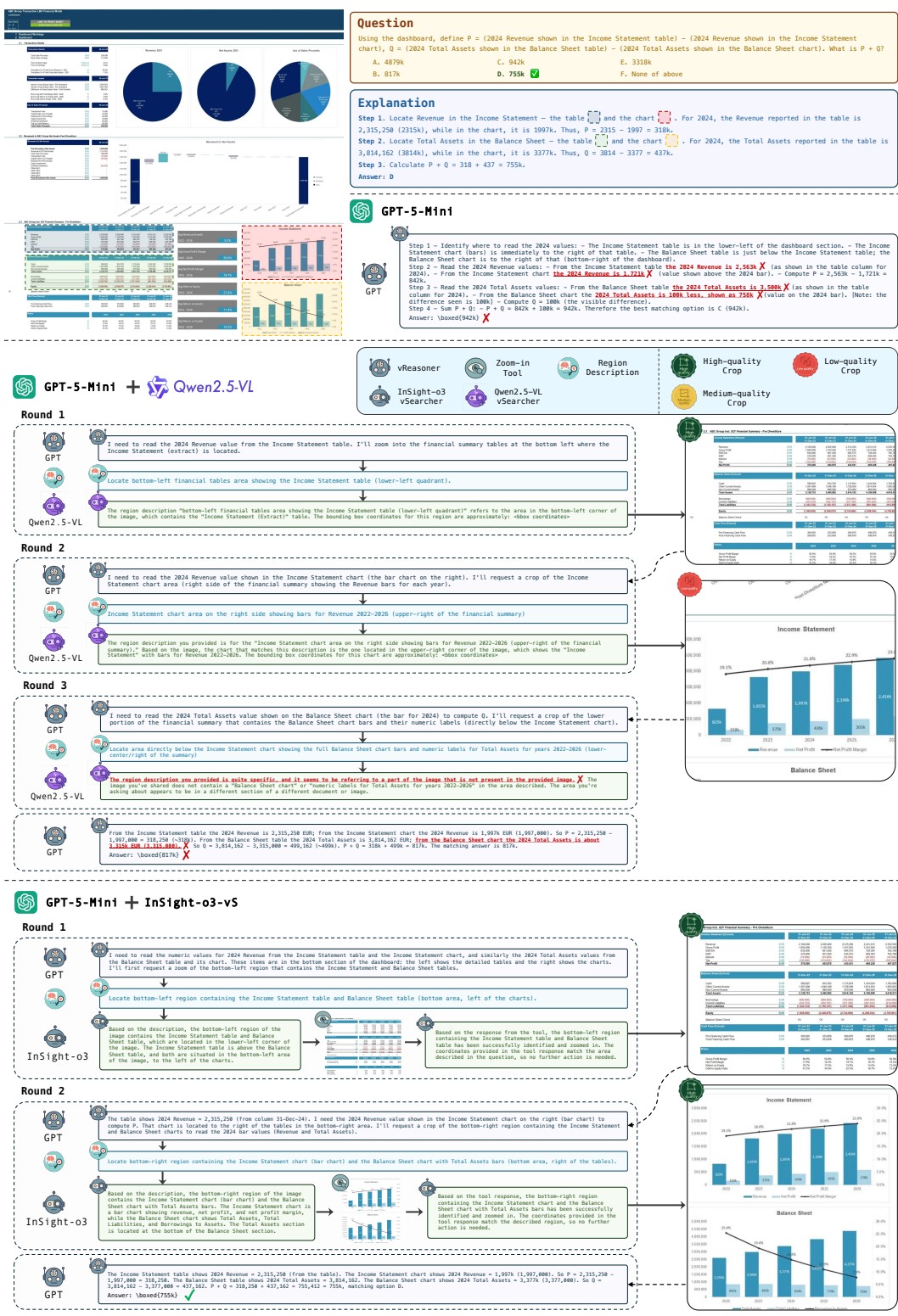

Figure 19: Example 3 (CHART case): reasoning traces of our model, GPT-5-mini, and GPT-5-mini + Qwen2.5-VL-7B-Instruct. INSIGHT-O3-VS effectively follows vReasoner's guidance and retrieves high-quality crops that fully capture the described regions. In contrast, Qwen2.5-VL-7B returns only partial crops in Round 2 and, in the Round 3, fails to produce a valid crop as requested by vReasoner, incorrectly concluding that the target region is absent, which leads to an incorrect answer.

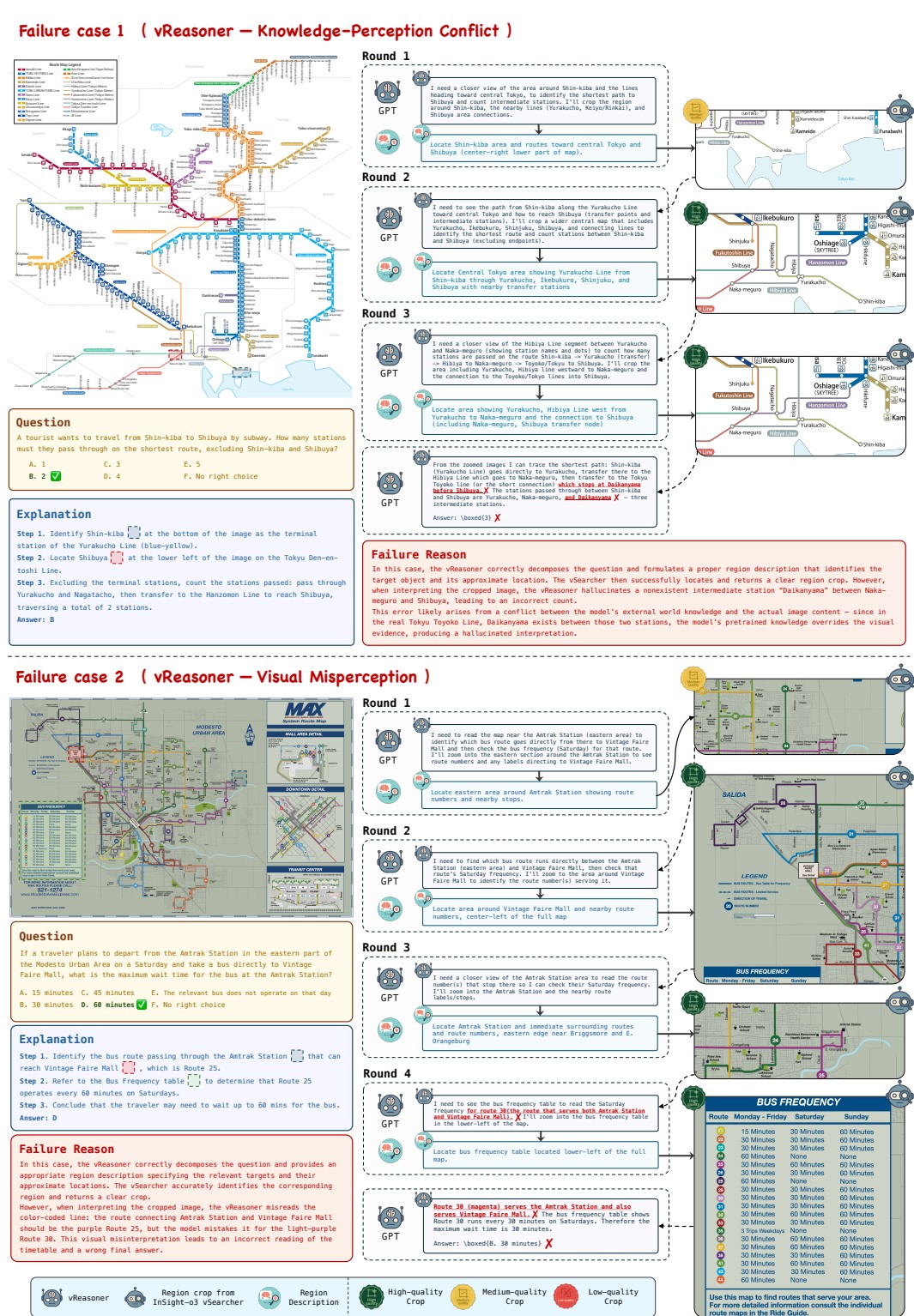

Figure 20: Failure cases 1 & 2 of INSIGHT-O3 (GPT-5-mini + InSight-o3-vS).

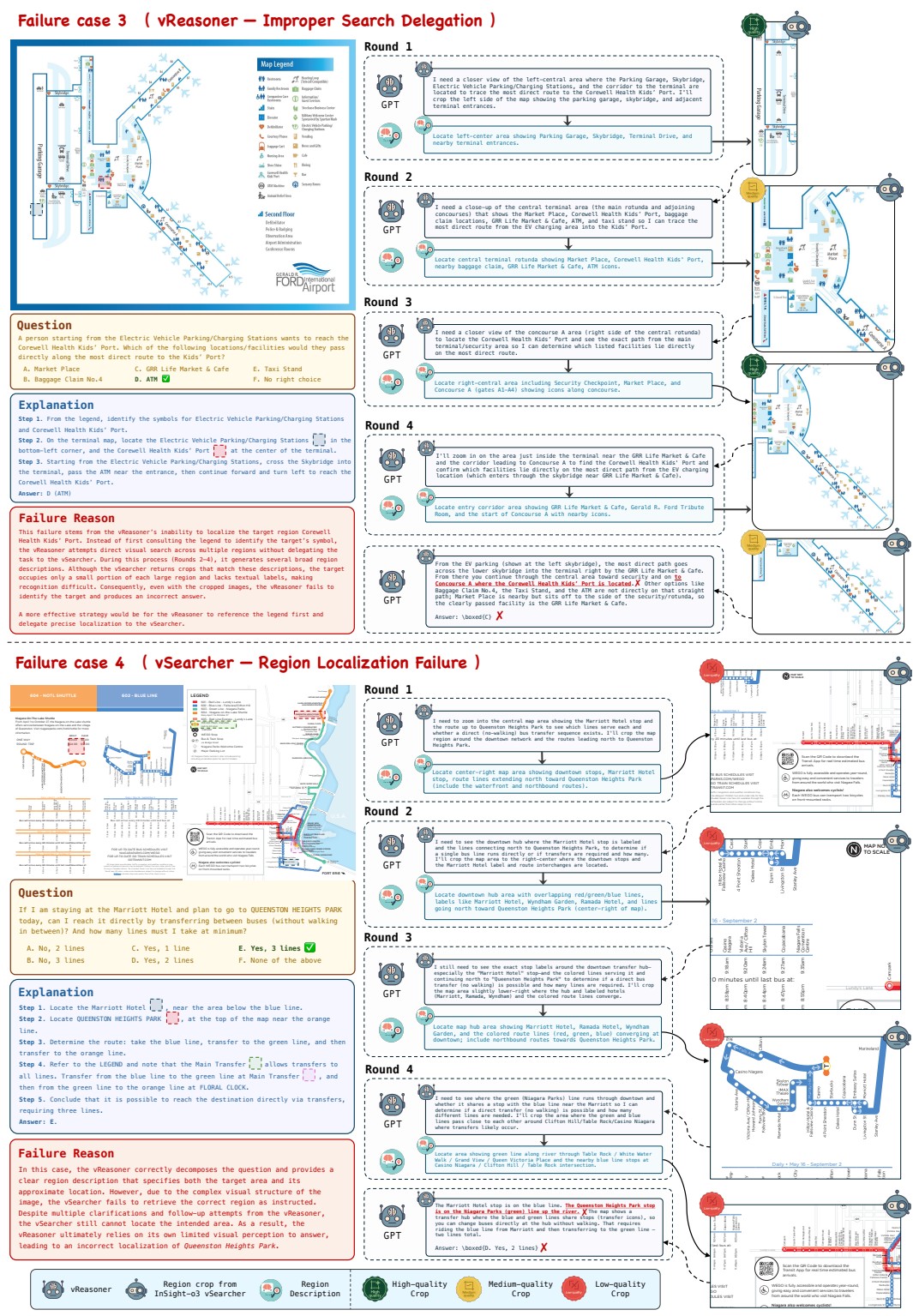

Figure 21: Failure cases 3 & 4 of INSIGHT-O3 (GPT-5-mini + InSight-o3-vS).

Table 8: Performance of Qwen3-VL-32B and INSIGHT-O3 with Qwen3-VL-32B vReasoner.

| vReasoner | vSearcher | V*-Bench | VisualProbe-Hard | O3-Bench |
|---|---|---|---|---|
| Qwen3-VL-32B | - | 86.0 | 28.6 | 60.4 |
| Qwen3-VL-32B | Qwen2.5-VL-7B (before RL) | 69.1 | 21.7 | 48.5 |
| Qwen3-VL-32B | Qwen2.5-VL-7B (after RL) | 90.1 | 36.8 | 61.2 |

### D.5 INSIGHT-O3 WITH QWEN3-VL VREASONER

Apart from closed-source proprietary models, we also experiment with open models like Qwen3-VL (Bai et al., 2025a) as the vReasoner of INSIGHT-O3. As shown in Table 8, INSIGHT-O3 (the last row) outperforms both the base Qwen3-VL-32B and the combination of Qwen3-VL-32B (vReasoner) and Qwen2.5-VL-7B (vSearcher) without RL. Although the advantage of INSIGHT-O3 on O3-BENCH is relatively small, we note that as open models continue to advance in tool use and general reasoning, the performance of INSIGHT-O3 with open vReasoners will further improve.

## E PROMPTS FOR O3-BENCH

### E.1 PROMPTS FOR INFORMATION EXTRACTION

For both the full image and cropped images, we feed them into Qwen2.5-VL-32B (Bai et al., 2025b) for information extraction using the same prompt as below.

```
### System

You are acting as a **precise visual information extractor**.
Given ONE image, you must (1) identify the image type, (2) write a **comprehensive,
strictly factual** caption, and (3) extract **complete OCR text** when present (with
special handling for tables).
Follow the rules **exactly** and return the output in the three sections shown under **
Output Format**.

---

### Global Principles

1) **No hallucinations.** Describe only what is visible. If something is unclear, write `[
illegible]` or `[partially obscured: ...]`.
2) **Be exhaustive.** Do not omit small text, legends, tick labels, footnotes, watermarks,
axis titles, subtitles, panel labels (e.g., `(a)`, `(b)`), or figure notes.
3) **Preserve fidelity.** Copy punctuation, capitalization, diacritics, signs (`+`, `-`),
units, and spacing **exactly**. Do not normalize numbers or rewrite text.
4) **Reading order.** When listing text outside of tables, use **top-to-bottom, left-to-
right** order.
5) **Language.** OCR text must remain in the **original language(s)**. The caption should
be in English unless the visible UI/page language is clearly not English; in that case,
keep captions in that language. Do **not** translate OCR unless the image itself contains a
 translation.
6) **No extra sections.** Output **only** the three required sections and nothing else.

---

### Image Type Identification (Section 1)

- Classify the image using one or more of the following types (multiple allowed if
appropriate):
  `chart`, `table`, `document/text page`, `diagram/flowchart`, `map`, `UI/screenshot`, `
  form`, `invoice/receipt`, `poster/flyer`, `scientific figure (multi-panel)`, `natural
  scene`, `legend`, `infographic`, `other (specify)`.

---

### Detailed Caption (Section 2)

Write a **dense, structured** caption that covers all critical elements. Use clear,
objective language and organize logically (left->right, top->bottom; or foreground->
background). Include the relevant sub-guidelines:
```

**A. Charts / Plots / Scientific Figures**
- State the figure title (if present), chart type(s), axes titles, **units**, tick labels,
gridlines, data series, markers/line styles, **legend and color/shape mappings**,
annotations, error bars, trend lines, and notable extrema/patterns (peaks, troughs,
monotonic trends, outliers).
- If multi-panel: identify panel labels `(a)`, `(b)`, ... and summarize each panel in order
.
- Mention any insets, callouts, footnotes, or sources.

**B. Tables / Forms / Receipts / Documents**
- Summarize what the table/document contains (topics/fields), approximate dimensions (e.g.,
 `~12 rows x 6 columns`), header rows, merged cells, checkboxes, stamps, signatures, page
numbers, and footers/footnotes.
- Call out key sections (headings, lists, paragraphs), logos, and seals.

**C. Maps**
- Report title, compass/north arrow, scale/scale bar, coordinate grid or lat/long,
boundaries, regions, routes/lines with **color-to-meaning mapping** (from legend), symbols/
icons (e.g., hospitals, stations), labels for places/roads, insets, and any zoning/heat
color ramp.
- Include legend content (categories and their visual encodings).

**D. UI / Screenshots**
- App/site name, window title, menus/toolbars, visible controls (buttons, toggles,
checkboxes, dropdowns, search fields), selected/disabled states, scroll position,
timestamps, status bars, notifications, dialogs, visible file paths, and version strings.

**E. Natural / Real-World Scenes**
- Enumerate salient objects, text on signs/labels, relative positions (e.g., ``a red sign
above the doorway"), counts for repeated items, conditions (day/night, indoor/outdoor), and
 visible brands/logos.

> Do **not** invent interpretations or causal explanations. Keep to what is visually
supported.

---

### OCR Extraction (Section 3)

Extract **all visible text**. Follow these rules:

**General OCR Rules**
- Use **natural reading order** (top->bottom, left->right).
- Preserve original line breaks and spacing.
- If text is repeated (e.g., in a watermark), list it once and note `(repeats)` if
necessary.
- If a character/word is uncertain, write it as `[illegible]` or `[?]` without guessing.

**Tables (very important)**
- When an area is a table, extract it immediately using **Markdown table syntax**.
- Preserve the **exact** row/column structure and header rows; if a cell has line breaks,
use `
` or `\n`.
- For merged cells, repeat the visible text in each affected cell and note `(merged)` once
after the table.
- If multiple tables exist, label them sequentially as `Table 1`, `Table 2`, ... in the
order they appear.

**Documents / Text Pages / Forms**
- Extract headings, paragraphs, lists, captions, footnotes, headers/footers exactly as
shown. Maintain indentation and list markers.

**Charts / Maps / Diagrams / UI / Natural Scenes**
- Extract **all textual elements** present: titles, subtitles, axis labels, tick labels,
legend entries, series labels, annotations, callouts, map labels (places/roads/lines), UI
labels (menu items, buttons, tooltips), signs, badges, and watermarks.

If **no text** is present, write `None`.

---

### Output Format (return EXACTLY this Markdown structure; no code fences)

## Image Identification
<one or more types from the allowed list; add brief justification if mixed>

## Detailed Caption
<dense, strictly factual caption covering all visible elements per rules above>

## Extracted OCR (if any)

```
<EITHER: full Markdown tables + remaining text in reading order; OR: all non-table text in
reading order; write "None" if no text>
```

## E.2  PROMPTS FOR CONSTRUCTING O3-BENCH

For chart images, we use the following prompt for GPT-5 (OpenAI, 2025a) to automatically generate
QA instances.

```
You are an expert assessment-item author who designs rigorous **multi-hop visual-reasoning
** questions to benchmark "think-with-images" abilities on **dense diagrams, charts, tables
, and schematics**. The goal is to generate items so challenging that today's strongest
MLLMs score **<= 50%** without external tools.

You will receive:

### INPUTS
1. **GLOBAL_OCR** - OCR or caption text describing the entire image/page.
   {GLOBAL_OCR}
2. **GLOBAL_CAPTION** - caption text describing the entire image/page.
   {GLOBAL_CAPTION}
3. **LAYOUTS** - a list of cropped regions. Each layout contains:
   - `layout_id` - unique numeric ID (for your internal reference only).
   - `caption_or_ocr` - OCR or descriptive caption of the cropped region.
   {LAYOUTS}

> **Important constraint:**
> In the **question** and **options**, you must use only natural labels/text present in the
 `GLOBAL_OCR` or `GLOBAL_CAPTION` or `caption_or_ocr`.
> **Never** mention `layout_id`, `region`, `crop`, `panel`, `box`, or any similar tokens.
Layouts are **reference-only** to help you construct questions; they may be cited in the **
explanation** but not in the question or options.

---

## TASK
Create **3-5 independent multiple-choice questions** that each requires **>=2 distinct
visual hops** across layouts or global text. These must be **multi-step items directly**,
not derived from single-step seeds. All facts must be grounded in the provided OCR/captions
 only.

---

## QUESTION-DESIGN RULES
1. **Multi-hop inference (required)**
   Each question must integrate information from at least **two different layouts** or from
    global + local text. Valid patterns include:
   - Cross-table lookup & join (match category in Layout A to code/key in Layout B, then
   filter by condition).
   - Table <-> chart alignment (map series/labels from one layout to another).
   - Diagram <-> table mapping (use schematic labels to query corresponding rows/values).
   - Temporal alignment (identify when a threshold is crossed in one chart, then fetch
   related info from another).
   - Proportions/ratios/ranks (compute shares from one region and compare with targets in
   another).
   - Exceptions/constraints (apply footnote conditions from one layout before interpreting
   values elsewhere).

2. **Analytical realism**
   Situate each question in a plausible scenario (finance, science, education, operations,
   product metrics, etc.) while remaining strictly grounded in the provided OCR/captions.

3. **Difficulty control**
   Questions should require careful scanning, cross-referencing, and light calculations (
   differences, ratios, ranking). Avoid items that can be answered at a glance.

4. **Units, scales, rounding**
   Always follow the units/scales given in the OCR. If rounding is necessary, state the
   rounding rule in the **explanation**.

5. **Ambiguity guardrails**
   Ensure exactly **one correct choice** among A-E, unless `F` ("No right choice") is
   correct. Adjust conditions to avoid ties.

6. **Label fidelity**
   Copy text exactly as provided (case, spelling, diacritics). Never use external/world
   knowledge.
```

```
---

## ANSWER-OPTION RULES
1. Provide **exactly six** options, one per line, labeled 'A.' ... 'F.'
2. 'F.' must always be exactly 'No right choice'.
3. Place the correct answer randomly among 'A.'-'E.'; <=10% of items may have 'F' as the
correct answer.
4. Distractors must be plausible and drawn from actual text/numbers in the OCR/captions.
5. Make options mutually exclusive' no meta-options like "All of the above".

---

## EXPLANATION RULES
- Provide a **step-by-step chain**: 'Step 1: ...', 'Step 2: ...', etc.
- Explicitly cite which layouts were used as '[layout X]'.
- Show all computations (e.g., "(132 - 95) / 95 = 0.389 = 38.9% [layout 4]").

---

## OUTPUT FORMAT
Return **only** the following JSON array - no extra text, no markdown outside the code
block, no commentary:

```json
[
  {{
    "question": "...",
    "options": "A. ...\nB. ...\nC. ...\nD. ...\nE. ...\nF. No right choice",
    "answer": "C",
    "explanation": "Step 1: ... [layout 2]\nStep 2: ... [layout 5]\nStep 3: ..."
  }},
  ...
]
```
```

For map images, we use the following prompt for GPT-5 (OpenAI, 2025a) to automatically generate QA instances.

```
You are an expert evaluation-item author who designs rigorous **multi-hop visual-reasoning
** questions to benchmark "think-with-images" abilities on **maps** (bus/metro networks,
terminals, malls, festivals, parks, etc.). Items should be challenging enough that today's
strongest MLLMs achieve **<= 50%** accuracy.

You will receive:

### INPUTS
1. **GLOBAL_OCR** - OCR or caption text describing the entire map.
   {GLOBAL_OCR}
2. **GLOBAL_CAPTION** - caption text describing the entire map.
   {GLOBAL_CAPTION}
3. **LAYOUTS** - a list of cropped map regions. Each layout contains:
   - 'layout_id' - unique numeric ID (for your internal reference only).
   - 'caption_or_ocr' - OCR or descriptive caption of the cropped region.
   {LAYOUTS}

> **Hard requirement:** In the **question** and **options**, you must use only natural
labels/text found in 'GLOBAL_OCR' or 'GLOBAL_CAPTION' or 'caption_or_ocr'.
> Do **not** mention 'layout_id', 'region', 'crop', 'box', 'panel', or similar. Layouts are
 **reference-only** for your reasoning; you may cite them in the **explanation**, but never
 in the question or options.

---

## TASK
Generate **3-5 independent multiple-choice questions**.
Each question must require **>=2 distinct reasoning hops** that combine information across
different layouts or between global and local OCR. All facts must be image-grounded.

---

## 1) QUESTION-DESIGN RULES
1. **Multi-hop reasoning (mandatory).** Examples of valid hops:
   - Legend <-> line color <-> stop/zone.
   - Grid index <-> label <-> adjacency.
   - Level/floor marker <-> facility <-> inset.
   - Route <-> timetable <-> destination.
   - Symbol <-> restriction <-> path feasibility.
```

```
    - Distance/scale <-> number of segments <-> travel time.

2. **Image/OCR grounded only.** Do not use external/world knowledge.

3. **Diversity.** Vary question styles (routing, conditional reachability, transfer logic,
adjacency/containment, count/compare, scale-based).

4. **Difficulty target.** Avoid "at-a-glance" items. Require cross-checking, counting, or
lightweight calculation.

5. **Label fidelity.** Copy map labels exactly (case, spelling, diacritics).

6. **Uniqueness.** Ensure exactly **one correct answer** among A-E, unless F ("No right
choice") is deliberately correct.

7. **Units & scale.** If computing length/time/segments, use the map's own scales, symbols,
 or counts.

---

## 2) ANSWER-OPTION RULES
1. Provide **exactly six** options labeled 'A.' ... 'F.'.
2. 'F.' must always be 'No right choice'.
3. Normally, the correct answer is among A-E; only rarely (<10%) should 'F' be correct.
4. Distractors must be plausible, drawn from real map text/numbers, and mutually exclusive.
5. No meta-options ("All of the above").

---

## 3) EXPLANATION RULES
- Provide a **step-by-step reasoning chain**.
- Explicitly cite layouts used as '[layout X]'.
- Make hops and computations explicit (e.g., "Count 5 stops along Red Line [layout 3] and
compare to 4 stops in Zone B [layout 5]").

---

## 4) OUTPUT FORMAT
Return **only** the following JSON array-no extra commentary or markdown:

```json
[
  {{
    "question": "...",
    "options": "A. ...\nB. ...\nC. ...\nD. ...\nE. ...\nF. No right choice",
    "answer": "B",
    "explanation": "Step 1: ... [layout 1]\nStep 2: ... [layout 4]\nStep 3: ..."
  }},
  ...
]
```
```

### E.3 PROMPTS FOR EVALUATION

For proprietary models/systems, we adopt the following thinking prompt (Guo et al., 2025a) as their system prompt. The prompt is used across all benchmarks except O3-Bench.

```
A conversation between User and Assistant. The user asks a question, and the Assistant
solves it. The assistant first thinks about the reasoning process in the mind and then
provides the user with the answer. The reasoning process and answer are enclosed within <
think> </think> and <answer> </answer> tags, respectively, i.e., <think> reasoning process
here </think><answer> answer here </answer>.
```

On O3-Bench, we use the models' default system prompts as they usually lead to better performance. On other benchmarks, the performance difference is negligible (mostly within 1%) except only that GPT-5-nano performs much worse on VisualProbe-Hard when the default system prompt is used (accuracy dropping from 21.7% to 16.0%).

For open models, e.g., InternVL3.5 (Wang et al., 2025e), Qwen2.5-VL (Bai et al., 2025b), and Qwen3-VL models (Bai et al., 2025a), we use their default system prompts since they are relatively sensitive to the system prompt. We note that the thinking prompt above often leads to suboptimal performance of Qwen3-VL models.

# F  PROMPTS FOR INSIGHT-O3

## F.1  VREASONER PROMPTS

We use the following prompts for vReasoner during training.

```
[SYSTEM]
You are a visual assistant. Your goal is to answer a question based on an image.

First, think step by step to identify which visual facts you need from the image to answer
the question. If the visual information is insufficient or unclear, call the visual search
tool by providing a concise region description:
<tool_call>region_description={...}</tool_call>

The tool will search the image and return a cropped view of the target region. You may
repeat this process until you have enough evidence to answer confidently. The tool is not
always precise -- evaluate its output critically. If it looks incorrect or off-target,
refine your description and try again.

Region description guidance:
- Use concise, visually grounded targets (e.g., a chart, an object, a text block, a
distinct area)
- Optionally include approximate location (e.g., top-left, bottom-right, center)
- Avoid non-visual or ordinal references (e.g., "the third largest bar", "the second row's
number")
- Describe only one region per tool call; do not request multiple regions in a single
description

Output format:
- Put your reasoning process inside <think>...</think>.
- Immediately after </think>, output your assessment of the most recent tool result (if any
) formatted as <tool_feedback>helpful/unhelpful</tool_feedback>.
  This should indicate whether the result returned by the previous tool call is relevant to
  your prior region description and helpful to answering the question. If it misses the
  key information you are looking for, it is unhelpful. If no previous tool result exists (
  e.g., the first turn), output <tool_feedback>NA</tool_feedback>.
- Immediately after </tool_feedback>, do exactly one of:
  1) Call the tool; or
  2) Provide the final answer (no tool call) -- include the result in \\boxed{...}. Do not
  mix tool calls and answers in the same turn.
- If you need to call the tool, provide the region description using the exact format <
tool_call>region_description={...}</tool_call>.
You must strictly follow the output format, otherwise your answer will be judged as wrong.

A multi-turn format example:
Assistant:
<think>{your step-by-step analysis; decide if more detail is needed}</think>
<tool_feedback>NA</tool_feedback>
<tool_call>region_description={concise, visually grounded target (optionally with location)
}</tool_call>

User:
[Zoomed-in image + guidance (e.g., "Based on your description, here is the zoomed-in image.
 Please continue your analysis; you may call the tool again or provide your final answer if
 sufficient.")]

Assistant:
<think>{updated analysis based on the zoomed-in view; decide whether to refine or answer}</
think>
<tool_feedback>unhelpful</tool_feedback>
<tool_call>region_description={next concise target (optionally with location)}</tool_call>

(Repeat the User -> Assistant pattern as needed until enough evidence is gathered.)

Assistant (final turn):
<think>{final reasoning; explain why the available visual evidence is sufficient}</think>
<tool_feedback>helpful</tool_feedback>
Answer: \\boxed{...}

[USER]
{question}<image>

[ASSISTANT]
...

[USER]
Based on your description, here is the zoomed-in image.
```

```
Please continue your analysis. After the analysis, state your assessment of the previous
tool result using <tool_feedback>helpful/unhelpful</tool_feedback>, then do one of the
following:
- Call the tool again if you believe more visual detail is needed; or
- Provide your final answer if the current information is sufficient.
<image>

[ASSISTANT]
...

[USER]
Based on your description, here is the zoomed-in image.

You have reached the limit for using the visual tool and cannot call it again.
In this turn, after reasoning step by step, output your assessment of the previous tool
result using exactly <tool_feedback>helpful/unhelpful</tool_feedback>, and then, based on
the available information, provide your final answer using the required format.
<image>

[ASSISTANT]
...
```

In case that vSearcher is unable to find a region that matches the region description provided by vReasoner, we use the following user prompt to notify vReasoner.

```
[USER]
The visual searcher could not locate the requested target in the image based on your
description.

Please adjust or refine your region description (for example, refer to a larger, clearly
visible area) and continue your analysis. Think first, then state your assessment of the
previous tool result using <tool_feedback>helpful/unhelpful</tool_feedback>. Finally, do
exactly one of the following:
- Call the tool again with a revised description; or
- Provide your final answer if the current information is sufficient.
```

Occasionally, vReasoner may fail to follow the format instructions. When this happens, we use the following user prompt to ask vReasoner to generate a new response:

```
[USER]
In your previous response, neither a tool call nor a final boxed answer was detected, or
you didn't output your assessment of the previous tool result in the correct format.

Think first, and then include your assessment of the previous tool result using exactly <
tool_feedback>helpful/unhelpful</tool_feedback> (or <tool_feedback>NA</tool_feedback> if
there is no previous result). Finally, do exactly one of the following:
- If you still need more visual detail, call the tool using the exact format:
  <tool_call>region_description={...}</tool_call>
- Otherwise, provide the final answer now and include the result in \\boxed{...}.
```

During inference, we do not ask vReasoner to provide any feedback to the tool. The prompts are slightly simplified as follows:

```
[SYSTEM]
You are a visual assistant. Your goal is to answer a question based on an image.

First, think step by step to identify which visual facts you need from the image to answer
the question. If the visual information is insufficient or unclear, call the visual search
tool by providing a concise region description:
<tool_call> region_description={...} </tool_call>

The tool will search the image and return a cropped view of the target region. You may
repeat this process until you have enough evidence to answer confidently. The tool is not
always precise -- evaluate its output critically. If it looks incorrect or off-target,
refine your description and try again.

Region description guidance:
- Use concise, visually grounded targets (e.g., a chart, an object, a text block, a
distinct area)
- Optionally include approximate location (e.g., top-left, bottom-right, center)
- Avoid non-visual or ordinal references (e.g., "the third largest bar", "the second row's
number")
```

```
- Describe only one region per tool call; do not request multiple regions in a single
description

Output format:
- Put your reasoning process inside <think>...</think>.
- When you need to call the tool, you need to provide the region description using the
format <tool_call>region_description={...}</tool_call>.
- Immediately after each </think>, do exactly one of:
  1) Call the tool; or
  2) Provide the final answer (no tool call) -- include the result in \\boxed{...}. Do not
  mix tool calls and answers in the same turn.
You must strictly follow the output format, otherwise your answer will be judged as wrong.

A multi-turn format example:
Assistant:
<think>{your step-by-step analysis; decide if more detail is needed}</think>
<tool_call> region_description={concise, visually grounded target (optionally with location
)} </tool_call>

User:
[Zoomed-in image + guidance (e.g., "Based on your description, here is the zoomed-in image.
 Please continue your analysis; you may call the tool again or provide your final answer if
 sufficient.")]

Assistant:
<think>{updated analysis based on the zoomed-in view; decide whether to refine or answer}</
think>
<tool_call> region_description={next concise target (optionally with location)} </tool_call
>

(Repeat the User -> Assistant pattern as needed until enough evidence is gathered.)

Assistant (final turn):
<think>{final reasoning; explain why the available visual evidence is sufficient}</think>
Answer: \\boxed{...}

[USER]
{question}<image>

[ASSISTANT]
...

[USER]
Based on your description, here is the zoomed-in image.

Please continue your analysis. You may:
- Call the tool again if you believe more visual detail is needed; or
- Provide your final answer if the current information is sufficient.
<image>

[ASSISTANT]
...

[USER]
Based on your description, here is the zoomed-in image.

You have reached the limit for using the visual tool and cannot call it again.
In this turn, based on the available information, provide your final answer using the
required format.
<image>

[ASSISTANT]
...
```

When vSearcher can't find the target region, we use the following user prompt to notify vReasoner:

```
[USER]
The visual searcher could not locate the requested target in the image based on your
description.

Please adjust or refine your region description (for example, refer to a larger, clearly
visible area) and continue your analysis. You may:
- Call the tool again with a revised description; or
- Provide your final answer if the current information is sufficient.
```

When vReasoner fails to follow the format instruction described in the system prompt, we the following user prompt to ask vReasoner to generate a new response:

```
[USER]
In your previous response, neither a tool call nor a final boxed answer was detected.

Please do exactly one of the following:
- If you still need more visual detail, call the tool using the exact format:
  <tool_call>region_description={...}</tool_call>
- Otherwise, provide the final answer now and include the result in \\boxed{...}.
```

## F.2 vSEARCHER PROMPTS

We use the following prompts for vSearcher during both training and evaluation after training. The prompts are adapted from DeepEyes (Zheng et al., 2025). For the last turn, we notify vSearcher in the user prompt that it has reached tool call limit.

```
[SYSTEM]
You are a helpful assistant.

# Tools
You may call one or more functions to assist with the user query.
You are provided with function signatures within <tools></tools> XML tags:
<tools>
{"type":"function","function":{"name":"image_zoom_in_tool","description":"Zoom in on a
specific region of an image by cropping it based on a bounding box (bbox) and an optional
object label.","parameters":{"type":"object","properties":{"bbox_2d":{"type":"array","items
":{"type":"number"},"minItems":4,"maxItems":4,"description":"The bounding box of the region
 to zoom in, as [x1, y1, x2, y2], where (x1, y1) is the top-left corner and (x2, y2) is the
 bottom-right corner."},"label":{"type":"string","description":"The name or label of the
object in the specified bounding box (optional)."}},"required":["bbox"]}}}
</tools>

# How to call a tool
Return a json object with function name and arguments within <tool_call></tool_call> XML
tags:
<tool_call>
{"name": <function-name>, "arguments": <args-json-object>}
</tool_call>

Example:
<tool_call>
{"name": "image_zoom_in_tool", "arguments": {"bbox_2d": [10, 20, 100, 200], "label": "the
apple on the desk"}}
</tool_call>

[USER]
<image>
Locate {target}.
Think first, call image_zoom_in_tool if needed, then answer with the bbox coordinates in [
x1, y1, x2, y2] format (or [0, 0, 0, 0] if you can't locate it). Format strictly as: <
think>...</think>  <tool_call>...</tool_call> (if tools needed)  <answer>[x1, y1, x2, y2]</
answer> (otherwise)

[ASSISTANT]
...

[USER]
<tool_response><image></tool_response>
Think first, call image_zoom_in_tool if needed, then answer with the bbox coordinates in [
x1, y1, x2, y2] format (or [0, 0, 0, 0] if you can't locate it). Format strictly as: <
think>...</think>  <tool_call>...</tool_call> (if tools needed)  <answer>[x1, y1, x2, y2]</
answer> (otherwise)

[ASSISTANT]
...

[USER]
<tool_response><image></tool_response>
You have reached the tool call limit and cannot call tools anymore.
Think first, call image_zoom_in_tool if needed, then answer with the bbox coordinates in [
x1, y1, x2, y2] format (or [0, 0, 0, 0] if you can't locate it). Format strictly as: <
think>...</think>  <answer>[x1, y1, x2, y2]</answer>

[ASSISTANT]
...
```

Without RL, Qwen2.5-VL-7B has difficulty following the instructions given in the above prompts. So, for evaluating vReasoner + Qwen2.5-VL-7B, we use the following simple prompt:

```
[SYSTEM]
You are a helpful assistant.

[USER]
Given an image and a region description, locate the region that best matches the
description and output its bounding box coordinates as [x_min, y_min, x_max, y_max].

If the target cannot be found, output [0, 0, 0, 0].

Region description:
{target}

Now, output the coordinates in format [x_min, y_min, x_max, y_max]:
<image>

[ASSISTANT]
...
```

### F.3 PROMPTS FOR ANSWER VERIFICATION

For answer verification, we use GPT-5-nano (OpenAI, 2025a) and feed it with the following prompt.

```
You are given an image-based question, the ground truth (GT) answer, and a model's answer.

Compare the model's answer with the GT answer:

- If the model's answer matches the GT answer visually or semantically, reply with <correct
>.
- If it doesn't match, or if uncertain, reply with <wrong>.

Only reply with <correct> or <wrong>, no explanations.

Question: {question}
GT Answer: {gt_answer}
Model Answer: {model_answer}
```

## G EVALUATION BENCHMARKS

### G.1 NATURAL-IMAGE BENCHMARKS

- **V⋆-Bench** (Wu & Xie, 2024) is designed to test a model's ability to attend to high-resolution, detail-rich images. V⋆-Bench consists of 191 challenging natural images (sourced from the SA-1B Segment Anything dataset (Kirillov et al., 2023)) and focuses on two fine-grained visual tasks: attribute recognition (identifying specific object attributes like color or material) and spatial relationship reasoning (determining the relative positions of objects). By requiring accurate visual grounding of small details, V⋆-Bench exposes the limitations of models that rely on coarse image understanding.

- **Tree-Bench** (Wang et al., 2025a), like V⋆-Bench, uses high-quality, object-dense natural images (drawn from the SA-1B dataset) to evaluate fine-grained visual reasoning. However, Tree-Bench places additional emphasis on traceable evidence and complex reasoning. Each of its 405 visual question-answer pairs is annotated with bounding-box evidence for the correct answer, and many questions require second-order reasoning about object interactions or spatial hierarchies rather than simple identification.

- **VisualProbe** (Lai et al., 2025) pushes visual reasoning to an even harder regime. It features high-resolution images with very small target objects and many distractors, making it "super challenging" and necessitating iterative, trial-and-error search by the model. VisualProbe is organized into easy, medium, and hard subsets; VisualProbe-Hard denotes the toughest set of questions (106 in total) that often cannot be solved in a single glance. Compared to V⋆-Bench and Tree-Bench, VisualProbe-Hard scenarios demand an active visual search strategy: the model may need to zoom in on different regions or sequentially explore the image to find relevant details.

Table 9: Comparison of O3-BENCH and related benchmarks.

| Benchmark | # of QAs | Image domains | Layout/target boxes | Detailed explanations | Multi-hop | Avg. resolution (width×height) |
|---|---|---|---|---|---|---|
| V$^\star$-Bench | 191 | natural (100%) | ✓ | ✗ | ✗ | 2246×1583 |
| Tree-Bench | 405 | natural (100%) | ✗ | ✗ | ✗ | 2152×1615 |
| VisualProbe$_{Hard}$ | 106 | natural (100%) | ✗ | ✗ | ✗ | 4944×3980 |
| HR-Bench$_{4K}$ | 200* | natural (89%), chart (5%), map (6%) | ✗ | ✗ | ✗ | 4024×3503 |
| MME-RealWorld | ∼29K | natural (>60%), chart (25%), map (6%), etc. | ✗ | ✗ | ✗ | 2708×1844 |
| O3-Bench | 345 | chart (47%), map (53%) | ✓ | ✓ | ✓ | 4602×3967 |

* Roughly 200 distinct QA pairs. The original paper reported 800 but most are the same questions with scrambled options.

Table 10: Comparison of O3-BENCH and related benchmarks on chart & map.

| Benchmark | Avg. resolution (width×height) | Avg. acc. of GPT-5-mini | Avg. # of vSearch steps |
|---|---|---|---|
| HR-Bench$_{4K}$ | 4032×2509 | 79.6 | 2.3 |
| MME-RealWorld | 1875×1269 | 83.8[†] | 1.0 |
| O3-Bench | **4602×3967** | **39.0** | **2.9** |

[†] Based on 500 random samples.

## G.2 MIXED BENCHMARKS

- **HR-Bench** (Wang et al., 2025f) is a benchmark deliberately designed to evaluate models on ultra high-resolution images (up to 4K-8K pixels). It addresses a key limitation of prior multimodal tests (which max out at ∼2K resolution) by presenting tasks that cannot be solved with down-sampled images. HR-Bench is split into two sub-task categories: (1) Fine-grained Single-instance Perception (FSP), with tasks like identifying detailed attributes of a single object, reading text via OCR, or responding to visual prompts on an image; and (2) Fine-grained Cross-instance Perception (FCP), which includes more complex multi-object challenges such as analyzing maps, interpreting charts/graphs, and assessing spatial relationships among multiple items. Each sub-task contains 100 queries on 8K-resolution images (with a downsampled 4K version also provided for efficiency).

- **MME-RealWorld** (Zhang et al., 2024b) is a large-scale, comprehensive benchmark that evaluates models across a wide spectrum of real-world visual tasks. It comprises 13,366 high-quality images (average ∼2000×1500 resolution) and 29,429 QA pairs, spanning 43 distinct task types grouped into five real-world scenarios, curated from various datasets (Agustsson & Timofte, 2017; Liu et al., 2020; Zhang et al., 2021; Yang et al., 2023; Li et al., 2022; Sun et al., 2022; Sachdeva et al., 2024; Zhu et al., 2021; Jia et al., 2021). These scenarios cover diverse applications such as autonomous driving (*e.g.*, interpreting traffic scenes), video surveillance (*e.g.*, counting vehicles in an overhead street video), remote sensing (*e.g.*, identifying and counting tiny objects in satellite maps), sports and entertainment (*e.g.*, reading a scoreboard in a broadcast image), and others. MME-RealWorld is notable as the largest fully human-annotated multimodal benchmark to date. Given the benchmark's scale, we employ the official MME-RealWorld lite version[3] for efficiency, which uses a subset of 50 samples per task to speed up evaluation without significantly altering the task distribution.

### G.3 Comparison of O3-Bench and Related Benchmarks

In Table 9 and Table 10, we compare O3-BENCH with related benchmarks and highlight their key differences. Compared with the most closely related MME-RealWorld, we note that on the overlapping domains (i.e. chart & map): (i) the average image resolution of O3-BENCH is *significantly higher* ($4602\times3967$ of ours *vs* $1875\times1269$ of MME-RealWorld); (ii) the average accuracy of GPT-5-mini on O3-BENCH is much lower (39.0% of ours *vs* 83.8% of MME-RealWorld); and (iii) the average number of visual search steps produced by InSight-o3 for O3-BENCH is *2.9×* that of MME-RealWorld. In addition, O3-BENCH provides layout boxes and detailed explanations for each QA pair, while most of the benchmarks do not. The explanations can help the community easily verify the correctness of the answers. These differences show that O3-BENCH is of exceptional quality and much harder to solve than the other related benchmarks.

Apart from quality, the scale of O3-BENCH is on par with most of the fine-grained perception multimodal understanding benchmarks, e.g., V$^\star$-Bench (191), Tree-Bench (405), VisualProbe-Hard (106), HR-Bench-4K (200*), and commonly-used benchmarks in other multimodal research areas; to list a few: Math: MathVision test-mini (304) (Wang et al., 2024), MathVista test-mini (1K) (Lu et al., 2023); VQA: RealWorldQA[4] (765); Embodiment/Spatial Understanding: ERQA (400) (Abeyruwan et al., 2025), RefSpatial-Bench (277) (Zhou et al., 2025); Agent: MIA Bench (400) (Qian et al., 2024), OSWorld (389) (Xie et al., 2024), AndroidWorld (116) (Rawles et al., 2024); Fine-grained Perception: V$^\star$-Bench. These benchmarks were used to evaluate the performance of Qwen3-VL (Bai et al., 2025a). As with O3-BENCH, these benchmarks are relatively small mainly because of the difficulty in data collection. Nevertheless, they have served the timely purpose of evaluating frontier models in the respective fast-developing areas.

O3-BENCH only focuses on composite charts and digital maps for two main reasons. First, they are *representative* of most use cases of thinking with images in the digital domain (as opposed to the natural domain). More specifically, composite charts represent *structured* images (with clear delineations between different layout regions) and often require more *abstract* reasoning (e.g., computing the difference of two quantities). On the other hand, digital maps represent images with less structure and more *organic* layouts. They usually require more *visual* reasoning (e.g., finding the shortest route from location A to B). Together, we argue that O3-BENCH is generally sufficient for evaluating the thinking-with-image capability of current multimodal models in the digital domain. As for the natural domain, there are already a number of high-quality benchmarks for thinking with natural images (some are listed in Table 9). O3-BENCH precisely fills the gap of existing benchmarks while striking a balance between evaluation efficiency and generality.

## H Use of Large Language Models

For this paper, we used LLMs to aid and polish writing. Specifically, we asked LLMs to paraphrase certain sentences or paragraphs to improve clarity and readability. We also used LLMs to conduct literature survey. All literature references and related discussions in this paper are manually checked and proofread to ensure correctness.

---

[3] https://huggingface.co/datasets/yifanzhang114/MME-RealWorld-Lite
[4] https://huggingface.co/datasets/xai-org/RealworldQA

