# OpenReview forum: "InSight-o3: Empowering Multimodal Foundation Models with Generalized Visual Search"
_ICLR.cc/2026/Conference — ICLR 2026 Poster_

### Official Review · Reviewer_7mrZ · 2025-10-28

**Soundness:** 2
**Presentation:** 3
**Contribution:** 2
**Rating:** 4
**Confidence:** 3

**Summary:**

This paper proposes a novel multi-agent framework that decomposes multimodal reasoning into two agents: a visual reasoning agent (vReasoner) that performs CoT reasoning and provides answers, and a visual search agent (vSearcher) that outputs coordinates given a description. Given an input, the vReasoner processes the answer and, when needed, calls the vSearcher to find a relevant visual region. The authors also built InSight-o3-vS using a novel RL training pipeline and presented the O3-Bench to evaluate complex visual reasoning tasks.

**Strengths:**

1. I think decomposing mulitmodal reasoning model into two agents can be efficient without the need of training the whole model, especially with strong closed source models we cannot train.
2. Training pipeline and constructing collage data for vSearcher is interesting and novel.
3. O3-Bench can be beneficial to the community.
4. Nice addiotional observation report from the hands-on experience.

**Weaknesses:**

1. **Reliance on Closed-Source Models**: The experiments rely heavily on closed-source models (like GPT-5-mini), which are difficult to use for reproduction or further training. As a result, training the vSearcher based on a closed model lacks scalability and reproducibility for the wider research community. I think experiments with open-source models can improve this paper significantly
2. **Lack of Experiments**: While the vSearcher is an interesting approach, its validity would be more convincing if it were tested more rigorously.
3. **Limited Benchmark Coverage**: The O3-Bench has a relatively small scale (185 images, 318 QA samples). This is a limitation, especially since the chart images are sourced from the existing MME-RealWorld dataset.

**Questions:**

1. Could you provide results for the vSearcher using different model sizes, or paired with open-source vReasoner models (such as Qwen2.5-VL or InternVL3)?
2. The paper states that the O3-BENCH chart images are from MME-RealWorld. Could you clarify in more detail how the tasks (i.e., the questions) for this domain differ from the original MME-RealWorld chart domain?
3. Does the number of layouts in the O3-Bench data affect model performance? (For example, is there a correlation between a higher layout count and lower model accuracy?)

---

> ### Author Response · Authors · 2025-11-25
> **Official Comment by Authors (1/2)**
>
> We are thankful for your time and constructive feedback, especially the ones on additional experiments which helped us significantly improve our paper. We are glad to hear that you find our approach “efficient”, “interesting and novel”, and that “O3-Bench can be beneficial to the community.”
>
> ---
>
> **(W1 & Q1) “Reliance on Closed-Source Models: … training the vSearcher based on a closed model lacks scalability and reproducibility for the wider research community. I think experiments with open-source models can improve this paper significantly.” & “Could you provide results for the vSearcher using different model sizes, or paired with open-source vReasoner models (such as Qwen2.5-VL or InternVL3)?”**
>
> Thank you for the great suggestions. We totally agree that experiments with open-source models can improve our paper. The only reason we did not include such experiments in our initial submission is that most available open models had relatively limited tool-calling abilities at that time. However, “thinking with images” is a challenging problem that has a high requirement on the models’ general reasoning and (multi-turn) tool-calling capabilities. We did experiment with open-source models early on, but their performance was not yet sufficient for our setting, which led us to use closed-source models for the main results.
>
> Fortunately, open models are rapidly improving. As shown in the general response (Table R2, copied below for your convenience), our approach already works reasonably well with a recent open model, Qwen3-VL-32B (released less than a month ago). The results show that training under the Qwen3-VL-32B vReasoner and the GPT-5-mini vReasoner both yield good performance under the Qwen3-VL-32B vReasoner during evaluation. We expect that, as open models continue to advance in tool use and general reasoning, the reliance of our approach on closed-source vReasoner models will be further mitigated.
>
> | vReasoner | vSearcher | V$^\\star$-Bench | VisualProbe-Hard | O3-Bench |
> | ------------ | ------------------------------------------- | ---------------- | ---------------- | -------- |
> | Qwen3-VL-32B | \- | 88.0 | 30.2 | 40.3 |
> | Qwen3-VL-32B | Qwen2.5-VL-7B (before RL) | 56.3 | 19.5 | 24.4 |
> | Qwen3-VL-32B | Qwen2.5-VL-7B (after RL under GPT-5-mini)   | 91.8 | 35.5 | 44.4 |
> | Qwen3-VL-32B | Qwen2.5-VL-7B (after RL under Qwen3-VL-32B) | 90.2 | 37.7 | 43.1 |
>
> Following your advice, we have also conducted experiments with vSearcher of different model sizes, and the results are provided below.
>
> | vReasoner  | vSearcher | V$^\\star$-Bench | VisualProbe-Hard | O3-Bench |
> | --- | --- | --- | --- | --- |
> | GPT-5-mini | \- | 73.8 | 26.4 | 38.1 |
> | GPT-5-mini | Qwen2.5-VL-3B  | 86.7 | 41.4 | 59.5 |
> | GPT-5-mini | Qwen2.5-VL-7B  | 86.9 | 41.2 | 63.4 |
> | GPT-5-mini | Qwen2.5-VL-32B | 86.2 | 38.1 | 60.8 |
>
> From the results, we observe that for traditional visual search tasks like V$^\star$-Bench and VisualProbe-Hard, the performance of the 3B model and the 7B model are similar. The 7B model only outperforms the 3B model on O3-Bench, suggesting that O3-Bench requires more model capacity to solve than the other two benchmarks.
>
> Interestingly, as we continue to increase the model size to 32B, we observe inverse scaling [1] on the three benchmarks. While we do not know the exact reason for this, the literature suggests that it could be due to memorization of undesired patterns in similar tasks present in the pre-training/mid-training data [1]. The 7B model we used turns out to be the best among these model sizes. We will include the above results in the next version of our paper.
>
> [1] McKenzie, I. R., A. Lyzhov, M. Pieler, A. Parrish, A. Mueller, A. Prabhu, E. McLean et al. "Inverse Scaling: When Bigger Isn’t Better." Transactions on machine learning research (2024).
>
> ---
>
> **(W2) “Lack of Experiments: While the vSearcher is an interesting approach, its validity would be more convincing if it were tested more rigorously.”**
>
> Thank you again for your constructive suggestion. Our initial submission only provided the core results of our approach. Since the initial submission, we have been continuously running supporting experiments to validate the effectiveness of our approach. The new experiments---covering results of vSearcher trained under Gemini-2.5-Flash and under different vReasoner input image resolutions, as well as more comprehensive ablation studies---are summarized in the “Updates since initial submission” section under the general response (you could also check them in the updated manuscript). Together with the results on open-source vReasoner and different vSearcher model sizes you suggested, we hope these additions could convince you of the validity of our approach. Please kindly let us know if there is any more experiment you think we can do to improve our paper.
>
> ---
>
> **(W3) “Limited Benchmark Coverage: The O3-Bench has a relatively small scale …”**
>
> Please see “Limited scale of O3-Bench” in the general response.

---

> ### Author Response · Authors · 2025-11-25
> **Official Comment by Authors (2/2)**
>
> **(Q2) “The paper states that the O3-BENCH chart images are from MME-RealWorld. Could you clarify in more detail how the tasks (i.e., the questions) for this domain differ from the original MME-RealWorld chart domain?”**
>
> The original questions from MME-RealWorld are relatively simple, usually focusing on a single value in a chart, which does not require any multi-hop reasoning. For example, the original question of MME-RealWorld for the chart we show in Figure 10 of our paper is:
>
> > What is the cost inflation rate in the General Settings section of the General Assumptions table?
>
> In comparison, the new question in O3-Bench is:
>
> > If the Condo unit's Purchase Cost is depreciated evenly over its stated operation period, what percentage does this depreciation amount account for of its Avg. Annual Rental Fee (annualized based on the Year 10 Avg. Daily Rental Fee and 30 days/month)?
>
> As illustrated in Figure 10, answering this question requires gathering detailed information from *three* different tables and connecting the information through multi-step reasoning and calculations as follows:
>
> > Step 1: From layout A, Condo Purchase Costs = 150,000 USD.
> > Step 2: From layout B, depreciation period = 15 years. Therefore, Annual depreciation = 150,000 / 15 = 10,000 USD/year.
> > Step 3: From layout C, Condo Year 10 Avg. Daily Rental Fee = 114 USD/day.
> > Step 4: Annual rental income at Year 10 rate with 30days/month = 114 * 30 * 12 = 41,040 USD/year.
> > Step 5: Percentage = (10,000 / 41,040) * 100 ≈ 24.37%.
> > Step 6: 24.37% is not listed in options A–E, so the correct choice is F.
>
> More such examples can be found on [HuggingFace](https://huggingface.co/datasets/InSight-o3/O3-Bench) where we have released O3-Bench anonymously for review. Overall, the questions of O3-Bench (chart) are significantly harder than the MME-RealWorld counterparts. This can also be seen from the following statistics:
> - The average accuracy of GPT-5-mini on O3-Bench (chart) is about 38.2%, whereas on MME-RealWorld (chart), the accuracy is about 82.4%.
> - The average number of vSearch steps of InSight-o3 on O3-Bench (chart) is about 3.1, whereas on MME-RealWorld (chart), the number is about 1.1.
> - The average response length of GPT-5-mini vReasoner on O3-Bench (chart) is about 1942.3 characters, whereas on MME-RealWorld (chart), the average response length is about 730.0 characters.
>
> The above discussion has been added to Appendix B.5. We hope these help clarify the differences in the tasks between O3-Bench and MME-RealWorld on the chart domain.
>
> ---
>
> **(Q3) “Does the number of layouts in the O3-Bench data affect model performance? (For example, is there a correlation between a higher layout count and lower model accuracy?)”**
>
> Yes, there is a correlation between a higher layout count and lower model accuracy. The Pearson correlation coefficient between the two quantities is -0.14 with a p-value of 0.01 (statistically significant). More detailed statistics are shown below:
>
> | \# of target layouts | Avg. accuracy |
> | -------------------- | ------------- |
> | $\leq$ 2                 | 66.8          |
> | 3 $\sim$ 4                  | 50.4          |
> | $\geq$ 5                 | 50.0          |
>
> ---
>
> Thank you again for taking the time to review our paper and providing helpful feedback! Does the above response address your concerns with the paper? If not, please kindly let us know what further clarification or modifications could we make to improve it.

---

> > ### Comment · Reviewer_7mrZ · 2025-11-27
> >
> > Thank you for the comprehensive response and the additional experiments.
> >
> > The inclusion of open-source vReasoner results and the analysis of different vSearcher sizes largely address my concerns regarding reproducibility. The clarification on the multi-hop nature of O3-Bench (vs. MME-RealWorld) is also convincing.
> >
> > However, I have two follow-up questions:
> >
> > 1. Qwen3-VL-32B model performance: I noticed that the Qwen3-VL-32B achieves 40.3 on O3-Bench, which appears to underperform compared to the smaller Qwen3-VL-8B model (which achieves 45.6). Do the authors have any insight into why the larger model underperforms in this specific setup?
> >
> > 2. Training Cost: Regarding the RL training pipeline for the vSearcher, could the authors provide an estimate of the API costs incurred for training a single vSearcher model using the closed-source vReasoners (GPT-5-mini and Gemini-2.5-Flash)? This information would be valuable for the community to gauge the reproducibility and resource requirements of the proposed method.

---

> ### Author Response · Authors · 2025-12-03
> **Response to follow-up questions of Reviewer 7mrZ**
>
> **(Follow-up Q1) “Qwen3-VL-32B … appears to underperform compared to the smaller Qwen3-VL-8B model ... why the larger model underperforms in this specific setup?”**
>
> Thank you for pointing this out. We investigated the issue and found that the gap is caused by the sensitivity of Qwen3-VL models to the system prompt. In our original experiments, we used the “think prompt” proposed by DeepSeek R1 to encourage the models to think before answering. This practice is common in the literature, and in our preliminary experiments it improved the performance of several models. However, this behavior does not appear to transfer to O3-Bench. When we switched from the think prompt to the default system prompt of Qwen3-VL, the performance of Qwen3-VL-32B on O3-Bench improved significantly (see the table below), while the performance of Qwen3-VL-8B dropped. Under this new setup, Qwen3-VL-32B clearly outperforms Qwen3-VL-8B on O3-Bench. We hypothesize that O3-Bench is more “out-of-distribution” for Qwen3-VL than other benchmarks like V$^\star$-Bench and VisualProbe-Hard, making the models more brittle when the system prompt deviates from the default one.
>
> Based on this observation, we also re-ran our method using the default Qwen3-VL system prompt for tool calling. We find that our method continues to provide performance gains for Qwen3-VL-32B on challenging benchmarks including VisualProbe-Hard and O3-Bench. This is consistent with our previous results.
>
> | vReasoner                              | vSearcher                                   | V$^\\star$-Bench | VisualProbe-Hard | O3-Bench |
> | -------------------------------------- | ------------------------------------------- | ---------------- | ---------------- | -------- |
> | Qwen3-VL-8B (think prompt)             | \-                                          | 86.4             | 31.6             | 45.6     |
> | Qwen3-VL-8B (default prompt)           | \-                                          | 86.4             | 31.1             | 41.2     |
> | Qwen3-VL-32B (think prompt)            | \-                                          | 88.0             | 30.2             | 40.3     |
> | Qwen3-VL-32B (default prompt)          | \-                                          | 86.0             | 28.3             | 59.6     |
> | Qwen3-VL-32B (default prompt w/ tools) | Qwen2.5-VL-7B (before RL)                   | 69.1             | 21.7             | 49.7     |
> | Qwen3-VL-32B (default prompt w/ tools) | Qwen2.5-VL-7B (after RL under GPT-5-mini)   | 90.1             | 36.8             | 61.1     |
> | Qwen3-VL-32B (default prompt w/ tools) | Qwen2.5-VL-7B (after RL under Qwen3-VL-32B) | 91.6             | 36.5             | 62.0     |
>
> Furthermore, we examined whether using the default prompt has a similarly strong effect on proprietary models such as GPT-5-mini. As shown in the table below, the default prompt only slightly improves GPT-5-mini and GPT-5-nano on O3-Bench, but leads to worse GPT-5-nano performance on VisualProbe-Hard. Overall, the new results do not change the main findings or conclusions of our paper.
>
> |                             | V$^\\star$-Bench | VisualProbe-Hard | O3-Bench |
> | --------------------------- | ---------------- | ---------------- | -------- |
> | GPT-5-mini (think prompt)   | 73.8             | 26.4             | 38.1     |
> | GPT-5-mini (default prompt) | 73.3             | 26.4             | 39.9     |
> | GPT-5-nano (think prompt)   | 64.0             | 21.7             | 28.9     |
> | GPT-5-nano (default prompt) | 64.6             | 16.0             | 30.4     |
>
> ---
>
> **(Follow-up Q2) “Regarding the RL training pipeline for the vSearcher, could the authors provide an estimate of the API costs incurred for training a single vSearcher model using the closed-source vReasoners (GPT-5-mini and Gemini-2.5-Flash)?”**
>
> During training, each problem in the training dataset takes about 2 steps on average to solve, and a visual search step on average takes about 300 output tokens. The number of input tokens is roughly 4K on average per step and most of the input tokens are cached for multi-turn conversations. The cost of answer verification is negligible compared to solving the problems in the training dataset. The total cost to train a single vSearcher model using GPT-5-mini is roughly: 150 (training steps) \* 24 (batch size) \* 8 (rollouts) \* [0.25 (\\$/1M input tokens) \* 4K (input tokens) / 1M (tokens) + 2 (\\$/1M output tokens) \* 300 (output tokens/step) \* 2 (steps) / 1M (tokens)] $\approx$ \\$60. The cost of using Gemini-2.5-Flash is slightly more expensive---about \\$75-\\$100. We note that such costs are fairly modest in typical academic and industrial research settings.

---

### Official Review · Reviewer_Pra6 · 2025-10-28

**Soundness:** 3
**Presentation:** 2
**Contribution:** 2
**Rating:** 4
**Confidence:** 4

**Summary:**

This paper proposes to augment visual-language reasoning models with the capability to perform a visual search function. In particular, it trains an additional visual search module that predicts the bounding box coordinate related to a given description (e.g., instructions derived from the input question), and feeds the retrieved image patches to pretrained reasoning models. A new multi-modal reasoning dataset is also presented, which focuses on reasoning on complex charts and digital maps. Experimental results show that the method improves the performance of several state-of-the-art reasoning models across different datasets.

**Strengths:**

(1) It is a promising direction to improve multi-modal reasoning models by incorporating external tools (i.e., the visual search module in this case), which could provide both performance boost and enhanced transparency.

(2) The use of collages for visual search training reduces the reliance on large-scale naturalistic data.

(3) The paper develops a new evaluation benchmark for multi-modal reasoning, and can benefit the development of subsequent models.

(4) The proposed method shows generalizability across several models and datasets.

**Weaknesses:**

(1)  It is not a new idea to combine VLMs with external tools (e.g., some compositional reasoning models [ref1, ref2] already explore the tool usage with reinforcement learning). The paper experiments with a single tool (i.e., visual search framed as a visual grounding task), while solving real-life problems could require diverse abilities. It is unclear whether visual search (especially when trained independently) can help generalize reasoning across different scenarios.

(2) One advantage of having an additional visual search agent is to provide a transparent interface of the decision-making process. Nevertheless, the paper only reports the accuracy on datasets, without any analysis of how the improvement is achieved with visual search.

(3) Looking at Table 1, it appears that the proposed method only shows consistent improvement on models from the GPT family, and can have negative effects on the best-performing Gemini model. Please justify the inconsistency in performance.

(4) The proposed visual search module is trained on synthetic collages created by stitching together different images. Such a paradigm ignores the contextual relationship between different regions within a visual scene, and also introduces boundary artifacts. Since the search agent is essentially trained on a visual grounding task, I wonder how it will perform when training on naturalistic grounding datasets.

(5) The new dataset contains a very limited set of stimuli (~350 images), making it difficult to be used for training or comprehensive evaluation.

[ref1] ViperGPT: Visual Inference via Python Execution for Reasoning. CVPR, 2023.

[ref2] HYDRA:AHyper Agent for Dynamic Compositional Visual Reasoning. ECCV, 2024.

**Questions:**

(1) What are the advantages of visual search over other types of external tools?

(2) How does the visual search agent help the reasoning agent? Please provide in-depth analysis of the decision-making process.

(3) Why does the model only improve the GPT models but lead to worse performance on Gemini? Is it related to the use of GPT-nano for evaluation?

(4) Please justify how the proposed training paradigm could accommodate the artifacts in synthetic data.

---

> ### Author Response · Authors · 2025-11-24
> **Official Comment by Authors (1/4)**
>
> We are thankful for your time and constructive feedback. They have greatly helped us improve our paper. We are encouraged to hear that you think we are working in a promising direction and that our work can benefit the development of subsequent models.
>
> ---
>
> **(W1.1) “It is not a new idea to combine VLMs with external tools … [ref1, ref2] already explored the tool usage with reinforcement learning”**
>
> Thank you for highlighting the pioneering work of ViperGPT [ref1] and HYDRA [ref2] on integrating VLMs with external tools. We have cited them in the related work and added a section in Appendix A to discuss them. We appreciate the opportunity to clarify the fundamental distinctions between our InSight-o3 framework and these prior works, spanning core objectives, problem scope, and technical design:
>
> - **Core objectives.** The core goal of ViperGPT and HYDRA is to decompose visual reasoning tasks into executable Python code, where Python execution acts as a core external dependency. Both frameworks rely entirely on successful code generation and execution to produce final results, with no recourse to VLMs’ inherent visual understanding. In contrast, InSight-o3 treats vSearcher (the external tool) as a complementary component: it is only invoked on-demand when the base VLM (vReasoner) requires fine-grained visual search for high-information-density regions, and the vReasoner retains the ability to complete reasoning independently when visual search is unnecessary.
> - **Problem scope.** ViperGPT and HYDRA focus on simple visual tasks for natural images (e.g., object counting, visual grounding of discrete objects). InSight-o3 targets a far more challenging problem: generalized visual search for high-information-density non-natural images (e.g., complex charts, high-resolution maps) that require multi-region evidence aggregation and multi-hop reasoning. Our external tool vSearcher is designed for relational/fuzzy/conceptual region localization (e.g., "the chart showing revenue trends over the last decade")---a capability neither ViperGPT nor HYDRA addresses.
> - **Technical design.** ViperGPT adopts a single feed-forward pipeline, where tools are invoked via hard-coded API functions to generate Python code for task execution without reinforcement learning (RL). HYDRA introduces an RL agent as a cognitive controller to select optimal instruction samples (generated by a planner module) for subsequent code generation, and relies on a State Memory Bank to store historical reasoning states. In comparison, our InSight-o3 employs a streamlined RL framework with clear task specialization: the vReasoner handles high-level abstract reasoning (e.g., map navigation logic, cross-chart data aggregation), while the vSearcher is dedicated to solving generalized visual search. Our hybrid RL training not only incentivizes the vReasoner to invoke the vSearcher properly but also ensures the vSearcher tightly collaborates with the vReasoner to meet its specific visual search needs.
>
> We believe these distinctions clearly underscore the novel contributions of InSight-o3.
>
> ---
>
> **(W1.2) “The paper experiments with a single tool … while solving real-life problems could require diverse abilities. It is unclear whether visual search (especially when trained independently) can help generalize reasoning across different scenarios.”**
>
> Thank you for this insightful comment. We fully agree that solving real-life problems could require diverse abilities such as utilizing different tools. The visual search tool (vSearcher) is, as with other kinds of tools, not a one-size-fits-all solution. However, a nice thing about tools is that they are largely independent and complementary to each other. In other words, they are not mutually exclusive---we can allow the models to use all of them. E.g., to solve a real-world programming problem, the model may first use the visual search tool to gather relevant information and then run the program execution tool to solve the problem based on the gathered information. Furthermore, the mutually independent nature of tools also allow them to be built/trained largely independently.
>
> By and large, we argue that the problems in O3-Bench are already a representative subset of real-life problems that require thinking with images. In addition, since our training data is distinct from the evaluation data, obtaining good performance on O3-Bench (and the other benchmarks) is already an indicator of good generalization across many different scenarios. That said, there are admittedly still some specific scenarios (e.g., counting a large number of objects) where visual search may currently struggle, but in general, we believe it will progressively help improve future models’ performance in most of the meaningful scenarios as it is essential to how human vision works (one focus at a time, only attending to necessary information guided by reasoning, and usually starting from global pictures down to details).

---

> ### Author Response · Authors · 2025-11-24
> **Official Comment by Authors (2/4)**
>
> **(Q1) “What are the advantages of visual search over other types of external tools?”**
>
> This is a really good question which made us think as well. Visual search is closely related to visual grounding and object detection, which are commonly framed as locating specific objects/figures in natural images based on natural language descriptions or predefined classes. Visual search generalizes this to a broader task: the target may be a general region without focusing on a single object/figure because one may not know exactly what to look at in the beginning when solving a real-life problem, e.g., find a hard, dense object in a messy room to break the window glass. To solve such a problem, an agent must first look at possible regions like “the left corner of the room” and “the area beneath the countertop” where such items may lie without knowing what the item is in advance.
>
> Another example is to find the shortest route from location A to B using a map. Here, the information is encoded purely in a visual artifact: the agent must first locate the legend area of the map on a cluttered screen or page, then visually search for the labels corresponding to location A and B, and finally scan along the possible routes to infer the shortest path. There is no obvious symbolic “API call” that can replace this process if the only available input is an image of the map.
>
> These examples also highlight where visual search offers advantages over non-visual external tools such as text retrieval, code execution, or structured route-planning APIs. When the relevant information exists only in images (maps, dashboards, GUIs, diagrams) or when the target is not specified as a known category in advance, these tools alone cannot directly access or localize it. Visual search acts as a general, perception-level tool that (i) discovers *where* to look before other reasoning or tools are applied, and (ii) supports *open-ended*, *region-level* exploration that is not restricted to a fixed label set (visual grounding datasets are less restrictive but still limited to descriptions of a certain, albeit large and fine-grained, set of objects). Visual search complements traditional tools and enables agents to operate in more realistic, unstructured environments.

---

> ### Author Response · Authors · 2025-11-24
> **Official Comment by Authors (3/4)**
>
> **(W2 & Q2) “... the paper only reports the accuracy on datasets, without any analysis of how the improvement is achieved with visual search.” & “How does the visual search agent help the reasoning agent? Please provide in-depth analysis of the decision-making process.”**
>
> Thank you very much for pointing this out. We have provided an in-depth analysis of the decision making process of InSight-o3 in our paper, and added figures comparing the reasoning traces of InSight-o3 with stand-alone vReasoner (GPT-5-mini w/o vSearcher) and vReasoner combined with vanilla Qwen2.5-VL-7B vSearcher. Please see Appendix F in the updated manuscript for details.
>
> Here, we briefly summarize the analysis of how the improvement is achieved with visual search. In the most basic setting where vReasoner does not have access to vSearcher, it often uses similar reasoning patterns as follows to reach its conclusion: “I first locate … I see … I then look for … There is … Therefore …” During this process, vReasoner often hallucinates and makes factual errors about what it sees, suggesting that it does not really see the relevant visual details clearly but still pretends so anyways.
>
> With the vanilla Qwen2.5-VL-7B vSearcher, vReasoner is able to take closer looks at regions of interest and makes less factual errors. However, the vanilla vSearcher is often unreliable, returning inaccurate/wrong crops to vReasoner or simply concluding that the target is not in the image, usually after a minimal amount of (sometimes none) reasoning. In such cases, vReasoner would eventually give up and resort to its own perception after multiple failed visual search attempts, leading to wrong final answers.
>
> In comparison, our visual search agent, InSight-o3-vS, would usually first reason about the visual search request based on the image, and then crop the candidate region to verify before returning it to vReasoner. In some cases, InSight-o3-vS is able to correct an initial bad crop after reviewing the crop:
>
> > Based on the response, the zoomed-in area does not match the intended area around grid B4. The bounding box needs to be adjusted to better capture the car parks and nearby cafe icons and labels.
>
> Finally, given the accurate crops of the requested regions, vReasoner is able to reach the correct answer step by step. More examples can be found in Figure 16-19 (Appendix F) of the updated manuscript. From those cases, we can see that vSearcher mostly helps vReasoner by (i) precisely locating the regions requested by vReasoner and (ii) presenting them to vReasoner and directing its attention to those regions, thereby reducing hallucination and facilitating evidence-based reasoning.
>
> ---
>
> **(W3 & Q3) “... the proposed method only shows consistent improvement on models from the GPT family, and can have negative effects on the best-performing Gemini model. Please justify the inconsistency in performance.” & “Why does the model only improve the GPT models but lead to worse performance on Gemini? Is it related to the use of GPT-nano for evaluation?”**
>
> Please see “Suboptimal performance of Gemini-2.5-Flash on certain benchmarks” in the general response first.
>
> Regarding the performance gap between GPT models and Gemini-2.5-Flash, we believe the reason is three-fold:
> - Their input image resolution is different. The maximum input resolution of Gemini-2.5-Flash is much higher than that of GPT models, so it can see more details than GPT models without using any visual search. This results in high baseline performance of Gemini-2.5-Flash and relatively low baseline performance of GPT models.
> - As mentioned in the general response, the tool-calling capability of Gemini-2.5-Flash seems to be worse than GPT models, especially the GPT-5 series.
> - There is a small generalization gap between GPT-5-mini and Gemini-2.5-Flash.
>
> As a result, Gemini-2.5-Flash is unable to improve the strong baseline on some of the benchmarks, and GPT models are able to significantly outperform the relatively low baseline consistently. To make this clearer, we added some footnotes around Table 1 in the updated manuscript to explain the difference in their input resolution, and also mentioned the relatively weak tool-calling ability of Gemini-2.5-Flash in the main text.
>
> Additional experiments on Gemini-2.5-Flash under the same resolution as GPT models support the above explanation. The new results in Table 1 shows that Gemini-2.5-Flash is on par with GPT-5-mini on average when neither has access to vSearcher. With vSearcher, both models significantly improve but Gemini-2.5-Flash lags behind GPT-5-mini on average. Training under Gemini-2.5-Flash mitigates the issue. Interestingly though, in Table 2 of the latest manuscript, we find that the training resolution of vReasoner has little to none impact on final performance on Gemini-2.5-Flash, showing generalization of vSearcher across different vReasoner image scales.

---

> ### Author Response · Authors · 2025-11-24
> **Official Comment by Authors (4/4)**
>
> **(W4) “The proposed visual search module is trained on synthetic collages … Such a paradigm ignores the contextual relationship between different regions … and also introduces boundary artifacts … I wonder how it will perform when trained on naturalistic grounding datasets.”**
>
> We would like to correct that our proposed visual search module is *not* only trained on synthetic collages. It is the dynamic RL component* that only uses synthetic collages. For the static RL component, we *do* use naturalistic (non-synthetic) grounding data: InfographicVQA (although they are not natural images). As shown in Table 5 of our updated manuscript, combining the static and dynamic components leads to the best overall performance.
>
> For the static component, we choose InfographicVQA because it has high-resolution, information-dense images with both structured and organic layouts (see examples in Figure 13). We do not directly use natural-image grounding datasets like RefCOCO/+/g and Flickr30K Entities because the image resolution and the information density is often low. Instead, we fill such data (Visual CoT) into collages to make the original grounding problem a compound grounding problem---the model needs to first locate the relevant sub-image within a collage and then further locate the target inside the (naturalistic) sub-image. In this sense, our visual search module is also trained on those naturalistic grounding datasets. To ensure difficulty, we filtered the original images of Visual CoT for the ones with small target regions before using them to construct the collages. Intuitively, such compound grounding data can better incentivize visual search during reinforcement learning.
>
> To validate the importance of using high-resolution, high-density images like collages for training, we conducted some experiments using a naturalistic grounding dataset. For direct comparison with our approach, we used the filtered subset of Visual CoT (which was used to fill the synthetic collages) for dynamic RL. The results are shown below, suggesting that using synthetic collages is indeed beneficial. The variants directly using the naturalistic grounding dataset underperform our approach mainly because the low-resolution, information-sparse images can hardly invoke any visual search behaviors from vReasoner (which can usually answer the questions on its own).
>
> | Static RL data | Dynamic RL data    | V$^\\star$-Bench | VisualProbe-Hard | O3-Bench |
> | -------------- | ------------------ | ---------------- | ---------------- | -------- |
> | \-             | Synthetic collages | 86.4             | 39.0             | 60.9     |
> | \-             | Visual CoT         | 85.9             | 31.1             | 54.1     |
> | InfographicVQA | Synthetic collages | **86.9**             | **41.2**             | **63.4**     |
> | InfographicVQA | Visual CoT         | 84.1             | 36.8             | 58.5     |
>
>
> \* Note: we find the words “static’’ and “dynamic” a bit incompatible with the concept of reinforcement learning. To better reflect the core idea and avoid confusion, we have renamed “dynamic RL” and “static RL” as “in-loop sub-agent RL” and “out-of-loop sub-agent RL” in our paper, respectively.
>
> ---
>
> **(Q4) “Please justify how the proposed training paradigm could accommodate the artifacts in synthetic data.”**
>
> Apart from the reasons we provided in our response to (W4), we would like to note that there are more similarities between synthetic collages and naturalistic data like composite charts and maps than it appears. For example, many charts have clear layout borders, and maps have legend/information boxes, whose boundary artifacts resemble that of the synthetic data. The image labels serve as additional cues (for region localization) which are sometimes also present in charts and maps, e.g., figure titles and map markers.
>
> ---
>
> **(W5) “The new dataset contains a very limited set of stimuli …”**
>
> Please see “Limited scale of O3-Bench” in the general response.
>
> ---
>
> Thank you again for taking the time to review our paper and providing helpful feedback! Does the above response address your concerns with the paper? If not, please kindly let us know what further clarification or modifications could we make to improve it.

---

### Official Review · Reviewer_EKkk · 2025-10-30

**Soundness:** 3
**Presentation:** 2
**Contribution:** 2
**Rating:** 4
**Confidence:** 4

**Summary:**

This paper addresses the limitations of current multimodal agents in visual reasoning by introducing a new benchmark, o3-bench, which requires models to integrate fine-grained information from multiple image regions while performing complex reasoning. To tackle this challenge, the researchers developed the InSight-o3 multi-agent framework, focusing on its vSearcher module with generalized visual search capabilities. Trained via reinforcement learning, vSearcher can locate conceptual visual regions based on natural language instructions and, as a plug-and-play component, significantly enhances the performance of state-of-the-art models across multiple benchmarks.

**Strengths:**

1. The paper clearly identifies a specific weakness in current multimodal agents—their inability to perform complex reasoning that requires integrating fine-grained visual details. It presents a challenging benchmark (o3-bench) designed explicitly to measure this underdeveloped capability.

2. The proposed InSight-o3 framework presents a sophisticated multi-agent architecture that decomposes the complex problem into specialized sub-tasks (reasoning and search).

**Weaknesses:**

The claim that the framework's search steps decrease with increasing resolution is not sufficiently supported, as the reported variations across resolutions are minimal. This suggests the search pattern may be overly reliant on the characteristics of the training data, raising concerns about its scalability and effectiveness in real-world, multi-step search-and-reasoning tasks involving high-resolution images.

The framework's performance on powerful yet tool-agnostic models like GPT-4o and Gemini-2.5 (as seen in VisualProbe-Hard and MME-RW-Lite results) is suboptimal. This indicates a strong dependency on models pre-equipped with tool-calling capabilities trained with multi-turn RL, highlighting a limitation in its general applicability. Further optimization is required to improve its stability and performance across a wider range of model architectures.

The ablation study on hybrid RL training reveals a trade-off: while combining static and dynamic RL improves performance on the proprietary O3-bench, it results in higher per-step inference latency without a clear balance of their respective advantages (static RL's speed vs. dynamic RL's adaptability). Furthermore, the performance gains appear primarily concentrated on O3-bench, questioning the hybrid method's generalization to broader tasks.

**Questions:**

N/A

---

> ### Author Response · Authors · 2025-11-24
> **Official Comment by Authors (1/2)**
>
> We are thankful for your time and constructive feedback, especially the one about the general applicability of our approach. It prompted us to think deeper about the current limitations and the future applications of our approach. We are glad to see that you find that our paper “clearly identifies a specific weakness in current multimodal agents.”
>
> ---
>
> **(W1) “The claim that the framework's search steps decrease with increasing resolution is not sufficiently supported ... This suggests the search pattern may be overly reliant on the characteristics of the training data, raising concerns about its scalability …”**
>
> Thank you very much for your thoughtful feedback. We believe there may be a misunderstanding regarding the setup in Table 5. The table reports the performance and *average* number of visual search steps of InSight-o3 under different **maximum input resolutions of vSearcher**. The relatively small differences in search steps across resolutions are in fact *expected*: many visual search requests issued by vReasoner target only a coarse or rough region, which typically does not require very high image resolution for vSearcher to locate. In addition, the average image resolution of V$^\star$-Bench is 2246 x 1538 (~3.5M pixels, some of the images are even smaller), so increasing the maximum image resolution has limited practical effect in this setting. For these reasons, the trend in Table 5 is not indicative of the underlying search pattern or of potential scalability issues.
>
> Finally, we would like to note that the evaluation benchmarks in our paper span a wide range of image resolutions (from 3.5M to 18.3M average pixels), and the strong performance of our method on most of these benchmarks provides empirical evidence of its scalability and effectiveness on real-world, multi-step search-and-reasoning tasks involving high-resolution images.
>
> ---
>
> **(W2) “The framework's performance on powerful yet tool-agnostic models … is suboptimal. This indicates a strong dependency on models pre-equipped with tool-calling capabilities trained with multi-turn RL, highlighting a limitation in its general applicability.”**
>
> Please see “Suboptimal performance of Gemini-2.5-Flash on certain benchmarks” in the general response first.
>
> Regarding the general applicability of our approach, we would like to offer a few clarifications. We agree that the effectiveness of our method currently relies in part on the tool-calling capabilities of vReasoner. As discussed in the paper, “thinking with images” is a challenging problem, and at the time of our original submission, most available open models had relatively limited tool-calling abilities. We did experiment with open models early on, but their performance was not yet sufficient for our setting, which led us to use proprietary models for the main results.
>
> That said, open models are rapidly improving. As shown in the general response (Table R2), our approach already works reasonably well with a recent open model, Qwen3-VL-32B, which was released after our ICLR submission. We expect that, as open models continue to advance in tool use and general reasoning, this dependency will become less restrictive in practice. In addition, our framework naturally provides a way to construct high-quality SFT data for “thinking with images,” which can in turn be used to further improve the tool-calling and image reasoning abilities of open models. We view this as an important step toward building open multimodal agents with more reliable and useful thinking-with-images capabilities.

---

> ### Author Response · Authors · 2025-11-24
> **Official Comment by Authors (2/2)**
>
> **(W3) “The ablation study on hybrid RL training reveals a trade-off … it results in higher per-step inference latency without a clear balance of their respective advantages (static RL's speed vs. dynamic RL's adaptability) ... the performance gains appear primarily concentrated on O3-bench, questioning the hybrid method's generalization to broader tasks.”**
>
> We agree that there is an inherent trade-off between training efficiency and final performance, and InSight-o3 is intentionally designed to prioritize final performance. In practice, training InSight-o3 with the proposed hybrid RL scheme requires only about 320 GPU-hours (i.e., 40 hours on 8 GPUs, or 10 hours on 32 GPUs), which we consider modest and affordable in typical academic and industrial settings; thus training efficiency is not the primary limiting factor in our design.
>
> Within this design, the dynamic RL component* does make training slower compared to using only the static component, but it consistently improves overall performance and provides a more general mechanism for sub-agent RL training. The dynamic component can be directly reused to train sub-agents for other tasks, whereas the static component is more task-specific.
>
> To substantiate the benefit of this hybrid RL design, (i) we add ablations under a smaller vReasoner (GPT-5-nano) and on three more benchmarks (HR-Bench-4K, VisualProbe-Hard, and MME-RealWorld); and (ii) we refine the original results by averaging over three random trials (instead of a single run). These updates make the advantage of the hybrid RL approach clearer and more statistically stable. Below, we show the updated results of InSight-o3 under GPT-5-mini. Some of the key results below are also shown in Table 5 (originally Table 2) of our latest manuscript.
>
> | Dynamic | Static | V\*-Bench | HR-Bench-4K | VisualProbe-Hard | MME-RealWorld-Lite | O3-Bench |
> | ----------------- | -------------------- | --------- | ----------- | ---------------- | ------------------ | -------- |
> | \-                | \-                   | 80.6      | 72.0        | 37.7             | 56.1               | 52.3     |
> | Yes               | No                   | 86.4      | 84.0        | 39.0             | 57.0               | 61.9     |
> | No                | Yes                  | 84.8      | 84.5        | **41.2**             | 58.6               | 60.8     |
> | Yes               | Yes                  | **86.9**      | **86.7**        | **41.2**             | **59.0**               | **63.4**     |
>
> In many practical scenarios, the usefulness of a model is ultimately determined by its final performance, while training efficiency can often be improved later through engineering efforts (e.g., replacing external APIs with locally hosted models, improving implementation-level concurrency), which are beyond the scope of this paper. At the same time, if efficiency is the primary concern, our results show that using only the static RL component still yields reasonably strong performance.
>
> Finally, we would like to note that the training data we used are largely distinct from those of the evaluation benchmarks (including O3-Bench). Therefore, obtaining good performance under such distribution gaps already suggests good generalization of our method. We hope these additional results and clarifications make the design choice behind our approach more transparent, and we will incorporate a more explicit discussion of this trade-off in the revised manuscript.
>
> \* Note: we find the words “static’’ and “dynamic” a bit incompatible with the concept of reinforcement learning. To better reflect the core idea and avoid confusion, we have renamed “dynamic RL” and “static RL” as “in-loop sub-agent RL” and “out-of-loop sub-agent RL” in our paper, respectively.
>
> ---
>
> Thank you again for taking the time to review our paper and providing helpful feedback! Does the above response address your concerns with the paper? If not, please kindly let us know what further clarification or modifications could we make to improve it.

---

### Official Review · Reviewer_CxbX · 2025-10-31

**Soundness:** 3
**Presentation:** 3
**Contribution:** 3
**Rating:** 8
**Confidence:** 3

**Summary:**

This paper addresses the problem of using an LLM reasoning with a specialist LLM visual reasoner which can be called as a tool. The paper makes two contributions: (1) a new, complex, yet quite small dataset of  questions which requires multi-modal reasoning. Half of these are complex questions about maps, as well illustrated in Fig 1, requiring zooming in and reading the legends of the maps. (2) optimize a special visual searcher (i.e. visual LLM) in the context of a strong LLM-based reasoner. The second contribution seems the largest. They formulate the visual reasoning problem as a cooperative task between a visual reasoner (a pre-trained LLM) and a visual searcher (a visual LLM which is called as a tool by the visual reasoner and optimized with RL for the task at hand). Since optimization in a cooperative setting is really hard, the paper chooses to fix the visual reasoner to GPT-5-mini-2025-08-07  and optimize the visual searcher (i.e.  Qw2n2.5-VL-Instruct in this paper).

On standard academic benchmarks such as V*-Bench, VisualProbe, and MME-RealWorld results demonstrate that they can adding their now-trained visual searcher to several GPT variants and Gemini Flash and obtain significant improvements on top of using those models without their visual searcher. The fact that is works on many visual reasoners shows generalization.  Additionally, their best result with GPT-5-mini and Gemini-2.5-Flash outperforms the state-of-the-art on the majority of datasets. Note that while Mini-O3 seems to be stronger, AFAIK this appeared on ArXiv only recently and should be counted as contemporary work.

**Strengths:**

* This paper decouples visual reasoning from visual search, leading to a modular system which is more understandable and whose components can be trained independently.
* Results are state-of-the-art.
* Good improvement using Gemini as a visual reasoner suggests that even though their visual search model was optimized on GPT-5-mini, it generalizes to multiple visual reasoners.
* Ablation shows good improvement of RL fine-tuning and training efficiency benefit of using their static RL setup in conjunction with the dynamic RL setup.
* New and complex visual reasoning dataset.

**Weaknesses:**

* The new dataset is rather small and limited in domain.
* Minor: Table 1 is a bit confusing: the bottom part seems the most important while the top part is hardly discussed and only used for context; I would suggest to show the bottom part either on top or maybe better shown separately as Figure 1.
* Minor: it is unclear which tools are used by Qwen2.5-VL. Anything other than 'crop'?
* Minor: specialization of agents and sub-agents in agentic frameworks has been shown to work in prior art. Examples are [Socratic Models, Zeng et al., ICLR'23] and [HAMMR, Castrejon et al., NeurIPS workshop 2024]. Would be good to cite some of these works.

**Questions:**

I don't really have specific questions.

---

> ### Author Response · Authors · 2025-11-24
>
> We are thankful for your time and constructive feedback, especially the minor points which helped us improve our paper on the important details. We are glad to see that you found our work a good contribution to the field.
>
> ---
>
> **(W1) “The new dataset is rather small and limited in domain.”**
>
> Please see “Limited scale of O3-Bench” in the general response.
>
> ---
>
> **(W2) “Minor: Table 1 is a bit confusing … I would suggest to show the bottom part either on top or maybe better shown separately as Figure 1.”**
>
> Thanks for your great suggestion. We marked the top part of Table 1 in gray to indicate that it is mostly for context. We find this to be better than showing the bottom part on top (which would seem to trivialize the context) and showing the two parts separately (which would take additional space and make comparison less convenient). Please kindly let us know if this is better in your opinion as well.
>
> ---
>
> **(W3) “Minor: it is unclear which tools are used by Qwen2.5-VL. Anything other than 'crop'?”**
>
> Thank you very much for pointing this out. It is indeed unclear which tools are used as we only very briefly mentioned it in the paper. To clarify, we only allow Qwen2.5-VL to use the most essential “crop” tool. Our framework, however, does not impose such a constraint in general. It is easy to incorporate other kinds of tools for vSearcher to use in the future. We have added a footnote in the “reward design” paragraph for clarification.
>
> ---
>
> **(W4) “Minor: specialization of agents and sub-agents in agentic frameworks has been shown to work in prior art ... Would be good to cite some of these works.”**
>
> Absolutely. We have cited some of these works including the two you mentioned in the introduction of our paper (the part is highlighted in blue). You can also find a more comprehensive discussion in Appendix A. Thank you for letting us know about these important relevant works. Socratic Models (SMs) [1] proposed a modular framework that composes domain-specialized pretrained models (e.g., visual-language models, large language models, audio-language models) as sub-agents, with language serving as a universal intermediate representation to enable zero-shot multimodal reasoning. HAMMR [2] further formalized hierarchical specialization for generic VQA, with a high-level dispatcher agent routing queries to task-specific sub-agents. This design reduced reasoning confusion in naive LLM plus tools approaches, yielding a 16.3% performance gain. These work empirically validated the specialization of agents and sub-agents as an effective paradigm for enhancing multimodal reasoning capabilities, supporting InSight-o3 as a way forward to building generally intelligent multimodal agents.
>
> [1] Andy Zeng, Maria Attarian, Krzysztof Marcin Choromanski, Adrian Wong, Stefan Welker, Federico Tombari, Aveek Purohit, Michael S Ryoo, Vikas Sindhwani, Johnny Lee, et al. Socratic models: Composing zero-shot multimodal reasoning with language. In The Eleventh International Conference on Learning Representations, 2023.
>
> [2] Lluis Castrejon, Thomas Mensink, Howard Zhou, Vittorio Ferrari, Andre Araujo, and Jasper Uijlings. Hammr: Hierarchical multimodal react agents for generic vqa. In NeurIPS 2024 Workshop on Compositional Learning: Perspectives, Methods, and Paths Forward, 2024.
>
> ---
>
> Thank you again for taking the time to review our paper and providing helpful feedback! Does the above response address your concerns with the paper? If not, please kindly let us know what further clarification or modifications could we make to improve it.

---

### Author Response · Authors · 2025-11-24
**General response (1/5)**

We would like to thank all the reviewers for their constructive feedback and valuable insights. They have helped us to improve our paper in many meaningful ways! In addition, we are really glad that Reviewer CxbX finds our work a good contribution to the field, and that Reviewer EKkk thinks our paper “clearly identifies a specific weakness in current multimodal agents.” We are also greatly encouraged by Reviewer Pra6 who feels that we are working on a “promising direction,” and Reviewer 7mrZ who thinks our approach is “efficient”, “interesting and novel”, and that “O3-Bench can be beneficial to the community.” **We have released [O3-Bench on HuggingFace](https://huggingface.co/datasets/InSight-o3/O3-Bench) (properly anonymized for double-blind review).** Welcome to check it out ;)

Since the initial submission, we have been continuously working on improving and refining our paper. In this general response, we will first address some general concerns of the reviewers, and then summarize the main updates we made to our manuscript since the initial submission (the updates are also **highlighted in blue** in the updated manuscript).

---

> ### Author Response · Authors · 2025-11-24
> **General response (2/5)**
>
> # Response to general concerns
>
> ### Limited scale of O3-Bench (Reviewer CxbX, Pra6, 7mrZ)
>
> O3-Bench is limited in scale mainly because of the difficulty of (i) collecting diverse, information-dense, high-resolution images; and (ii) designing challenging, real-world questions that require interleaved reasoning with multi-hop attention to visual details, based on these complex images. **Creating such a *highly challenging* benchmark is not easy, so we choose to *prioritize quality over quantity* in order to maximize its reliability and usefulness to the research community.** To this end, all problems in O3-Bench are carefully curated and annotated by *experts* (PhDs and PhD students) in the field, following a strict multi-stage pipeline described in our paper. On average, it took **more than 20 minutes** to manually annotate a *single* example (designing/checking the QAs and writing the explanations) and **another 15-20 minutes** per example to cross-check (two independent rounds) for image quality, problem difficulty, and answer/explanation correctness. In particular, we have taken great care in ensuring that every question is *clear*, *unambiguous* (with a definite, unique answer), and requires looking at *multiple* visual details at *distinct* regions of the images. These are extremely time-consuming and often overlooked by existing benchmarks.
>
> **The above qualities make O3-Bench a *unique* benchmark for thinking with images, and a *significant addition* to the current suite of multimodal understanding benchmarks.** In Table R1, we compare O3-Bench with related benchmarks and highlight their key differences. Compared with the most closely related MME-RealWorld, we note that on the overlapping domains (i.e. chart & map): (i) **the average image resolution** of O3-Bench is *significantly higher* (4602 x 3967 of ours *vs* 1875 x 1269 of MME-RealWorld); (ii) **the average accuracy** of GPT-5-mini on O3-Bench is much lower (38.1% of ours *vs* 83.8% of MME-RealWorld); and (iii) **the average number of visual search steps** produced by InSight-o3 for O3-Bench is *2.9$\times$* that of MME-RealWorld. In addition, O3-Bench provides layout boxes and detailed explanations for each QA pair, while most of the benchmarks do not. The explanations can help the community easily verify the correctness of the answers. By highlighting these differences, we hope the reviewers can now be assured that O3-Bench is of exceptional quality and much harder to solve than the other related benchmarks. For clarity, we have added Table R1 and some further discussion on those benchmarks in Appendix J.
> &nbsp;
> &nbsp;
> &nbsp;&nbsp;&nbsp;&nbsp;&nbsp;&nbsp;&nbsp;&nbsp;&nbsp;&nbsp;&nbsp;&nbsp;&nbsp;&nbsp;&nbsp;&nbsp;&nbsp;&nbsp;&nbsp;&nbsp;&nbsp;&nbsp;&nbsp;&nbsp;&nbsp;&nbsp;&nbsp;&nbsp;&nbsp;&nbsp;&nbsp;&nbsp;&nbsp;&nbsp;&nbsp;&nbsp;&nbsp;&nbsp;&nbsp;&nbsp;&nbsp;&nbsp;&nbsp;&nbsp;&nbsp;&nbsp;&nbsp;&nbsp;&nbsp;&nbsp;**Table R1. Comparison of O3-Bench and related benchmarks.**
>
> | Benchmark | \# of QAs | Image domains | Layout/target boxes | Detailed explanations | Multi-hop required | Avg. resolution (all / chart & map) | Avg. accuracy of GPT-5-mini (chart & map) | Avg. # of vSearch steps (chart & map) |
> | --- | --- | --- | --- | --- | --- | --- | --- | --- |
> | V$^\\star$-Bench | 191 | natural image (100%) | Yes | No | No | 2246 x 1583 / - | \- | \- |
> | Tree-Bench | 405 | natural image (100%) | No | No | No | 2152 x 1615 / - | \- | \- |
> | VisualProbe-Hard | 106 | natural image (100%) | No | No | No | 4944 x 3980 / - | \- | \- |
> | HR-Bench-4K | 200\* | natural image (89%), chart (5%), map (6%) | No | No | No | 4024 x 3503 / 4032 x 2509 | 79.6 | 2.3 |
> | MME-RealWorld | ~29K | natural image (>60%), chart (25%), map (6%), etc. | No | No | No | 2708 x 1844 / 1875 x 1269 | 83.8$^\dagger$                           | 1.0 |
> | O3-Bench | 318 | chart (43%), map (57%) | **Yes** | **Yes** | **Yes** | 4602 x 3967 / **4602 x 3967** | **38.1** | **2.9** |
>
> \* ~200 distinct QA pairs. The original paper reported 800 but most of them are the same questions with scrambled options.
> $^\dagger$ Based on 500 random samples.

---

> ### Author Response · Authors · 2025-11-24
> **General response (3/5)**
>
> **Apart from quality, the scale of O3-Bench is on par with most of the multimodal understanding benchmarks for fine-grained perception**, e.g., V$^\star$-Bench (191), Tree-Bench (405), VisualProbe-Hard (106), HR-Bench-4K (200*), and commonly-used benchmarks in other multimodal research areas; to list a few: Math: MathVision test-mini (304), MathVista test-mini (1K); VQA: RealWorldQA (765); Embodiment/Spatial Understanding: ERQA (400), RefSpatial-Bench (277); Agent: MIA Bench (400), OSWorld (389), AndroidWorld (116); Fine-grained Perception: V$^\star$-Bench. These benchmarks were used to evaluate the performance of Qwen3-VL by the Qwen team. As with O3-Bench, these benchmarks are relatively small mainly because of the difficulty in data collection. Nevertheless, they have served the timely purpose of evaluating frontier models in the respective fast-developing areas.
>
> Finally, we would like to further clarify why O3-Bench only focuses on composite charts and digital maps. First, they are *representative* of most use cases of thinking with images in the digital domain (as opposed to the natural domain). More specifically, composite charts represent *structured* images (with clear delineations between different layout regions) and often require more *abstract* reasoning (e.g., computing the difference of two quantities). On the other hand, digital maps represent images with less structure and more *organic* layouts. They usually require more *visual* reasoning (e.g., finding the shortest route from location A to B). Together, we argue that **O3-Bench is generally sufficient for evaluating the thinking-with-image capability of current multimodal models** in the digital domain. As for the natural domain, there are already a number of high-quality benchmarks for thinking with natural images (listed in Table R1). For these reasons, we hope the reviewers can see that the focus of O3-Bench is more of a strength than a weakness; it precisely fills the gap of existing benchmarks while striking a balance between evaluation efficiency and generality.
>
> To conclude, despite its limited scale, we firmly believe O3-Bench will still be of great use to the research community, and truly hope it can benefit future multimodal research. As of now, we have expanded O3-Bench to 345 QA pairs and plan to gradually scale up O3-Bench further upon publication.

---

> ### Author Response · Authors · 2025-11-24
> **General response (4/5)**
>
> ### Suboptimal performance of Gemini-2.5-Flash on certain benchmarks (Reviewer EKkk, Pra6)
> We thank the reviewers for raising this important concern. Indeed, the performance of Gemini-2.5-Flash (and GPT-4o) with vSearcher are suboptimal on some of the benchmarks. Before getting into the reasons for such results, we would like to gently point out that **the observed performance drops are mostly very small, and negligible compared to the significant performance gains on the other benchmarks**. For example, the performance drop of Gemini-2.5-Flash on MME-RealWorld-Lite and HR-Bench-4K is only 0.1% and 1.2%, respectively; and the performance drop of GPT-4o thereon is just 1.1%. Meanwhile, the performance gains on the other benchmarks are mostly around 6-12%. The *only* bad case that stands out is Gemini-2.5-Flash on VisualProbe-Hard with a 3.4% drop.
>
> There are multiple factors contributing to the suboptimal performance of Gemini-2.5-Flash on VisualProbe-Hard (and some other benchmarks):
> - Gemini-2.5-Flash is good at fine-grained perception when it can see images clearly on its own at a high resolution, but not so good at tool-calling to gather more visual information and perform multi-turn interleaved reasoning/perception (compared with GPT-5-mini). This can be seen from the fact that its performance on VisualProbe-Hard suddenly drops from 39.4% to 31.4% (and from 17.9% to 16.7% at a lower resolution) when allowed to call Qwen2.5-VL-7B vSearcher, while we see a significant performance improvement (from 26.4% to 41.2%) for GPT-5-mini.
> - The benchmarks are relatively simple in the sense that they barely require any multi-step reasoning based on *multiple* visual details, so vSearcher is less useful. In comparison, the impact of using vSearcher on O3-Bench is significantly more positive, reflecting the difficulty of O3-Bench.
> - There is a small generalization gap from GPT-5-mini to Gemini-2.5-Flash. This can be seen from the new results in Table 1 showing that training under Gemini-2.5-Flash (instead of GPT-5-mini) can further improve the performance by 1-3% on average over the benchmarks.
>
> In summary, the observed performance drops are largely due to limitations of Gemini-2.5-Flash and the relevant benchmarks. Thankfully, the frontier MLLMs are quickly improving and many of them are already equipped with general tool-calling capabilities trained with multi-turn RL. **We experimented with the latest Qwen3-VL-32B model** (which was released after we submitted our paper) and found **our approach is able to further improve its performance, as shown in the table below.** It is clear that the models will continue improving and have better tool calling abilities, making vSearcher even more useful in the future.
> &nbsp;
> &nbsp;
> &nbsp;&nbsp;&nbsp;&nbsp;&nbsp;&nbsp;&nbsp;&nbsp;&nbsp;&nbsp;&nbsp;&nbsp;&nbsp;&nbsp;&nbsp;&nbsp;&nbsp;&nbsp;&nbsp;&nbsp;&nbsp;&nbsp;&nbsp;&nbsp;&nbsp;
> **Table R2. Performance of InSight-o3 under Qwen3-VL-32B vReasoner.**
> | vReasoner | vSearcher | V$^\\star$-Bench | VisualProbe-Hard | O3-Bench |
> | --- | --- | --- | --- | --- |
> | Qwen3-VL-32B | \- | 88.0 | 30.2 | 40.3 |
> | Qwen3-VL-32B | Qwen2.5-VL-7B (before RL) | 56.3 | 19.5 | 24.4 |
> | Qwen3-VL-32B | Qwen2.5-VL-7B (after RL under GPT-5-mini)   | 91.8 | 35.5 | 44.4 |
> | Qwen3-VL-32B | Qwen2.5-VL-7B (after RL under Qwen3-VL-32B) | 90.2 | 37.7 | 43.1 |
>
> ---
>
> **Update on the Qwen3-VL-32B results.** In response to our rebuttal, Reviewer 7mrZ pointed out that Qwen3-VL-32B appears to underperform compared to the smaller Qwen3-VL-8B model. We investigated this issue and found that the gap is caused by the sensitivity of Qwen3-VL models to the system prompt. With the default system prompt, the performance of Qwen3-VL-32B significantly improves from 40.3 to 59.6, outperforming Qwen3-VL-8B on O3-Bench. Meanwhile, our method continues to provide performance gains for Qwen3-VL-32B under this setup. As we further experimented with proprietary models such as GPT-5-mini, we note that the new results do not change the main findings or conclusions of our paper. For more details, please refer to the our response to the follow-up questions of Reviewer 7mrZ at the bottom of this page.

---

> ### Author Response · Authors · 2025-11-24
> **General response (5/5)**
>
> # Updates since initial submission
>
> ### Updates related to reviewers’ concerns
>
> - **More results in Table 1.** To better demonstrate the effectiveness of our approach: (i) We further include results of vSearcher trained under Gemini-2.5-Flash, and find that it further improves our previous results (trained under GPT-5-mini) and that it generalizes under GPT-5-mini as well. (ii) We experimented with a lower image resolution budget for Gemini-2.5-Flash (same budget as that for the GPT models, as per OpenAI API) and found our vSearcher offers even more significant performance improvement, reducing the gap from Gemini-2.5-Flash with high image resolution budget. These updates help address W2 of Reviewer EKkk and W3 & Q3 of Reviewer Pra6.
> - **Additional results of ablation study on hybrid RL training (Table 5).** To validate the proposed hybrid RL training, we further include results under a much smaller vReasoner model (GPT-5-nano) and results on the challenging VisualProbe-Hard benchmark. The old results are also refined, now averaged over three random trials (previously only one), showing clearer differences between the settings. These updates help address W3 of Reviewer EKkk.
> - **More comprehensive related work discussion (Appendix A.3).** We added some discussion on the related works suggested by Reviewer CxbX and Reviewer Pra6. The most relevant works are cited in the main paper as well.
> - **More details on training data construction for the static RL component (Appendix C.2).** For clarity and future reproduction, we describe the full implementation of the training data construction pipeline and show several examples of the resulting data (Figure 13). These updates help address W4 of Reviewer Pra6.
> - **Qualitative analysis of InSight-o3 reasoning behaviors (Figure 16-19, Appendix F).** To better understand why InSight-o3 is able to improve baseline performance, we compare the behavior of InSight-o3 (GPT-5-mini + InSight-o3-vS) with two baselines: (i) GPT-5-mini and (ii) GPT-5-mini + Qwen2.5-VL-7B. This comparative qualitative analysis is based on three examples of O3-Bench, showing that our vSearcher is able to produce much more accurate crops for all the visual requests of vReasoner. In particular, we show a case where the vSearcher is able to realize and recover from a bad initial crop. These updates help address W2 & Q2 of Reviewer Pra6.
> - **Discussion of benchmarks related to O3-Bench (Appendix J).** To better demonstrate the novelty and significance of O3-Bench, we provide more information about the related benchmarks of O3-Bench and highlight their key differences with O3-Bench (Table 7 and 8). These updates help address W1 of Reviewer CxbX, W5 of Reviewer Pra6, and W3 of Reviewer 7mrZ.
>
> ### Additional updates
>
> - **More results of recent work in Table 1.** To provide a better context, we added some more results for the recent works: (i) on O3-Bench, Mini-o3 only shows modest performance on par with Qwen2.5-VL-7B; and (ii) Qwen3-VL-8B considerably outperforms Qwen2.5-VL-7B across all the benchmarks, achieving the state of the art of open models on O3-Bench.
> - **A new table (Table 2)** showing more results on the impact of vReasoner input image resolution on the performance of InSight-o3. The table presents the performance of Gemini-2.5-Flash (+vSearcher) under different training/test image resolution budgets (for Gemini-2.5-Flash). We find that training under a much lower budget has almost no impact on test performance under a higher budget (and vice versa), showing generalization across different image scales.
> - **Ablation study on reward design and advantage estimation (Table 3 & Figure 3).** To better understand the impact of different design choices we made for our training algorithm, we experimented with ablated variants of reward design and advantage estimation. Results in Table 3 show that our proposed setting outperforms all the variants with a small average lead on three benchmarks. Their training dynamics are shown in Figure 3. As vSearcher gradually learns to better locate the regions described by vReasoner, we observe that both the localization IoU and the vReasoner accuracy improve.
> - **Failure cases of InSight-o3 (Figure 20-21, Appendix G).** To better document the limitation of InSight-o3, we show several typical failure cases of InSight-o3 including vReasoner hallucination due to internal knowledge bias, vReasoner jumping to conclusion without examining visual evidence, and vSearcher failure to locate a given region.
> - **Better naming for static/dynamic RL.** We find the words “static’’ and “dynamic” a bit incompatible with the concept of reinforcement learning. To better reflect the core idea and avoid confusion, we have renamed “dynamic RL” and “static RL” as “in-loop sub-agent RL” and “out-of-loop sub-agent RL”, respectively.

---

### Comment · Reviewer_CxbX · 2025-11-28
**I maintain my rating: 8 - accept, good paper (poster)**

After carefully reading the other reviews and the author response I maintain my rating.

One complaint of the other reviewers was that this paper relies either on closed-source models or models which can handle tool calls natively. I do not think this is a problem since I see tool calls with LLMs as an important nascent research field while closed-source models are currently the most powerful models while their APIs are open for everybody. Besides, the new results using Qwen3-VL by the authors makes this possible objection go away completely.

Another observation was that using LLMs as agents with tools is not new. But in my opinion this should be seen as a *research field*, not a method. And this paper is clearly advancing this field in the right direction: it has good ideas with strong experimental results. In short, I am quite confident in maintaining my 'accept' rating.

---

### Author Response · Authors · 2025-12-03
**Rebuttal summary for AC**

We would like to thank the AC for their time and for reviewing our work. Due to the circumstances this year, we understand that the AC must work under a very tight schedule. To assist with a timely and accurate evaluation of our work, below we briefly summarize the key contributions of our work and the highlights of our rebuttal.

In this work, we present (i) a novel, challenging, high-quality benchmark named **O3-Bench**, carefully designed to evaluate the thinking-with-image capability of multimodal models, filling the gap of existing benchmarks which focus on perception instead of interleaved perception/reasoning; and (ii) a multi-agent framework, **InSight-o3**, that addresses thinking with images with two cooperative agents, vReasoner and vSearcher, achieving the state of the art on a wide range of challenging benchmarks including O3-Bench.

During the review process, the reviewers provided valuable feedback on our paper, and we have incorporated the feedback into the manuscript and addressed all the related concerns with detailed explanations and supporting experiments. The reviewers’ responses to our rebuttal are clearly positive:

- **Reviewer CxbX** maintained their rating: **8 - accept, good paper (poster)** after *“carefully reading the other reviews and the author response.”* They explicitly disagreed with several concerns raised by other reviewers and wrote that *“this paper is **clearly** advancing this field in the right direction: it has good ideas with strong experimental results. In short, I am **quite confident** in maintaining my 'accept' rating.”*

- **Reviewer 7mrZ** acknowledged *“the comprehensive response and the additional experiments”* we provided. They noted that *“the inclusion of open-source vReasoner results and the analysis of different vSearcher sizes **largely address my concerns** regarding reproducibility,”* and that our clarification of the multi-hop nature of O3-Bench (vs. MME-RealWorld) is *“**convincing**.”* These comments indicate that their main concerns have been largely resolved by the rebuttal.

- **Reviewer EKkk and Pra6** did not respond to our rebuttal. However, our rebuttal systematically addresses their points, which we believe mostly stem from clarifiable misunderstandings. We provided additional analyses and clarifications aimed precisely at these concerns.

We will now go through the main concerns of the reviewers and our rebuttal of these concerns in more detail.

---

> ### Author Response · Authors · 2025-12-03
> **Rebuttal summary for AC (cont.)**
>
> ### Limited scale of O3-Bench
> Reviewer CxbX, Pra6, 7mrZ raised concerns about the limited scale of O3-Bench. We addressed these concerns in the “Limited scale of O3-Bench” section of the general response. In short, O3-Bench prioritizes quality over quantity to maximize its utility and reliability for the research community. In the general response, we provided ample evidence that O3-Bench is of exceptional quality and presents a unique challenge that is substantially more difficult than related benchmarks (please refer to the general response for more details).
>
> Regarding scale, O3-Bench is on par with most of the multimodal understanding benchmarks for fine-grained perception and benchmarks in other multimodal research areas (see also the general response). Despite their limited scale, these benchmarks have served the timely purpose of evaluating frontier models in the respective, fast-developing areas. In this respect, we believe that O3-Bench is generally sufficient for evaluating the thinking-with-image capability of current multimodal models and that it will be useful to the research community (as also noted by Reviewer 7mrZ: *“O3-Bench can be beneficial to the community”*). To clearly present our dataset, we have released [O3-Bench](https://huggingface.co/datasets/InSight-o3/O3-Bench) on HuggingFace (properly anonymized for double-blind review).
>
> ---
> ### Suboptimal performance of Gemini-2.5-Flash on certain benchmarks
> Reviewer EKkk and Pra6 raised concerns about the suboptimal performance of Gemini-2.5-Flash on certain benchmarks. However, the observed performance drops are mostly very small (about 1%) and negligible compared to the significant performance gains (6-12%) on other benchmarks. For more discussion, please refer to the “Suboptimal performance of Gemini-2.5-Flash on certain benchmarks” section of the general response.
>
> In summary, the observed performance drops largely reflect the current limitations of Gemini-2.5-Flash, which is less effective at multi-turn tool calling and tool-assisted reasoning compared to the GPT-5 series.
> As frontier MLLMs continue to evolve, this limitation will be mitigated. Indeed, we experimented with the latest Qwen3-VL-32B model (released after our submission) and found that our approach improves its performance on the most challenging benchmarks (e.g., VisualProbe-Hard and O3-Bench), as shown in Table R2 of the general response.
>
> ---
> ### Reliance on closed-source models with tool-call capabilities
> Reviewer EKkk noted that our method relies on vReasoner models that can handle tool calls natively. While this reliance is real, we do not view it as a weakness of our method. To quote Reviewer CxbX, *“I do not think this is a problem since I see tool calls with LLMs as an important nascent research field while closed-source models are currently the most powerful models.”* In addition, our approach already works reasonably well with a recent open model, Qwen3-VL-32B, which has built-in tool-call capabilities (see our response to W2 of Reviewer EKkk for more discussion). The new experiments on Qwen3-VL-32B also address the concern of Reviewer 7mrZ who asked for experiments based on open-source vReasoner models for better reproducibility.
>
> ---
> ### Other minor concerns
> - Reviewer EKkk raised concerns about the scalability and the generalization capability of our method. These concerns largely stem from a misunderstanding of our experiment setup. We have clarified the setup in more detail and conducted additional ablation studies, which further support the generalization capability of our method (see our response to W1 and W3 of Reviewer EKkk).
> - Reviewer Pra6 questioned the novelty of combining VLMs with external tools. To address this concern, we highlighted the differences between our method and the related methods mentioned by the reviewer (see our response to W1.1 of Reviewer Pra6). Moreover, as noted by Reviewer CxbX, *“using LLMs as agents with tools … should be seen as a research field, not a method”* and *“this paper is clearly advancing this field in the right direction,”* which we believe captures the broader contribution of our work.
> - Reviewer Pra6 and 7mrZ noted the lack of supporting analysis/experiments. We have provided the requested analysis in Appendix F (see also the discussion in response to W2 & Q2 of Reviewer Pra6) and supplemented our paper with additional supporting experiments (see “Updates since initial submission” in the general response). After reading our rebuttal, Reviewer 7mrZ did not raise any further concern regarding this point in their response.
>
> To conclude, we believe our paper makes good contributions to the field, and we have systematically addressed all the main concerns raised by the reviewers. In light of the strongly positive assessment from Reviewer CxbX and the substantially alleviated concerns of Reviewer 7mrZ following our rebuttal and new experiments, we hope that this context will be helpful for the AC’s final decision.

---

### Meta-Review · Area_Chair_bFRG · 2026-01-11

**Summary:**

The main concerns from the reviewers are following:

- Reviewer **CxbX**:
  - **W1**: The proposed benchmark is of small scale.

- Reviewer **EKkk**:
  - **W1**: The claim that the framework's search steps decrease with increasing resolution is not sufficiently supported.
  - **W2**: The framework's performance on powerful yet tool-agnostic models like GPT-4o and Gemini-2.5 is suboptimal.
  - **W3**: The time cost of introducing both outer and inner loops of RL training.
  - **W4**: The generalizability to broader tasks.

- Reviewer **Pra6**:
  - **W1**: Combining VLMs with external tools has also been studied in previous researches.
  - **W2**: It is unclear whether visual search (especially when trained independently) can help generalize reasoning across different scenarios
  - **W3**: The paper only reports the accuracy on datasets, without any analysis of how the improvement is achieved with visual search.
  - **W4**: In Table 1, the proposed method only shows consistent improvement on models from the GPT family, and can have negative effects on the best-performing Gemini model.
  - **W5**: The visual search module is trained on synthetic data, it remains in doubt how it will perform when training on naturalistic grounding datasets.
  - **W6**:  The proposed benchmark is of small scale.

- Reviewer **7mrZ**:
  - **W1**: The experiments rely heavily on closed-source models.
  - **W2**: The proposed benchmark is of small scale.

**Reviewer Concerns:**

- **Concerns addressed in the rebuttal:**
  - A major concern lies in the small scale of the proposed benchmark (**W1** of Reviewer **CxbX**, **W6** of Reviewer **Pra6**, **W2** of Reviewer **7mrZ**). The authors have provided detailed clarifications, showing that the benchmark is carefully designed to capture the complexity of multimodal reasoning under the digital domain. In my view, the clarification is reasonable, and this issue is not a major one for evaluation.
  -  The authors have provided detailed responses regarding clarity and technical issues, which clarifies a significant proportion of them (*W1, W3** of Reviewer **EKkk**, **W1, W3** of Reviewer **Pra6**, **W1** of Reviewer **7mrZ**)

- **Concerns remained outstanding:**
  - A remaining significant issue lies in generalizability. Two reviewers have pointed out issues related to generalization (**W2, W4** of Reviewer **EKkk**, **W2, W4, W5** of Reviewer **Pra6**). The author responses clarified that the training data of the proposed model is different from those in the testing benchmarks, which partially justifies generalization. On the other hand,  the authors also acknowledge that the paper includes only limited task scenarios, including charts/diagrams or navigating maps. Considering the training pipeline, the generalization ability may be limited to such tasks. Furthermore, the observation that the proposed approach performs suboptimal on tool-agnostic models show that the proposed approach has relatively strong dependence on the ability of the visual reasoner

**Reviewer Scores:**

As discussed in the Reviewer Concerns, some points are indeed addressed by the author responses. I think some reviewers will indeed raise their scores (e.g. Reviewer **7mrZ**). Considering the remaining issue of generalization, the key question is whether this drawback outweigh the key strengths of the paper. In my view, the training paradigm of dividing reasoning and searching, and build inner/outer loops of RL is original, interesting, and inspiring. The current results are solid to justify the major claims.

---

### Decision · Program_Chairs · 2026-01-26

Accept (Poster)